# Queueing Matching Bandits with Preference Feedback

**Jung-hun Kim**
Seoul National University
Seoul, South Korea
junghunkim@snu.ac.kr

**Min-hwan Oh**
Seoul National University
Seoul, South Korea
minoh@snu.ac.kr

## Abstract

In this study, we consider multi-class multi-server asymmetric queueing systems consisting of $N$ queues on one side and $K$ servers on the other side, where jobs randomly arrive in queues at each time. The service rate of each job-server assignment is unknown and modeled by a feature-based Multi-nomial Logit (MNL) function. At each time, a scheduler assigns jobs to servers, and each server stochastically serves at most one job based on its preferences over the assigned jobs. The primary goal of the algorithm is to stabilize the queues in the system while learning the service rates of servers. To achieve this goal, we propose algorithms based on UCB and Thompson Sampling, which achieve system stability with an average queue length bound of $\mathcal{O}(\min\{N, K\}/\epsilon)$ for a large time horizon $T$, where $\epsilon$ is a traffic slackness of the system. Furthermore, the algorithms achieve sublinear regret bounds of $\widetilde{\mathcal{O}}(\min\{\sqrt{T}Q_{\max}, T^{3/4}\})$, where $Q_{\max}$ represents the maximum queue length over agents and times. Lastly, we provide experimental results to demonstrate the performance of our algorithms.

## 1 Introduction

Multi-class multi-server queueing systems, which have been extensively studied by [20, 19, 37, 8, 4], are motivated by real-world applications such as ride-hailing platforms, where riders are assigned to drivers. Other examples are online job markets, where applicants are recommended for employment, and online labor service markets, where tasks are recommended to freelance workers. In these systems, there are two sides: queues (agents) on one side and servers (arms) on the other. At each time step, jobs stochastically arrive in multiple queues and are then assigned to multiple servers by a matchmaking or scheduler algorithm to stabilize the systems. Importantly, the previous work assumes that the service rates for each server are known for scheduling jobs.

However, real-world observations reveal that the service rate may not be known beforehand. Therefore, learning the service rates associated with the stochastic behavior of servers is essential for real-world applications. Furthermore, the behavior of servers may depend on their (relative) preferences over assigned jobs. For example, in ride-hailing platforms where there are riders and drivers, the preferences of drivers over riders (not necessarily based on personal information but rather a rider's pick-up location, destination, etc.) may not be known in advance to riders or even the systems, necessitating the learning of drivers' preferences through preference feedback over assigned riders.

Learning service rates in an online manner, based on partial feedback from each scheduling, is highly associated with multi-armed bandit problems [33]. These problems are fundamental sequential learning tasks where, for queueing systems, a job is assigned to a server and receives feedback on whether the job is served or not. The utilization of bandit strategies for queueing systems has been recently studied by Choudhury et al. [12], Stahlbuhk et al. [48], Gaitonde and Tardos [17]; Freund et al. [15], Hsu et al. [22], Krishnasamy et al. [30, 31, 32], Sentenac et al. [46], Yang et al. [52], Huang et al. [23], which primary focus is naturally on establishing stability of the systems while learning service rates.

38th Conference on Neural Information Processing Systems (NeurIPS 2024).

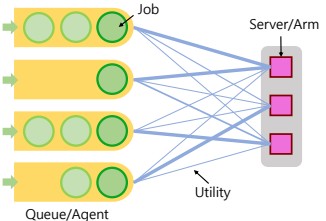 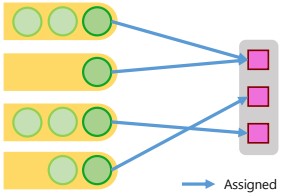 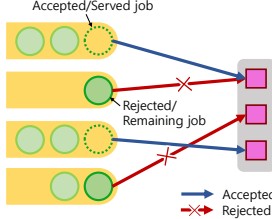

(a) Jobs randomly arrive in queues (agents) and there are unknown different values of utility between queues and servers (arms) closely related to preferences.

(b) Each queue is assigned to a server based on a scheduler.

(c) Each server accepts at most one of the assigned jobs based on its preference. A job of each accepted queue is served while jobs of rejected queues remain.

Figure 1: Illustration of queueing process with 4 queues/agents ($N = 4$) and 3 servers/arms ($K = 3$)

However, there are still gaps between the previous models and the real-world applications. In all previous work, the inherent service rates for each job-server assignment are determined regardless of other jobs assigned to the same server (or it is not allowed to assign multiple jobs to the same server). This differs from real-world scenarios where service rate may depend on (relative) preference over multiple assigned jobs, as seen in online labor markets or ride-hailing platforms. Additionally, in Freund et al. [15], Krishnasamy et al. [31, 32], which is closely related work to our study regarding multi-class multi-server queueing systems with asymmetric service rates, it was allowed to assign a job in an empty queue to a server to obtain feedback (null request), which may not be realistic. Furthermore, all of the previous work considers a simple model without a generalizable structure or utilizing features for service rates.

Another line of related work is matching bandits, in which there are two sides (agents and arms), and the behavior of arms is based on their preferences [36, 45, 35, 9, 54, 29]. However, their focus is on static settings, whereas our work considers dynamic job arrivals in queues. Additionally, it is worth mentioning online matching problems [27, 39, 40, 18, 16, 28], where the sole focus is on optimizing matching rather than learning underlying models from bandit feedback. In contrast, our study concentrates on bandit problems related to learning latent utilities by managing the tradeoff between exploration and exploitation while establishing the stability of the queueing systems. Importantly, neither previous work on matching bandits nor online matching problems addresses the stability of queueing systems, which is our primary focus.

In this work, we propose a novel and practical framework for queueing matching bandits under preference feedback. In the following, we describe our setting accompanied by an illustration in Figure 1. (a) Multiple queues (agents) and servers (arms) are involved, with jobs arriving randomly in queues. Additionally, there are unknown utility values between queues and servers regarding preferences. (b) Nonempty queue are then assigned to servers based on a scheduling policy. (c) Subsequently, each server stochastically accepts at most one of the assigned queues based on its unknown preference and serves one unit of job of the accepted queue. The rejected jobs remain in the queues. The processes of (a), (b), and (c) are repeated over time.

**Summary of Our Contributions**

- We propose a novel and practical framework for queueing matching bandits, where $N$ agents and $K$ arms are involved, and jobs randomly arrive in agents' queues each round. Subsequently, a scheduler assigns agents to arms, and the service rates for the assigned agents depend on the arms' unknown preferences over them. These service rates are modeled using a feature-based multinomial logit function. To the best of our knowledge, our paper is the first to investigate either a feature-based service function or preference feedback using a multinomial logit function for service rates.

- We propose algorithms based on Upper Confidence Bound (UCB) and Thompson Sampling (TS), which achieve stability with average queue length bounds of $\mathcal{O}(\frac{\min\{N,K\}}{\epsilon})$ for a large time horizon $T$, under a traffic slackness of $\epsilon$.

- Furthermore, the algorithms achieve sublinear regret bounds of $\widetilde{\mathcal{O}}(\min\{\sqrt{T}Q_{\max}, T^{3/4}\})$, where $Q_{\max}$ represents the maximum queue length over agents and times. It is worth mentioning that regret analysis has not been studied in the closely related queueing bandit literature such as Sentenac et al. [46], Freund et al. [15], Yang et al. [52].

- Finally, we present experimental results demonstrating the stability and regret performance of our algorithms with comparison to previously suggested methods.

## 2 Related Work

**Bandits for Queues.**    Learning the unknown service rates for queueing systems in an online manner under bandit feedback has recently gained widespread attention [12, 47, 31, 48, 22, 32, 46, 52, 23]. While Stahlbuhk et al. [47], Krishnasamy et al. [30], Choudhury et al. [12] focused on a single queue with multiple servers, aiming to efficiently assign the queue to the optimal server while learning the service rates, subsequent study of Krishnasamy et al. [32] expanded this scope to encompass multiple queues and multiple servers. However, this extension relied on a strong structural assumption that each agent possesses a unique and distinctly optimal server without sharing the same optimal server. Consequently, the algorithms employed in such scenarios do not necessarily consider the dynamic queue lengths when scheduling jobs; instead, they concentrate solely on learning service rates to achieve optimal matching. As the closest work to our study, another line of work of Sentenac et al. [46], Freund et al. [15], Yang et al. [52] has focused more on dynamic queue lengths for multi-queue and multi-server scenarios without assuming the unique and distinct optimal servers for each agent. In this context, all of the previous work aimed to achieve queue length stability.

However, to the best of our knowledge, none of the previous works has focused on a structured model with features for service rates. Furthermore, previous studies either did not allow the assignment of different types of jobs to a server simultaneously [32, 52], or, if they did, as in Sentenac et al. [46], Freund et al. [15], the inherent values of service rates for each job-server assignment remained constant regardless of the entire assignments. Additionally, the server's behavior given multiple assigned jobs was either simple or deterministic, based on the uniform-randomly selected job among the assigned jobs [46], or the job with the highest bid generated from the algorithm [15], respectively. However, these approaches may not reflect real-world scenarios, where the service rates depend on the relative preferences among the assigned jobs. Lastly, in [15, 32], it was allowed to assign a job to a server from an empty queue for obtaining feedback, which may not be realistic.

In our study, we adopt a structured model incorporating features for unknown asymmetric service rates. We enable the assignment of multiple agents to the same server, where the server stochastically selects at most one of the assigned agents based on its undisclosed (relative) preference. Also, the assignment of an agent is available only when the queue of the agent is not empty. This scenario mirrors common situations in real-world applications such as ride-hailing platforms and online labor markets where available multiple riders or tasks can be assigned to a driver or a worker, respectively, and the driver or worker behaves stochastically depending on its unknown preference.

It is noteworthy that we focus not only on the stability of the systems regarding queue lengths but also on regret against an oracle, while the closely related works by [46, 15, 52], which considered multi-class multi-server queueing systems, only addressed stability analysis. Regret analysis has only been studied under some limited scenarios, such as a single-server [23], a single-queue [48], or the strong assumption that each agent has a unique and well-separated optimal server [32].

**Matching Bandits.**    Next, we investigate two-sided matching bandits, a topic initially explored by Liu et al. [36] and subsequently studied by Sankararaman et al. [45], Liu et al. [35], Basu et al. [9], Zhang et al. [54], Kong and Li [29]. The primary goal is to minimize regret by attaining an optimal stable matching by learning agents' side preferences through stochastic reward feedback under static settings and deterministic behavior of arms with known preferences. Our study aligns with the model, wherein agents select arms based on preferences. However, our study sets itself apart from prior work on matching bandits in several key aspects. Firstly, we consider dynamic environments with job arrivals for agents. Secondly, we propose that arm behavior is stochastic, with unknown preferences, necessitating the learning of arms' preferences. Lastly, our main target is to stabilize the queue lengths in the systems rather than find stable matching for stable marriage problems.

**MNL Bandits.**    Lastly, we examine MNL bandits which were initially proposed by Agrawal et al. [6] and followed by Agrawal et al. [7], Chen et al. [11], Oh and Iyengar [43, 44]. In MNL bandits,

the goal is to select assortments of arms to maximize reward, which is based on preferences over the arms in the selected assortment. In our study, we adopt the MNL model for arms' choice preferences in dynamic systems, which, to the best of our knowledge, is the first consideration of bandits for queueing systems.

## 3    Problem Statement

There are $N$ agents (queues) and $K$ arms (servers). At each time, a job for each agent $n \in [N]$ arrives randomly following a Bernoulli distribution with an unknown arrival rate $\lambda_n \in [0, 1]$ [1]. Then at each time $t \in [T]$ where $T$ is the time horizon, each agent $n \in [N]$ is assigned to an arm $k_{n,t}^\pi \in [K]$ by a policy $\pi$. For notational simplicity, we use $k_{n,t}$ for $k_{n,t}^\pi$ when there is no confusion. Let $Q_n(t)$ be the length of the queue for jobs of agent $n \in [N]$ at the beginning of time slot $t$ in the system. We consider that at most $L$ agents can be assigned to each arm $k \in [K]$ at each time and $N \leq KL$ to ensure that all agents can be assigned to arms. Then, we define the set of agents (assortment) assigned to arm $k$ by policy $\pi$ at time $t$ as $S_{k,t} = \{n \in [N] : k_{n,t} = k, Q_n(t) \neq 0\}$, considering only available agents with nonempty queues.

Now we explain the structure of our model. Each agent $n$ has known $d$-dimensional feature information of $x_n \in \mathbb{R}^d$, and each arm $k$ has latent (unknown) parameter $\theta_k \in \mathbb{R}^d$ for preference utilities over agents. We adopt the Multi-nomial Logit (MNL) function commonly studied for preference feedback [43, 44, 6, 7, 11]. Then given assortment $S_{k,t}$, arm $k$ serves a job of agent $n \in S_{k,t}$ with a service probability (service rate) determined by the MNL model as

$$\mu(n|S_{k,t}, \theta_k) = \frac{\exp(x_n^\top \theta_k)}{1 + \sum_{m \in S_{k,t}} \exp(x_m^\top \theta_k)}.$$

We note that $\mu(n|S_{k,t}, \theta_k)$ represents the preference of arm $k$ to agent $n$ over the assigned agents $S_{k,t}$. The service rate of each $n$ depends on the assigned agents rather than static by reflecting real-world scenarios, which is different from the conventional queueing problems. Following the MNL function, the arm is allowed not to serve any agents (or allowed to serve null agent $n_0$) with the probability of $\mu(n_0|S_{k,t}, \theta_k) = \frac{1}{1 + \sum_{m \in S_{k,t}} \exp(x_m^\top \theta_k)}$. Under the MNL model for the service rate, at each time, each arm accepts at most one agent and serves that agent's job. The queue lengths for the accepted agents are each reduced by one, while the queue lengths for the refused agents remain the same.

**Objective Function.**    Here we provide the goal of this problem. For describing the stochastic process in the systems, at each time $t$, let $A_n(t) \in \{0, 1\}$ be a random variable with mean $\lambda_n$, which denotes whether a new job arrives in the queue of agent $n \in [N]$ (here, 1 denotes arrival). Also, given that $n$ is assigned to arm $k_{n,t}$, let $D_n(t|S_{k_{n,t},t}) \in \{0, 1\}$ be a random variable with mean $\mu(n|S_{k_{n,t},t}, \theta_{k_{n,t}})$, which represents whether the assigned agent $n$, given $S_{k_{n,t},t}$, is accepted by arm $k_{n,t}$ (here 1 denotes acceptance). Then the queue length of the agent $n$ evolves as $Q_n(t+1) = (Q_n(t) + A_n(t) - D_n(t|S_{k_{n,t},t}))^+$ where $x^+$ denotes $\max\{x, 0\}$ for $x \in \mathbb{R}$.

As in the previous work of queueing bandits for multiple types of agents and servers [46, 15, 52, 23], which are closely related work to ours, our primary focus is on stabilizing the dynamic systems regarding queue lengths. For analyzing the stability of the systems, we define the average queue lengths over horizon time $T$ as

$$\mathcal{Q}(T) = \frac{1}{T} \sum_{t \in [T]} \sum_{n \in [N]} \mathbb{E}\left[Q_n(t)\right].$$

Then the goal of this problem is to design an online matching algorithm to assign the agents to arms for stabilizing the systems by bounding $\mathcal{Q}(T)$. We define the system stability as follows.

**Definition 1.** *The systems are denoted to be stable when* $\lim_{T \to \infty} \mathcal{Q}(T) < \infty$.

---

[1]Our study can be easily extended to the multi-unit case where random variable $A_n(t)$ lies in $[0, M]$ for some $M > 0$ with mean $\lambda_n \in [0, M]$.

The same stability definition was considered in Neely [41, 42], Freund et al. [15], Yang et al. [52], Huang et al. [23]. It is noteworthy that according to [42, 41, 15], the system satisfying this stability condition of Definition 1 is denoted to be strongly stable.[2]

For the analyses, we first present the regularity conditions.

**Assumption 1.** $\|x_n\|_2 \leq 1$ *for all* $n \in [N]$ *and* $\|\theta_k\|_2 \leq 1$ *for all* $k \in [K]$.

**Assumption 2.** *There exists* $\kappa > 0$ *such that* $\inf_{\theta \in \mathbb{R}^d : \|\theta\|_2 \leq 1} \mu(n|S,\theta)\mu(n_0|S,\theta) \geq \kappa$ *for any* $n \in S$ *and* $S \subset [N]$.

These regularity conditions are commonly taken into account in the logistic and MNL bandit literature [13, 3, 43, 44]. We note that, in the worst-case, $1/\kappa = O(L^2)$.

We define the set of feasible disjoint assortments given any $\mathcal{N} \subseteq [N]$ as $\mathcal{M}(\mathcal{N}) = \{(S_1, ..., S_K) : S_k \subseteq \mathcal{N}, |S_k| \leq L \ \forall k \in [K], S_k \cap S_l = \varnothing \ \forall k \neq l, \bigcup_{k \in [K]} S_k = \mathcal{N}\}$. Then, we consider the condition of traffic slackness for stability as follows.

**Assumption 3.** *For some traffic slackness* $0 < \epsilon < 1$, *there exists* $\{S_k\}_{k \in [K]} \in \mathcal{M}([N])$ *such that* $\lambda_n + \epsilon \leq \mu(n|S_k, \theta_k)$ *for all* $n \in S_k$ *and* $k \in [K]$.

We note that similar slackness assumptions have been commonly considered in queueing bandits [15, 52, 23]. As $\epsilon$ decreases, achieving stability in our setting becomes more challenging. A discussion for a refined version of $\epsilon$ can be found in Appendix A.2.

## 4 Preliminary Study: An Oracle Algorithm of MaxWeight

For queueing systems, MaxWeight [50] has been proven to have optimal throughput keeping the queues in networks stable under the known service rates [51, 38, 37, 49]. Here, we analyze the oracle policy using MaxWeight under known $\theta_k$'s. We define the set of agents having non-empty queues at time $t$ as $\mathcal{N}_t = \{n \in [N] : Q_n(t) \neq 0\}$. Then, the oracle policy using MaxWeight in our setting is defined as $\{S_{k,t}\}_{k \in [K]} = \text{argmax}_{\{S_k\}_{k \in [K]} \in \mathcal{M}(\mathcal{N}_t)} \sum_{k \in [K]} \sum_{n \in S_k} Q_n(t)\mu(n|S_k, \theta_k)$, where priority in the assortment scheduling is on queues with either large queue lengths or high service rates. We provide an analysis of stability of the MaxWeight oracle for known $\theta_k$'s as follows.

**Proposition 1.** *Given the prior knowledge of* $\theta_k$ *for all* $k \in [K]$, *the average queue length of MaxWeight is bounded as* $\mathcal{Q}(T) = \mathcal{O}\left(\frac{\min\{N,K\}}{\epsilon}\right)$, *which implies that the algorithm achieves stability.*

*Proof.* The proof is provided in Appendix A.3 $\qquad\square$

We note that the term of $\min\{N, K\}$ in the average queue length is a result of the system's variance. Additionally, as $\epsilon$ decreases, the stability of the system deteriorates due to the reduced traffic slackness. Further discussion regarding $\epsilon$ can be found in Appendix A.2.

Since the oracle algorithm requires prior knowledge of $\theta_k$'s for the service rate, we cannot use it directly in our bandit setting. In the following section, we propose algorithms incorporating an efficient learning procedure to achieve stability in our setting.

## 5 Algorithms and Analyses

### 5.1 UCB-based Algorithm

We first propose an algorithm based on the UCB strategy, `UCB-QMB` (Algorithm 1). We define the negative log-likelihood as $f_{k,t}(\theta) := -\sum_{n \in S_{k,t} \cup \{n_0\}} y_{n,t} \log \mu(n|S_{k,t}, \theta)$ where $y_{n,t} \in \{0, 1\}$ is observed preference feedback (1 denotes service acceptance, and 0 denotes refusal) and define the gradient of the likelihood as

$$g_{k,t}(\theta) := \nabla_\theta f_{k,t}(\theta) = \sum_{n \in S_{k,t}} (\mu(n|S_{k,t}, \theta) - y_{n,t})x_n. \tag{1}$$

---

[2]As mentioned in [42, 15], in general, this stability condition is stronger than the mean rate stability of $\lim_{T \to \infty} \frac{\mathbb{E}[\sum_{n \in [N]} Q_n(T)]}{T} = 0$ but weaker than the uniform stability of $\mathbb{E}[\sum_{n \in [N]} Q_n(t)] \leq C$ for all $t > 0$ for some constant $C > 0$.

Then, we construct the estimator of $\hat{\theta}_{k,t}$ from online updates applying online newton step studied by [21, 53, 24, 14, 44] as $\hat{\theta}_{k,t} = \operatorname{argmin}_{\theta \in \Theta} g_{k,t-1}(\hat{\theta}_{k,t-1})^\top \theta + \frac{1}{2}\|\theta - \hat{\theta}_{k,t-1}\|^2_{V_{k,t}}$, where $\Theta = \{\theta \in \mathbb{R}^d : \|\theta\|_2 \leq 1\}$ and $V_{k,t} = \lambda I_d + \frac{\kappa}{2}\sum_{s=1}^{t-1}\sum_{n \in S_{k,s}} x_n x_n^\top$. Using the estimator, we define the UCB index for agent $n$ in assortment $S_k$ as

$$\widetilde{\mu}_t^{UCB}(n|S_k, \hat{\theta}_{k,t}) = \frac{\exp(h_{n,k,t}^{UCB})}{1 + \sum_{m \in S_k} \exp(h_{m,k,t}^{UCB})}, \tag{2}$$

where $h_{n,k,t}^{UCB} := x_n^\top \hat{\theta}_{k,t} + \beta_t \|x_n\|_{V_{k,t}^{-1}}$ with $\beta_t = C_1\sqrt{\lambda + \frac{d}{\kappa}\log(1 + \frac{tLK}{d\lambda})}$ for some constant $C_1 > 0$. We utilize the MaxWeight with the UCB indexes by setting $\lambda = 1$.

---

**Algorithm 1** UCB-Queueing Matching Bandit (UCB-QMB)
___
**Input:** $\lambda, \kappa, C_1 > 0$
**for** $t = 1, \ldots, T$ **do**
    **for** $k \in [K]$ **do**
        $\hat{\theta}_{k,t} \leftarrow \operatorname{argmin}_{\theta \in \Theta} g_{k,t-1}(\hat{\theta}_{k,t-1})^\top \theta + \frac{1}{2}\|\theta - \hat{\theta}_{k,t-1}\|^2_{V_{k,t}}$ with (1)
    $\{S_{k,t}\}_{k \in [K]} \leftarrow \underset{\{S_k\}_{k \in [K]} \in \mathcal{M}(\mathcal{N}_t)}{\operatorname{argmax}} \sum_{k \in [K]}\sum_{n \in S_k} Q_n(t)\widetilde{\mu}_t^{UCB}(n|S_k, \hat{\theta}_{k,t})$ with (2)
    Offer $\{S_{k,t}\}_{k \in [K]}$ and observe preference feedback $y_{n,t} \in \{0,1\}$ for all $n \in S_{k,t}, k \in [K]$
___

### 5.1.1 Stability Analysis of UCB-QMB

Here we provide an analysis for the stability of UCB-QMB (Algorithm 1).

**Theorem 1.** *The average queue length of Algorithm 1 is bounded as* $\mathcal{Q}(T) = \mathcal{O}\left(\frac{\min\{N,K\}}{\epsilon} + \frac{d^2 N^2 K^2}{\kappa^4 \epsilon^6}\frac{\operatorname{polylog}(T)}{T}\right)$, *which implies that the algorithm achieves stability as*

$$\lim_{T \to \infty} \mathcal{Q}(T) = \mathcal{O}\left(\frac{\min\{N,K\}}{\epsilon}\right).$$

We note that Algorithm 1 achieves the same average queue length bound of $\mathcal{O}(\frac{\min\{N,K\}}{\epsilon})$ with the oracle of MaxWeight (Proposition 1) when $T$ is large enough.

*Proof sketch.* Here we provide a proof sketch and the full version is provided in Appendix A.4. We define the set of queues $\mathbf{Q}(t) = [Q_n(t) : n \in [N]]$ and a Lyapunov function as $\mathcal{V}(\mathbf{Q}(t)) = \sum_{n \in [N]} Q_n(t)^2$. For simplicity, we use $D_n(t)$ for $D_n(t|S_{k_{n,t},t})$ and $D_n^*(t)$ for $D_n(t|S_{k_{n,t}^*,t})$ when there is no confusion. Then we analyze the Lyapunov drift as follows:

$$\sum_{t \in [T]} \mathcal{V}(\mathbf{Q}(t+1)) - \mathcal{V}(\mathbf{Q}(t))$$

$$= \sum_{t \in [T]}\sum_{n \in [N]} (Q_n(t) + A_n(t) - D_n^*(t))^2 - Q_n(t)^2$$

$$+ \sum_{t \in [T]}\sum_{n \in [N]} (Q_n(t) + A_n(t) - D_n(t))^{+2} - \sum_{t \in [T]}\sum_{n \in [N]} (Q_n(t) + A_n(t) - D_n^*(t))^2. \tag{3}$$

For the first two terms in Eq.(3), with Assumption 3, we can show that

$$\sum_{t \in [T]}\sum_{n \in [N]} \mathbb{E}[(Q_n(t) + A_n(t) - D_n^*(t))^2 - Q_n(t)^2] \leq -\sum_{t \in [T]}\sum_{n \in [N]} 2\epsilon\mathbb{E}[Q_n(t)] + 2\min\{N,K\}T, \tag{4}$$

For the last two terms in Eq.(3), we have

$$\mathbb{E}\Big[\sum_{t \in [T]}\sum_{n \in [N]} (Q_n(t) + A_n(t) - D_n(t))^{+2} - \sum_{t \in [T]}\sum_{n \in [N]} (Q_n(t) + A_n(t) - D_n^*(t))^2\Big]$$

$$\leq 2\mathbb{E}\Big[\sum_{t \in [T]}\sum_{n \in [N]} (\mu(n|S_{k_{n,t}^*,t}, \theta_{k_{n,t}^*}) - \mu(n|S_{k_{n,t},t}, \theta_{k_{n,t}}))Q_n(t)\Big] + 5\min\{N,K\}T, \tag{5}$$

where the first term of the last inequality is closely related to the regret analysis of the bandit strategy. In the following, we focus on analyzing the last term in Eq.(5). We define events $E_t^1 = \{\|\hat{\theta}_{k,t} - \theta_k^*\|_{V_{k,t}} \le \beta_t$ for all $k \in [K]\}$ and $E_{n,t}^2 = \{\max_{m \in S_{k,t}} \|x_m\|_{V_{k,t}^{-1}} \le C_2 \epsilon / 2\beta_t$ for $k = k_{n,t}^\pi\}$ for some constant $C_2 > 0$. We can show that $E_t^1$ holds with high probability so here we only consider the case when $E_t^1$ holds. Then we have

$$\mathbb{E}\Big[ \sum_{t \in [T]} \sum_{n \in [N]} \mu(n|S_{k_{n,t}^*,t}, \theta_{k_{n,t}^*}) - \mu(n|S_{k_{n,t},t}, \theta_{k_{n,t}}))Q_n(t) \Big]$$
$$\le \sum_{t \in [T]} \sum_{n \in [N]} \mathbb{E}[(\mu(n|S_{k_{n,t}^*,t}, \theta_{k_{n,t}^*}) - \mu(n|S_{k_{n,t},t}, \theta_{k_{n,t}}))Q_n(t)(\mathbb{1}(E_{n,t}^2) + \mathbb{1}((E_{n,t}^2)^c))]. \quad (6)$$

Then for the first term of Eq.(6), from the UCB strategy and $E_{n,t}^2$, we can show that

$$\sum_{t \in [T]} \mathbb{E}\Big[ \sum_{n \in [N]} (\mu(n|S_{k_{n,t}^*,t}, \theta_{k_{n,t}^*}) - \mu(n|S_{k_{n,t},t}, \theta_{k_{n,t}}))Q_n(t)\mathbb{1}(E_{n,t}^2) \Big] \le C_2 \epsilon \sum_{t \in [T]} \sum_{n \in [N]} \mathbb{E}[Q_n(t)]. \tag{7}$$

Now we provide a bound for the second term of Eq.(6). For some constant $C_3 > 0$, we can show that

$$\sum_{t \in [T]} \sum_{n \in [N]} \mathbb{E}[(\mu(n|S_{k_{n,t}^*,t}, \theta_{k_{n,t}^*}) - \mu(n|S_{k_{n,t},t}, \theta_{k_{n,t}}))Q_n(t)\mathbb{1}((E_{n,t}^2)^c)]$$
$$\le \sum_{t \in [T]} \sum_{n \in [N]} (\epsilon/C_3)\mathbb{E}[Q_n(t)] + \mathcal{O}\left( \frac{N^2 K^2 \beta_T^4}{\kappa^2 \epsilon^5} \right). \tag{8}$$

By putting the results of Eqs. (3), (4), (5), (6), (7), (8) altogether, we can obtain

$$\mathbb{E}\Big[ \sum_{t \in [T]} \mathcal{V}(\mathbf{Q}(t+1)) - \mathcal{V}(\mathbf{Q}(t)) \Big]$$
$$\le 7\min\{N, K\}T + 2(C_2 + (1/C_3) - 1)\epsilon \sum_{t \in [T]} \sum_{n \in [N]} \mathbb{E}[Q_n(t)] + \mathcal{O}(N\log(T)) + \mathcal{O}\left( \frac{N^2 K^2 \beta_T^4}{\kappa^2 \epsilon^5} \right).$$

Finally, with positive constants $C_2, C_3 > 0$ satisfying $C_2 + (1/C_3) < 1$, from $\mathcal{V}(\mathbf{Q}(1)) = 0$ and $\mathcal{V}(\mathbf{Q}(T+1)) \ge 0$, by using telescoping for the above inequality and rearrangement, we can conclude the proof by $\frac{1}{T}\sum_{t \in [T]} \sum_{n \in [N]} \mathbb{E}[Q_n(t)] = \mathcal{O}(\frac{\min\{N,K\}}{\epsilon} + \frac{d^2 N^2 K^2}{\kappa^4 \epsilon^6} \frac{\text{polylog}(T)}{T})$. $\square$

### 5.1.2 Regret Analysis of `UCB-QMB`

In addition to the stability analysis, we examine the cumulative regret of `UCB-QMB` (Algorithm 1). The regret is defined as the discrepancy between the performance of the oracle policy of MaxWeight $\pi^*$, which operates with the knowledge of the true parameters $\theta_k$'s, and that of our policy $\pi$. Given the queue lengths at each time $t$, we denote the oracle assignments as

$$\{S_{k,t}^*\}_{k \in [K]} = \operatorname*{argmax}_{\{S_k\}_{k \in [K]} \in \mathcal{M}(\mathcal{N}_t)} \sum_{k \in [K]} \sum_{n \in S_k} Q_n(t)\mu(n|S_k, \theta_k).$$

We show that this oracle policy achieves stability in Proposition 1. For simplicity, we use $k_{n,t}^*$ for $k_{n,t}^{\pi^*}$. Then, the cumulative regret under $\pi$ is defined as

$$\mathcal{R}^\pi(T) = \sum_{t \in [T]} \sum_{n \in [N]} \mathbb{E}\Big[ (\mu(n|S_{k_{n,t}^*,t}, \theta_{k_{n,t}^*}) - \mu(n|S_{k_{n,t},t}, \theta_{k_{n,t}}))Q_n(t) \Big]. \tag{9}$$

We define $Q_{\max} = \mathbb{E}[\max_{t \in [T], n \in [N]} Q_n(t)]$. Then the algorithm achieves the following regret bound.

**Theorem 2.** *The policy $\pi$ of Algorithm 1 achieves a regret bound of*

$$\mathcal{R}^\pi(T) = \widetilde{\mathcal{O}}\left( \min\left\{ \frac{d}{\kappa}\sqrt{KT}Q_{\max}, \left( \frac{dNK\min\{N,K\}^3}{\kappa^2 \epsilon^3} \right)^{1/4} T^{3/4} \right\} \right).$$

We emphasize that our algorithms achieve a sublinear regret bound, even in the worst-case scenario regarding queue lengths from the minimum in regret. In contrast, [23] achieves a regret bound of $\widetilde{O}(\max\{\sqrt{T}Q_{\max}, T^{3/4}\})$ for a stationary setting, where the worst-case bound is not guaranteed to be sublinear from the maximum in regret.

*Proof sketch.* Here we provide a proof sketch and the full version is provided in Appendix A.5. We first provide the proof for regret bound of $\mathcal{R}^\pi(T) = \widetilde{\mathcal{O}}(\frac{d}{\kappa}\sqrt{KT}Q_{\max})$. We define event $E_t^1 = \{\|\hat{\theta}_{k,t} - \theta_k^*\|_{V_{k,t}} \leq \beta_t \ \forall k \in [K]\}$ which holds with a high probability. Therefore, here we only consider the case when $E_t$ holds. Then we can show that $\sum_{n\in[N]}\mathbb{E}[Q_n(t)(\mu(n|S_{k_{n,t}^*,t}, \theta_{k_{n,t}^*}) - \mu(n|S_{k_{n,t},t}, \theta_{k_{n,t}}))] \leq \sum_{n\in[N]}\mathbb{E}[Q_n(t)(\tilde{\mu}_t^{UCB}(n|S_{k_{n,t},t}) - \mu(n|S_{k_{n,t},t}, \theta_{k_{n,t}}))] \leq \sum_{k\in[K]}\mathbb{E}[2\beta_t \max_{n\in S_{k,t}}\|x_n\|_{V_{k,t}^{-1}}Q_n(t)]$. From the inequality, with $\sum_{t=1}^T \max_{n\in S_{k,t}}\|x_n\|_{V_{k,t}^{-1}}^2 \leq (4d/\kappa)\log(1 + (TL/d\lambda))$, we have

$$\mathcal{R}^\pi(T) = \sum_{t\in[T]}\sum_{n\in[N]}\mathbb{E}[Q_n(t)(\mu(n|S_{k_{n,t}^*,t}, \theta_{k_{n,t}^*}) - \mu(n|S_{k_{n,t},t}, \theta_{k_{n,t}}))]$$

$$\leq 2\mathbb{E}\Big[\max_{t\in[T],n\in[N]}Q_n(t)\beta_T\sqrt{KT\sum_{t\in[T]}\sum_{k\in[K]}\max_{l\in S_{k,t}}\|x_l\|_{V_{k,t}^{-1}}^2}\Big] = \widetilde{\mathcal{O}}\Big(\frac{d}{\kappa}\sqrt{KT}Q_{\max}\Big). \quad (10)$$

Now we provide the proof for the worst-case regret bound of $\mathcal{R}^\pi(T) = \widetilde{\mathcal{O}}\left(\left(\frac{dNK\min\{N,K\}^3}{\kappa^2\epsilon^3}\right)^{1/4}T^{3/4}\right)$ in the following. We additionally define event $E_{n,t}^2 = \{\max_{m\in S_{k,t}}\|x_m\|_{V_{k,t}^{-1}} \leq \zeta \text{ for } k = k_{n,t}\}$ for some constant $C_2 > 0$. Under $E_t^1$, we have

$$\mathcal{R}^\pi(T) = \sum_{t\in[T]}\sum_{n\in[N]}\mathbb{E}[(\mu(n|S_{k_{n,t}^*,t}, \theta_{k_{n,t}^*}) - \mu(n|S_{k_{n,t},t}, \theta_{k_{n,t}}))Q_n(t)]$$

$$\leq \sum_{t\in[T]}\sum_{n\in[N]}\mathbb{E}[(\mu(n|S_{k_{n,t}^*,t}, \theta_{k_{n,t}^*}) - \mu(n|S_{k_{n,t},t}, \theta_{k_{n,t}}))Q_n(t)(\mathbb{1}(E_{n,t}^2) + \mathbb{1}((E_{n,t}^2)^c))].$$

$$(11)$$

Then for the first term of Eq.(11), we have

$$\sum_{t\in[T]}\mathbb{E}\left[\sum_{n\in[N]}(\mu(n|S_{k_{n,t}^*,t}, \theta_{k_{n,t}^*}) - \mu(n|S_{k_{n,t},t}, \theta_{k_{n,t}}))Q_n(t)\mathbb{1}(E_{n,t}^2)\right]$$

$$\leq \sum_{t\in[T]}\mathbb{E}\left[\sum_{n\in[N]}2\beta_t\|x_n\|_{V_{k_{n,t},t}^{-1}}Q_n(t)\mathbb{1}(E_{n,t}^2)\right] \leq \sum_{t\in[T]}\sum_{n\in[N]}2\beta_t\zeta\mathbb{E}[Q_n(t)], \quad (12)$$

where the last inequality is obtained from $E_{n,t}^2$. By analyzing the selected number of agent $n$ with $(E_{n,t}^2)^c$, we can show that

$$\sum_{t\in[T]}\sum_{n\in[N]}\mathbb{E}[(\mu(n|S_{k_{n,t}^*,t}, \theta_{k_{n,t}^*}) - \mu(n|S_{k_{n,t},t}, \theta_{k_{n,t}}))Q_n(t)\mathbb{1}((E_{n,t}^2)^c)]$$

$$\leq \sum_{t\in[T]}\sum_{n\in[N]}\zeta\beta_T\mathbb{E}[Q_n(t)] + \mathcal{O}\left(\frac{NK}{\kappa\zeta^3\beta_T}\right). \quad (13)$$

By putting the results of Eqs. (11), (12), (13), and Theorem 1, by setting $\zeta = (\epsilon NK/\min\{N,K\}T\kappa\beta_T^2)^{1/4}$, for large enough $T$, we have

$$\mathcal{R}^\pi(T) = \mathcal{O}\left(\zeta\beta_T\sum_{t\in[T]}\sum_{n\in[N]}\mathbb{E}[Q_n(t)] + \frac{NK}{\kappa\zeta^3\beta_T} + N\log(T)\right)$$

$$= \widetilde{\mathcal{O}}\left(N\left(\frac{dNK\min\{N,K\}^3}{\kappa^2\epsilon^3}\right)^{1/4}T^{3/4}\right), \quad (14)$$

which conclude the proof combined with Eq.(10). □

---
**Algorithm 2** Thompson Sampling-Queueing Matching Bandit (TS-QMB)
---
**Input:** $\lambda$, $M$, $\kappa$, $C_1 > 0$
**for** $t = 1, \ldots, T$ **do**
  **for** $k \in [K]$ **do**
    $\hat{\theta}_{k,t} \leftarrow \arg\min_{\theta \in \Theta} g_{k,t-1}(\hat{\theta}_{k,t-1})^\top \theta + \frac{1}{2}\|\theta - \hat{\theta}_{k,t-1}\|^2_{V_{k,t}}$ with (1)
    Sample $\{\tilde{\theta}_{k,t}^{(i)}\}_{i \in [M]}$ independently from $\mathcal{N}(\hat{\theta}_{k,t}, \beta_t^2 V_{k,t}^{-1})$
  $\{S_{k,t}\}_{k \in [K]} \leftarrow \underset{\{S_k\}_{k \in [K]} \in \mathcal{M}(\mathcal{N}_t)}{\arg\max} \sum_{k \in [K]} \sum_{n \in S_k} Q_n(t) \tilde{\mu}_t^{TS}(n|S_k, \{\tilde{\theta}_{k,t}^{(i)}\}_{i \in [M]})$ with (15)
  Offer $\{S_{k,t}\}_{k \in [K]}$ and observe preference feedback $y_{n,t} \in \{0, 1\}$ for all $n \in S_{k,t}$, $k \in [K]$
---

## 5.2 Thompson Sampling-based Algorithm

Here, we propose an algorithm based on Thompson Sampling, TS-QMB (Algorithm 2). As in the previous algorithm, we construct the estimator as $\hat{\theta}_{k,t} = \arg\min_{\theta \in \Theta} g_{k,t-1}(\hat{\theta}_{k,t-1})^\top \theta + \frac{1}{2}\|\theta - \hat{\theta}_{k,t-1}\|^2_{V_{k,t}}$. To facilitate exploration, we sample several $\tilde{\theta}_{k,t}^{(i)}$ for $i \in [M]$ from a Gaussian distribution of $\mathcal{N}(\hat{\theta}_{k,t}, \beta_t^2 V_{k,t}^{-1})$ and construct the Thompson Sampling (TS) index for assortment $S_k$ as

$$\tilde{\mu}_t^{TS}(n|S_k, \{\tilde{\theta}_{k,t}^{(i)}\}_{i \in [M]}) = \frac{\exp(h_{n,k,t}^{TS})}{1 + \sum_{m \in S_k} \exp(h_{m,k,t}^{TS})}, \tag{15}$$

where $h_{n,k,t}^{TS} = \max_{i \in [M]} x_n^\top \tilde{\theta}_{k,t}^{(i)}$ and $\beta_t = C_1 \sqrt{\lambda + \frac{d}{\kappa}\log(1 + \frac{tLK}{d\lambda})}$ for some constant $C_1 > 0$. Then we utilize the MaxWeight with TS indexes. We set $\lambda = 1$ and $M = \lceil 1 - \frac{\log(KL)}{\log(1 - 1/4\sqrt{e\pi})} \rceil$.

### 5.2.1 Stability Analysis of TS-QMB

Here we provide stability analysis for TS-QMB (Algorithm 2).

**Theorem 3.** *The average queue length of Algorithm 2 is bounded as* $\mathcal{Q}(T) = \mathcal{O}\left(\frac{\min\{N,K\}}{\epsilon} + \frac{d^4 N^2 K^2}{\kappa^4 \epsilon^6} \frac{\text{polylog}(T)}{T}\right)$, *which implies that the algorithm achieves stability as*

$$\lim_{T \to \infty} \mathcal{Q}(T) = \mathcal{O}\left(\frac{\min\{N, K\}}{\epsilon}\right).$$

*Proof.* The proof is provided in Appendix A.6 □

### 5.2.2 Regret Analysis of TS-QMB

We provide a regret analysis of TS-QMB (Algorithm 2) for the regret definition of (9) in the following.

**Theorem 4.** *The policy $\pi$ of Algorithm 2 achieves a regret bound of*

$$\mathcal{R}^\pi(T) = \tilde{\mathcal{O}}\left(\min\left\{\frac{d^{3/2}}{\kappa}\sqrt{KT}Q_{\max}, \left(\frac{d^2 NK \min\{N,K\}^3}{\kappa^2 \epsilon^3}\right)^{1/4} T^{3/4}\right\}\right).$$

*Proof.* The proof is provided in Appendix A.7 □

We note that the performance of Algorithm 1 and Algorithm 2 in the analysis results shows similar trends. However, the TS-based Algorithm 2 incurs a loss with respect to $d$ compared to the UCB-based Algorithm 1, as commonly seen in previous TS-based algorithms [5, 2, 43].

Here, we briefly discuss the combinatorial optimization of $\arg\max_{\{S_k\}_{k \in [K]} \in \mathcal{M}(\mathcal{N}_t)} \sum_{k \in [K]} f_k(S_k)$ for some function $f_k : S \subset [N] \to \mathbb{R}$ in our algorithms. The exact optimization can be expensive due to its NP-hard nature. To address this, we can utilize the technique of $\alpha$-approximation oracle with $0 \leq \alpha \leq 1$, first introduced in Kakade et al. [25], which is deferred to Appendix A.9.

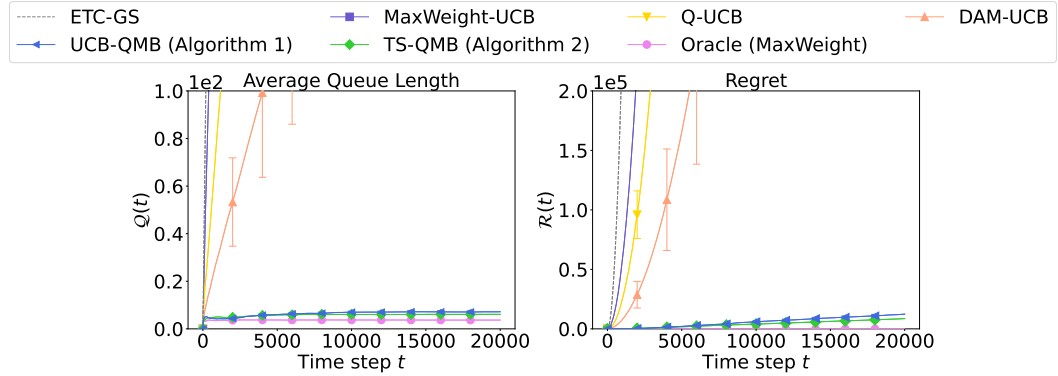

Figure 2: Experimental results for (left) average queue length and (right) regret

## 6 Experiments

Here, we provide experimental results to demonstrate the performance of our algorithms.[3] For the synthetic experiments, we consider $N = 4$, $K = 2$, $L = 2$, and $d = 2$. Each element in $x_n$ and $\theta_k$ is uniformly generated from $[0, 1]$ and then normalized, and $\lambda_n$'s are determined with $\epsilon = 0.1$. Even though no dedicated benchmark exists for our queueing matching scenario, we compare our algorithms with previously suggested ones for queueing bandits or matching bandits: `Q-UCB` [32], `DAM-UCB` [15], and `MaxWeight-UCB` [52] for multi-queue multi-server bandits with asymmetric service rates and `ETC-GS` [34] for matching bandits. In Figure 2, we can observe that our algorithms (Algorithms 1 and 2) outperform the previously suggested ones except for the oracle (MaxWeight) operated under known (latent) service rates. We demonstrate that our algorithms achieve stability, similar to the oracle in the left figure, which matches the results of our stability analysis (Theorems 1 and 3). Regarding regret shown in the right figure, the previously suggested algorithms exhibit superlinear performance due to the increasing $Q_n(t)$, while our algorithms show relatively small regret (Theorems 2 and 4). Additional experiments can be found in Appendix A.10.

## 7 Conclusion

In this paper, we introduce a novel framework for queueing matching bandits with preference feedback. To achieve stability in this framework, we propose UCB and TS-based algorithms utilizing the MaxWeight strategy. The algorithms achieve system stability with an average queue length bound of $\mathcal{O}(\min\{N, K\}/\epsilon)$. Furthermore, the algorithms achieve sublinear regret bounds of $\widetilde{\mathcal{O}}(\min\{\sqrt{T}Q_{\max}, T^{3/4}\})$. Lastly, we demonstrate our algorithms using synthetic datasets.

## 8 Acknowledgements

The authors thank Milan Vojnović for helpful discussions and Hyunjun Choi for providing useful code. JK was supported by the Global-LAMP Program of the National Research Foundation of Korea (NRF) grant funded by the Ministry of Education (No. RS-2023-00301976). MO was supported by the NRF grant funded by the Korea government(MSIT) (No. 2022R1C1C1006859 and 2022R1A4A1030579) and by AI-Bio Research Grant through Seoul National University.

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

# A   Appendix

## A.1   Limitations & Discussion

We leave several questions open for future research. Firstly, it remains an open problem to establish lower bounds for queue lengths and regret. However, we believe that constructing a lower bound would be much more challenging compared to other parametric bandit problems. Additionally, improving the dependency on $\kappa$ for queue length and regret bounds from the structure of MNL would be an interesting avenue for future work.

Regarding computation efficiency, we note that updating estimators in our algorithms is computationally efficient because they are based on online updates with convex optimization for updating estimators. Concerning combinatorial optimization in our algorithms, we can alleviate computational costs using $\alpha$-approximation oracle algorithms, as discussed in Appendix A.9. Note that almost all combinatorial bandit problems, including our proposed matching bandit framework, involve a combinatorial optimization step which often relies on some type of approximation optimization oracles. Hence, this is not a particular limitation specific to our study. However, developing an improved approximation oracle would also be an interesting direction for future research.

## A.2   Discussion on Traffic Slackness Parameter $\epsilon$

As in the closely related works of [15, 52, 23], the traffic slackness remains constant regardless of the number of agents $N$ and the number of servers $K$ as $\epsilon = \epsilon_0$ for some $0 < \epsilon_0 < 1$. However, this may not align with intuition, as a large $K$ might be beneficial in terms of traffic slackness, while a large $N$ could have the opposite effect. To address this, we can consider $\epsilon = \epsilon_0 \frac{\min(N,K)}{N}$. This reflects that when $N \geq K$, due to the lack of servers, increasing $K$ is critical for increasing traffic slackness, while increasing $N$ could decrease it. When $N < K$ implying there are enough servers, however, the value of $N$ doesn't impact the traffic slackness because each agent can be assigned to at most one server at each time, and there are sufficient servers to handle the agents.

If we consider the traffic slackness as $\epsilon = \epsilon_0 \frac{\min\{N,K\}}{N}$, the oracle strategy of MaxWeight achieves $\mathcal{Q}(T) = \mathcal{O}(\frac{N}{\epsilon_0})$ from Proposition 1. This result shows that as $N$ increases, causing the traffic slackness to decrease, the average queue length increases. Meanwhile, the positive influence of $K$ on traffic slackness is neutralized in the average queue length bound due to system variance.

## A.3   Proof of Proposition 1

Define the set of queues $\mathbf{Q}(t) = [Q_n(t) : n \in [N]]$ and a Lyapunov function as $\mathcal{V}(\mathbf{Q}(t)) = \sum_{n \in [N]} Q_n(t)^2$. For simplicity, we use $D_n(t)$ for $D_n(t|S_{k_{n,t},t})$ when there is no confusion. We observe that $\sum_{n \in [N]} \mathbb{E}[A_n(t)] \leq \sum_{n \in [N]} \lambda_n \leq \sum_{n \in [N]} \mu(n|S_k, \theta_k) \leq K$ for some $S_k$ from Assumption 3 and $\sum_{n \in [N]} \lambda_n \leq N$. This implies $\sum_{n \in [N]} [A_n(t)] \leq \min\{N, K\}$. We also have $\sum_{n \in [N]} \mathbb{E}[D_n(t)] = \sum_{n \in [N]} \mathbb{E}[\mu(n|S_{k,t}, \theta_k)] \leq \min\{K, N\}$. Then we have

$$\mathbb{E}[\mathcal{V}(\mathbf{Q}(t+1)) - \mathcal{V}(\mathbf{Q}(t))]$$

$$= \mathbb{E}\left[ \sum_{n \in [N]} (Q_n(t) - A_n(t) + D_n(t))^{+2} - Q_n(t)^2 \right]$$

$$\leq \mathbb{E}\left[ \sum_{n \in [N]} (Q_n(t) - A_n(t) + D_n(t))^2 - Q_n(t)^2 \right]$$

$$\leq \mathbb{E}\left[ 2\sum_{n \in [N]} Q_n(t)A_n(t) - 2\sum_{n \in [N]} Q_n(t)D_n(t) - 2\sum_{n \in [N]} A_n(t)D_n(t) + \sum_{n \in [N]} A_n(t)^2 + \sum_{n \in [N]} D_n(t)^2 \right]$$

$$\leq \mathbb{E}\left[ 2\sum_{n \in [N]} Q_n(t)A_n(t) - 2\sum_{n \in [N]} Q_n(t)D_n(t) + \sum_{n \in [N]} A_n(t) + \sum_{n \in [N]} D_n(t) \right]$$

$$\leq \mathbb{E}\left[ 2\sum_{n \in [N]} Q_n(t)A_n(t) - 2\sum_{n \in [N]} Q_n(t)D_n(t) + 2\min\{N, K\} \right] \qquad (16)$$

From Assumption 3, we define the corresponding assortments as $\{S'_k\}_{k\in[K]} \in \mathcal{M}(\mathcal{N})$, which satisfies $\lambda_n + \epsilon \leq \mu(n|S'_k, \theta_k)$ for all $n \in S'_k$ and $k \in [K]$. Then we define the set of non-empty queues in $S'_k$ at time $t$ as $S'_{k,t} = \{n \in S'_k : Q_n(t) \neq 0\}$. From the property of the MNL function, we can observe that $\mu(n|S'_k, \theta_k) \leq \mu(n|S'_{k,t}, \theta_k)$ for all $n \in S'_{k,t}$. We also note that $\{S'_{k,t}\}_{k\in[K]} \in \mathcal{M}(\mathcal{N}_t)$. Then we have

$$\sum_{n\in[N]} (\lambda_n + \epsilon)Q_n(t) \leq \sum_{k\in[K]} \sum_{n\in S'_k} \mu(n|S'_k, \theta_k)Q_n(t)$$

$$\leq \sum_{k\in[K]} \sum_{n\in S'_{k,t}} \mu(n|S'_{k,t}, \theta_k)Q_n(t)$$

$$\leq \sum_{k\in[K]} \sum_{n\in S_{k,t}} \mu(n|S_{k,t}, \theta_k)Q_n(t),$$

where the second inequality is obtained from $\mu(n|S'_k, \theta_k)Q_n(t) \leq \mu(n|S'_{k,t}, \theta_k)Q_n(t)$ when $n \in \mathcal{N}_t$, and otherwise $\mu(n|S'_k, \theta_k)Q_n(t) = \mu(n|S'_{k,t}, \theta_k)Q_n(t) = 0$.

This implies

$$\mathbb{E}\left[\sum_{k\in[K]} \sum_{n\in S_{k,t}} Q_n(t)D_n(t)\right] = \mathbb{E}\left[\mathbb{E}\left[\sum_{k\in[K]} \sum_{n\in S_{k,t}} Q_n(t)D_n(t)\bigg|\mathbf{Q}(t)\right]\right]$$

$$= \mathbb{E}\left[\sum_{k\in[K]} \sum_{n\in S_{k,t}} Q_n(t)\mu(n|S_{k,t}, \theta_k)\right]$$

$$\geq \mathbb{E}\left[\sum_{k\in[K]} \sum_{n\in S'_{k,t}} Q_n(t)\mu(n|S'_{k,t}, \theta_k)\right]$$

$$\geq \sum_{n\in[N]} (\lambda_n + \epsilon)\mathbb{E}[Q_n(t)]. \tag{17}$$

From Eqs.(16) and (17), we have

$$\mathbb{E}[\mathcal{V}(\mathbf{Q}(t+1)) - \mathcal{V}(\mathbf{Q}(t))]$$

$$\leq \mathbb{E}\left[2\sum_{n\in[N]} Q_n(t)A_n(t) - 2\sum_{n\in[N]} Q_n(t)D_n(t)\right] + 2\min\{N, K\}$$

$$\leq \mathbb{E}\left[\left[2\sum_{n\in[N]} Q_n(t)A_n(t) - 2\sum_{n\in[N]} Q_n(t)D_n(t)\bigg|\mathbf{Q}(t)\right]\right] + 2\min\{N, K\}$$

$$\leq \mathbb{E}\left[2\sum_{n\in[N]} \lambda_n Q_n(t) - 2(\lambda_n + \epsilon)Q_n(t)\right] + 2\min\{N, K\}$$

$$\leq -\mathbb{E}\left[2\epsilon\sum_{n\in[N]} Q_n(t)\right] + 2\min\{N, K\},$$

which implies from $\mathcal{V}(\mathbf{Q}(T+1)) \geq 0$ and $\mathcal{V}(\mathbf{Q}(1)) = 0$,

$$\sum_{t\in[T]} \mathbb{E}[\mathcal{V}(\mathbf{Q}(t+1) - \mathcal{V}(\mathbf{Q}(t))] = \mathcal{V}(\mathbf{Q}(T+1))$$

$$\leq -\sum_{t\in[T]} \mathbb{E}[2\epsilon\sum_{n\in[N]} Q_n(t)] + 2\min\{N, K\}T.$$

Finally, we can conclude that $\mathcal{Q}(T) = (1/T)\sum_{t\in[T]} \mathbb{E}[\sum_{n\in[N]} Q_n(t)] \leq 2\min\{N, K\}/\epsilon$.

## A.4 Proof of Theorem 1

We first define the set of queues $\mathbf{Q}(t) = [Q_n(t) : n \in [N]]$ and a Lyapunov function as $\mathcal{V}(\mathbf{Q}(t)) = \sum_{n \in [N]} Q_n(t)^2$. For simplicity, we use $D_n(t)$ for $D_n(t|S_{k_{n,t},t})$ and $D_n^*(t)$ for $D_n(t|S_{k_{n,t}^*,t})$ when there is no confusion. Then we analyze the Lyapunov drift as follows.

$$\sum_{t \in [T]} \mathcal{V}(\mathbf{Q}(t+1)) - \mathcal{V}(\mathbf{Q}(t))$$

$$= \sum_{t \in [T]} \sum_{n \in [N]} (Q_n(t) + A_n(t) - D_n(t))^{+2} - Q_n(t)^2$$

$$= \sum_{t \in [T]} \sum_{n \in [N]} (Q_n(t) + A_n(t) - D_n^*(t))^2 - Q_n(t)^2$$

$$+ \sum_{t \in [T]} \sum_{n \in [N]} (Q_n(t) + A_n(t) - D_n(t))^{+2} - \sum_{t \in [T]} \sum_{n \in [N]} (Q_n(t) + A_n(t) - D_n^*(t))^2. \quad (18)$$

We observe that $\sum_{n \in [N]} \mathbb{E}[A_n(t)] \le \sum_{n \in [N]} \lambda_n \le \sum_{n \in [N]} \mu(n|S_k, \theta_k) \le K$ for some $S_k$ from Assumption 3 and $\sum_{n \in [N]} \lambda_n \le N$. This implies $\sum_{n \in [N]}[A_n(t)] \le \min\{N, K\}$. We also have $\sum_{n \in [N]} \mathbb{E}[D_n(t)] = \sum_{n \in [N]} \mathbb{E}[\mu(n|S_{k,t}, \theta_k)] \le \min\{K, N\}$. For the first two terms in Eq.(18), by following the same procedure of Eqs.(16) and (17), we can obtain

$$\sum_{t \in [T]} \sum_{n \in [N]} \mathbb{E}[(Q_n(t) + A_n(t) - D_n^*(t))^2 - Q_n(t)^2]$$

$$\le \sum_{t \in [T]} \sum_{n \in [N]} 2\mathbb{E}[(\lambda_n - \mu(n|S_{k,t}^*, \theta_k))Q_n(t)] + 2\min\{N, K\}T$$

$$\le -\sum_{t \in [T]} \sum_{n \in [N]} 2\epsilon \mathbb{E}[Q_n(t)] + 2\min\{N, K\}T, \quad (19)$$

where the last inequality is obtained using Assumption 3. For the last two terms in Eq.(18), we have

$$\sum_{t \in [T]} \sum_{n \in [N]} (Q_n(t) + A_n(t) - D_n(t))^{+2} - \sum_{t \in [T]} \sum_{n \in [N]} (Q_n(t) + A_n(t) - D_n^*(t))^2$$

$$\le \sum_{t \in [T]} \sum_{n \in [N]} (D_n^*(t) - D_n(t))(2Q_n(t) + 2A_n(t) - D_n^*(t) - D_n(t))$$

$$\le 2\sum_{t \in [T]} \sum_{n \in [N]} A_n(t)(D_n^*(t) - D_n(t)) - \sum_{t \in [T]} \sum_{n \in [N]} D_n^*(t)^2 + \sum_{t \in [T]} \sum_{n \in [N]} D_n(t)^2$$

$$+ 2\sum_{t \in [T]} \sum_{n \in [N]} (D_n^*(t) - D_n(t))Q_n(t)$$

$$\le 4\sum_{t \in [T]} \sum_{n \in [N]} A_n(t) + 2\sum_{t \in [T]} \sum_{n \in [N]} (D_n^*(t) - D_n(t))Q_n(t) + \min\{N, K\}T$$

$$\le 2\sum_{t \in [T]} \sum_{n \in [N]} (D_n^*(t) - D_n(t))Q_n(t) + 5\min\{N, K\}T. \quad (20)$$

Now we provide a bound for Eq.(20). We first provide some lemmas for the concentration of estimators.

Using the above lemma, we can show the following lemma of the concentration.

**Lemma 1** (Lemma 9 in Oh and Iyengar [44]). *For $t \ge 1$ and some constant $C_1 > 0$, with probability at least $1 - 1/t^2$, for all $k \in [K]$ we have*

$$\|\hat{\theta}_{k,t} - \theta_k\|_{V_{k,t}} \le \beta_t.$$

*Proof.* For the completeness, we provide the proof in Appendix A.8. $\quad\square$

Define event $E_t^1 = \{\|\hat{\theta}_{k,t} - \theta_k^*\|_{V_{k,t}} \leq \beta_t$ for all $k \in [K]\}$ where $\beta_t = C_1\sqrt{\lambda + \frac{d}{\kappa}\log(1 + tLK/d\lambda)}$, which holds with high probability as $\mathbb{P}(E_t^1) \geq 1 - 1/t^2$ from the above lemma. We also define $E_{n,t}^2 = \{\max_{m \in S_{k,t}} \|x_m\|_{V_{k,t}^{-1}} \leq C_2\epsilon/2\beta_t$ for $k = k_{n,t}\}$ for some constant $C_2 > 0$. Then, we have

$$\sum_{t \in [T]} \sum_{n \in [N]} \mathbb{E}[(D_n^*(t) - D_n(t)Q_n(t)]$$

$$= \sum_{t \in [T]} \sum_{n \in [N]} \mathbb{E}[\mathbb{E}[(D_n^*(t) - D_n(t)Q_n(t)|Q_n(t)]]$$

$$= \mathbb{E}[\sum_{t \in [T]} \sum_{n \in [N]} \mu(n|S_{k_{n,t}^*,t}, \theta_{k_{n,t}^*}) - \mu(n|S_{k_{n,t},t}, \theta_{k_{n,t}}))Q_n(t)]$$

$$\leq \sum_{t \in [T]} \sum_{n \in [N]} \mathbb{E}[(\mu(n|S_{k_{n,t}^*,t}, \theta_{k_{n,t}^*}) - \mu(n|S_{k_{n,t},t}, \theta_{k_{n,t}}))Q_n(t)\mathbb{1}(E_t^1 \cap E_{n,t}^2)]$$

$$+ \sum_{t \in [T]} \sum_{n \in [N]} \mathbb{E}[(\mu(n|S_{k_{n,t}^*,t}, \theta_{k_{n,t}^*}) - \mu(n|S_{k_{n,t},t}, \theta_{k_{n,t}}))Q_n(t)(\mathbb{1}((E_t^1)^c) + \mathbb{1}((E_{n,t}^2)^c))].$$

$$(21)$$

Now we provide a lemma for bounding the first term in Eq.(21).

**Lemma 2.** *Under $E_t^1$, for any $n \in [N]$, we have $\widetilde{\mu}_t^{UCB}(n|S_{k_{n,t},t}, \hat{\theta}_{k_{n,t},t}) - \mu(n|S_{k_{n,t},t}, \theta_{k_{n,t}}) \leq 2\beta_t\|x_n\|_{V_{k_{n,t},t}^{-1}}$.*

*Proof.* Let $u_{n,k,t} = x_n^\top \theta_k$. Under $E_t^1$, for any $n \in [N]$ and $k \in [K]$ we have $x_n^\top \hat{\theta}_{k,t} - \beta_t\|x_n\|_{V_{k,t}^{-1}} \leq x_n^\top \theta_k \leq x_n^\top \hat{\theta}_{k,t} + \beta_t\|x_n\|_{V_{k,t}^{-1}}$, which implies $0 \leq h_{n,k,t}^{UCB} - u_{n,k,t} \leq 2\beta_t\|x_n\|_{V_{k,t}^{-1}}$. Then by the mean value theorem, there exists $\bar{u}_{n,k,t} = (1 - c)h_{n,k,t}^{UCB} + cu_{n,k,t}$ for some $c \in (0, 1)$ satisfying, for any $n \in S_k$, $S_k \subset [N]$, and $k \in [K]$,

$$\widetilde{\mu}_t^{UCB}(n|S_k, \hat{\theta}_{k,t}) - \mu(n|S_k, \theta_k) = \frac{\exp(h_{n,k,t}^{UCB})}{1 + \sum_{m \in S_k} \exp(h_{m,k,t}^{UCB})} - \frac{\exp(u_{n,k,t})}{1 + \sum_{m \in S_k} \exp(u_{m,k,t})}$$

$$= \nabla_{v_n}\left(\frac{\exp(v_n)}{1 + \sum_{m \in S_k} \exp(v_m)}\right)\Big|_{v_n = \bar{u}_{n,k,t}} (h_{n,k,t}^{UCB} - u_{n,k,t})$$

$$\leq \frac{\exp(\bar{u}_{n,k,t})(h_{n,k,t}^{UCB} - u_{n,k,t})}{1 + \sum_{n \in S_k} \exp(\bar{u}_{n,k,t})}$$

$$\leq h_{n,k,t}^{UCB} - u_{n,k,t}$$

$$\leq 2\beta_t\|x_n\|_{V_{k,t}^{-1}}.$$

$\square$

Since $x/(1 + x)$ is a non-decreasing function for $x > -1$ and $x_n^\top \theta_{k_{n,t}^*} \leq x_n^\top \hat{\theta}_{k_{n,t}^*,t} + \beta_t\|x_n\|_{V_{k_{n,t}^*,t}^{-1}}$ under $E_t^1$, we have $\mu(n|S_{k_{n,t}^*,t}, \theta_{k_{n,t}^*}) \leq \widetilde{\mu}_t^{UCB}(n|S_{k_{n,t}^*,t}, \hat{\theta}_{k_{n,t}^*,t})$. Then for the first term of Eq.(21),

we have

$$\sum_{t\in[T]}\mathbb{E}\left[\sum_{n\in[N]}(\mu(n|S_{k^*_{n,t},t},\theta_{k^*_{n,t}})-\mu(n|S_{k_{n,t},t},\theta_{k_{n,t}}))Q_n(t)\mathbb{1}(E^1_t\cap E^2_{n,t})\right]$$

$$\leq\sum_{t\in[T]}\mathbb{E}\left[\sum_{n\in[N]}(\widetilde{\mu}^{UCB}_t(n|S_{k^*_{n,t}},\hat{\theta}_{k^*_{n,t},t})-\mu(n|S_{k_{n,t},t},\theta_{k_{n,t}}))Q_n(t)\mathbb{1}(E^1_t\cap E^2_{n,t})\right]$$

$$\leq\sum_{t\in[T]}\mathbb{E}\left[\sum_{n\in[N]}(\widetilde{\mu}^{UCB}_t(n|S_{k_{n,t}},\hat{\theta}_{k_{n,t},t})-\mu(n|S_{k_{n,t},t},\theta_{k_{n,t}}))Q_n(t)\mathbb{1}(E^1_t\cap E^2_{n,t})\right]$$

$$\leq\sum_{t\in[T]}\mathbb{E}\left[\sum_{n\in[N]}2\beta_t\|x_n\|_{V^{-1}_{k_{n,t},t}}Q_n(t)\mathbb{1}(E^1_t\cap E^2_{n,t})\right]$$

$$\leq C_2\epsilon\sum_{t\in[T]}\sum_{n\in[N]}\mathbb{E}\left[Q_n(t)\right], \tag{22}$$

where the second inequality comes from the UCB strategy of the algorithm, the last second inequality is obtained from Lemma 2, and the last inequality is obtained from $E^2_{n,t}$.

Now we provide a bound for the second term of Eq.(21). We first have

$$\sum_{t\in[T]}\sum_{n\in[N]}\mathbb{E}\left[(\mu(n|S_{k^*_{n,t},t},\theta_{k^*_{n,t}})-\mu(n|S_{k_{n,t},t},\theta_{k_{n,t}}))Q_n(t)\mathbb{1}((E^1_t)^c)\right]$$

$$\leq\sum_{t\in[T]}\sum_{n\in[N]}\mathbb{E}\left[Q_n(t)\mathbb{1}((E^1_t)^c)\right]$$

$$\leq\sum_{t\in[T]}\sum_{n\in[N]}t\mathbb{P}((E^1_t)^c)$$

$$=\mathcal{O}\left(\sum_{t\in[T]}\sum_{n\in[N]}t(1/t^2)\right)=\mathcal{O}(N\log(T)), \tag{23}$$

where the second inequality is obtained from $Q_n(t)\leq t$.

Here we utilize some techniques introduced in Freund et al. [15]. Let $\mathcal{T}_n$ be the set of time steps $t\in[T]$ such that $Q_n(t)\neq 0$ and let $e_T=\sum_{n\in[N]}\sum_{t\in\mathcal{T}_n}\mathbb{1}((E^2_{n,t})^c)$ and $h=\lceil C_3 e_T/\epsilon\rceil$ for some constant $C_3>0$. Then if $t\leq h$, we have $Q_n(t)\leq t\leq h$. Otherwise, we have

$$Q_n(t)\leq\sum_{s=t-h+1}^t(1/h)(Q_n(s)+(t-s))\leq(1/h)\sum_{s=1}^t Q_n(s)+(1/h)h^2=(1/h)\sum_{s=1}^t Q_n(s)+h.$$

Then, from the above inequality, we have

$$\sum_{t \in [T]} \sum_{n \in [N]} \mathbb{E}[(\mu(n|S_{k_{n,t}^*,t}, \theta_{k_{n,t}^*}) - \mu(n|S_{k_{n,t},t}, \theta_{k_{n,t}}))Q_n(t)\mathbb{1}((E_{n,t}^2)^c)]$$

$$= \sum_{n \in [N]} \sum_{t \in \mathcal{T}_n} \mathbb{E}[(\mu(n|S_{k_{n,t}^*,t}, \theta_{k_{n,t}^*}) - \mu(n|S_{k_{n,t},t}, \theta_{k_{n,t}}))Q_n(t)\mathbb{1}((E_{n,t}^2)^c)]$$

$$\le \sum_{n \in [N]} \sum_{t \in \mathcal{T}_n} \mathbb{E}[Q_n(t)\mathbb{1}((E_{n,t}^2)^c)]$$

$$\le \sum_{n \in [N]} \sum_{t \in \mathcal{T}_n} \mathbb{E}[((\epsilon/C_3 e_T) \sum_{s=1}^{T} Q_n(s) + 2C_3 e_T/\epsilon)\mathbb{1}((E_{n,t}^2)^c)]$$

$$\le \mathbb{E}\left[(\epsilon/C_3) \sum_{n \in [N]} \sum_{s=1}^{T} Q_n(s)\right] + \mathbb{E}\left[2C_3 e_T^2/\epsilon\right]$$

$$= \sum_{t \in [T]} \sum_{n \in [N]} (\epsilon/C_3)\mathbb{E}[Q_n(t)] + 2C_3\mathbb{E}[e_T^2]/\epsilon. \tag{24}$$

Now we provide a bound for $\mathbb{E}[e_T^2]$. Define $N_{n,k}(t) = \sum_{s=1}^{t-1} \mathbb{1}(n \in S_{k,s})$ and $\widetilde{V}_{k_{n,t},t} = \frac{\kappa}{2} \sum_{s=1}^{t-1} \sum_{n \in S_{k,s}} x_n x_n^\top$. Then, we have

$$e_T = \sum_{n \in [N]} \sum_{t \in \mathcal{T}_n} \mathbb{1}((E_{n,t}^2)^c)$$

$$\le \sum_{n \in [N]} \sum_{t \in \mathcal{T}_n} \mathbb{1}(\|x_n\|_{V_{k_{n,t},t}^{-1}} \ge C_2\epsilon/2\beta_t)$$

$$\le \sum_{n \in [N]} \sum_{t \in \mathcal{T}_n} \mathbb{1}(\|x_n\|_{\widetilde{V}_{k_{n,t},t}^{-1}} \ge C_2\epsilon/2\beta_t)$$

$$\le \sum_{n \in [N]} \sum_{t \in \mathcal{T}_n} \mathbb{1}(1/N_{n,k_{n,t}}(t) \ge (\kappa/2)(C_2\epsilon/2\beta_t)^2)$$

$$\le \sum_{n \in [N]} \sum_{t \in \mathcal{T}_n} \mathbb{1}(N_{n,k_{n,t}}(t) \le (2/\kappa)(2\beta_t/C_2\epsilon)^2)$$

$$\le \sum_{n \in [N]} \sum_{k \in [K]} \sum_{t \in \mathcal{T}_n} \mathbb{1}(N_{n,k}(t) \le (2/\kappa)(2\beta_T/C_2\epsilon)^2 \text{ and } k_{n,t} = k)$$

$$\le NK(2/\kappa)(2\beta_T/C_2\epsilon)^2.$$

From the above we have $\mathbb{E}[e_T^2] \le 64N^2K^2\beta_T^4/\kappa^2(C_2\epsilon)^4$. Then from Eq.(24), we have

$$\sum_{t \in [T]} \sum_{n \in [N]} \mathbb{E}[(\mu(n|S_{k_{n,t}^*,t}, \theta_{k_{n,t}^*}) - \mu(n|S_{k_{n,t},t}, \theta_{k_{n,t}}))Q_n(t)\mathbb{1}((E_{n,t}^2)^c)]$$

$$\le \sum_{t \in [T]} \sum_{n \in [N]} (\epsilon/C_3)\mathbb{E}[Q_n(t)] + \mathcal{O}(N^2K^2\beta_T^4/\kappa^2\epsilon^5). \tag{25}$$

By putting the results of Eqs. (18), (19), (20), (21), (22), (23), (25) altogether, we can obtain

$$\mathbb{E}\left[\sum_{t\in[T]}\mathcal{V}(\mathbf{Q}(t+1))-\mathcal{V}(\mathbf{Q}(t))\right]$$

$$\leq 7\min\{N,K\}T-2\epsilon\sum_{t\in[T]}\sum_{n\in[N]}\mathbb{E}\left[Q_n(t)\right]+2\sum_{t\in[T]}\sum_{n\in[N]}\mathbb{E}\left[(D_n(t|S_{k_{n,t}^*,t})-D_n(t|S_{k_{n,t},t}))Q_n(t)\right]$$

$$\leq 7\min\{N,K\}T-2\epsilon\sum_{t\in[T]}\sum_{n\in[N]}\mathbb{E}\left[Q_n(t)\right]+2C_2\epsilon\sum_{t\in[T]}\sum_{n\in[N]}\mathbb{E}\left[Q_n(t)\right]$$

$$+\mathcal{O}(N\log(T))+(2\epsilon/C_3)\sum_{t\in[T]}\sum_{n\in[N]}\mathbb{E}\left[Q_n(t)\right]+\mathcal{O}(N^2K^2\beta_T^4/\kappa^2\epsilon^5)$$

$$\leq 7\min\{N,K\}T+2(C_2+(1/C_3)-1)\epsilon\sum_{t\in[T]}\sum_{n\in[N]}\mathbb{E}\left[Q_n(t)\right]+\mathcal{O}(N\log(T))+\mathcal{O}(N^2K^2\beta_T^4/\kappa^2\epsilon^5).$$

Finally, with positive constants $C_2, C_3 > 0$ satisfying $C_2 + (1/C_3) < 1$, from $\mathcal{V}(\mathbf{Q}(1)) = 0$ and $\mathcal{V}(\mathbf{Q}(T+1)) \geq 0$, by using telescoping for the above inequality, we can conclude the proof by

$$\frac{1}{T}\sum_{t\in[T]}\sum_{n\in[N]}\mathbb{E}[Q_n(t)]=\mathcal{O}\left(\frac{\min\{N,K\}}{\epsilon}+\frac{1}{T}\frac{d^2N^2K^2}{\kappa^4\epsilon^6}\text{polylog}(T)\right). \tag{26}$$

## A.5  Proof of Theorem 2

We first provide the proof for regret bound of $\mathcal{R}^\pi(T) = \widetilde{\mathcal{O}}\left(\frac{d}{\kappa}\sqrt{KT}Q_{\max}\right)$.

We define event $E_t = \{\|\hat{\theta}_{k,t} - \theta_k^*\|_{V_{k,t}} \leq \beta_t \; \forall k \in [K]\}$ which holds at least probability of $1 - 1/t^2$ from Lemma 1.

**Lemma 3.** *Under $E_t^1$, for any $S_k \subset [N]$, we have*

$$\sum_{n\in S_k}(\widetilde{\mu}_t^{UCB}(n|S_k,\hat{\theta}_{k,t})-\mu(n|S_k,\theta_k))Q_n(t)\leq 2\beta_t\max_{n\in S_k}\|x_n\|_{V_{k,t}^{-1}}Q_n(t).$$

*Proof.* Let $u_{n,k,t} = x_n^\top\theta_k$. Under $E_t^1$, for any $n \in [N]$ and $k \in [K]$ we have $x_n^\top\hat{\theta}_{k,t} - \beta_t\|x_n\|_{V_{k,t}^{-1}} \leq x_n^\top\theta_k \leq x_n^\top\hat{\theta}_{k,t} + \beta_t\|x_n\|_{V_{k,t}^{-1}}$, which implies $0 \leq h_{n,k,t}^{UCB} - u_{n,k,t} \leq 2\beta_t\|x_n\|_{V_{k,t}^{-1}}$. Then by the mean value theorem, there exists $\bar{u}_{n,k,t} = (1-c)h_{n,k,t}^{UCB} + cu_{n,k,t}$ for some $c \in (0,1)$ satisfying, for any $S \subset [N]$,

$$\sum_{n\in S_k}(\widetilde{\mu}_t^{UCB}(n|S_k,\hat{\theta}_{k,t})-\mu(n|S_k,\theta_k))Q_n(t)$$

$$=\sum_{n\in S_k}\left(\frac{\exp(h_{n,k,t}^{UCB})}{1+\sum_{m\in S_k}\exp(h_{m,k,t}^{UCB})}-\frac{\exp(u_{n,k,t})}{1+\sum_{m\in S_k}\exp(u_{m,k,t})}\right)Q_n(t)$$

$$=\sum_{n\in S_k}\nabla_{v_n}\left(\frac{\exp(v_n)}{1+\sum_{m\in S_k}\exp(v_m)}\right)\Big|_{v_n=\bar{u}_{n,k,t}}(h_{n,k,t}^{UCB}-u_{n,k,t})Q_n(t)$$

$$=\frac{(1+\sum_{n\in S_k}\exp(\bar{u}_{n,k,t}))(\sum_{n\in S_k}\exp(\bar{u}_{n,k,t})(h_{n,k,t}^{UCB}-u_{n,k,t})Q_n(t))}{(1+\sum_{n\in S_k}\exp(\bar{u}_{n,k,t}))^2}$$

$$-\frac{(\sum_{n\in S_k}\exp(\bar{u}_{n,k,t}))(\sum_{n\in S_k}\exp(\bar{u}_{n,k,t})(h_{n,k,t}^{UCB}-u_{n,k,t})Q_n(t))}{(1+\sum_{n\in S_k}\exp(\bar{u}_{n,k,t}))^2}$$

$$\leq\sum_{n\in S_k}\frac{\exp(\bar{u}_{n,k,t})}{1+\sum_{m\in S_k}\exp(\bar{u}_{m,k,t})}(h_{n,k,t}^{UCB}-u_{n,k,t})Q_n(t)$$

$$\leq\max_{n\in S_k}(h_{n,k,t}^{UCB}-u_{n,k,t})Q_n(t)$$

$$\leq 2\beta_t\max_{n\in S_k}\|x_n\|_{V_{k,t}^{-1}}Q_n(t).$$

$$\square$$

Under $E_t$, from Lemma 3, we have

$$\sum_{n \in S_{k,t}} (\widetilde{\mu}_t^{UCB}(n|S_{k,t}, \hat{\theta}_{k,t}) - \mu(n|S_{k,t}, \theta_k))Q_n(t) \leq 2\beta_t \max_{n \in S_{k,t}} \|x_n\|_{V_{k,t}^{-1}} Q_n(t), \qquad (27)$$

and since $x/(1+x)$ is a non-decreasing function for $x > -1$ and $x_n^\top \theta_{k_{n,t}^*} \leq x_n^\top \hat{\theta}'_{k_{n,t}^*,t} + \beta_t \|x_n\|_{V_{k_{n,t}^*,t}^{-1}}$ under $E_t^1$, we have

$$\mu(n|S_{k_{n,t}^*,t}, \theta_{k_{n,t}^*}) \leq \widetilde{\mu}_t^{UCB}(n|S_{k_{n,t}^*,t}, \hat{\theta}_{k_{n,t}^*,t}). \qquad (28)$$

Now we provide an elliptical potential lemma.

**Lemma 4.** *For any $k \in [K]$, we have*

$$\sum_{t=1}^{T} \max_{n \in S_{k,t}} \|x_n\|_{V_{k,t}^{-1}}^2 \leq (4d/\kappa) \log(1 + (TL/d\lambda)).$$

*Proof.* First, we can show that

$$\det(V_{k,t})$$
$$= \det\left(V_{k,t-1} + (\kappa/2) \sum_{n \in S_{k,t-1}} x_n x_n^\top\right)$$
$$= \det(V_{k,t-1}) \det\left(I_d + (\kappa/2) \sum_{n \in S_{k,t-1}} V_{k,t-1}^{-1/2} x_n (V_{k,t-1}^{-1/2} x_n)^\top\right)$$
$$\geq \det(V_{k,t-1})\left(1 + (\kappa/2) \sum_{n \in S_{k,t-1}} \|x_n\|_{V_{k,t-1}^{-1}}^2\right)$$
$$\geq \det(\lambda I_d) \prod_{s=1}^{t-1}\left(1 + (\kappa/2) \sum_{n \in S_{k,s}} \|x_n\|_{V_{k,s}^{-1}}^2\right) = \lambda^d \prod_{s=1}^{t-1}\left(1 + (\kappa/2) \sum_{n \in S_{k,s}} \|x_n\|_{V_{k,s}^{-1}}^2\right)$$

From the above, using the fact that $x \leq 2\log(1+x)$ for any $x \in [0,1]$ and $(\kappa/2)\|x_n\|_{\widetilde{V}_{k,s}^{-1}}^2 \leq (\kappa/2)\|x_n\|_2^2/\lambda \leq 1$ from $\kappa < 1$, we have

$$\sum_{t \in [T]} \max_{n \in S_{k,t}} (\kappa/2)\|x_n\|_{V_{k,t}^{-1}}^2 \leq \sum_{t \in [T]} \min\left\{\max_{n \in S_{k,t}} (\kappa/2)\|x_n\|_{V_{k,t-1}^{-1}}^2, 1\right\}$$
$$\leq 2 \sum_{t \in [T]} \log\left(1 + \sum_{n \in S_{k,t}} (\kappa/2)\|x_n\|_{V_{k,t-1}^{-1}}^2\right)$$
$$= 2 \log \prod_{t \in [T]}\left(1 + \sum_{n \in S_{k,t}} (\kappa/2)\|x_n\|_{V_{k,t-1}^{-1}}^2\right)$$
$$\leq 2 \log\left(\frac{\det(V_{k,T+1})}{\lambda^d}\right). \qquad (29)$$

From Lemma 10 in Abbasi-Yadkori et al. [1] with $\kappa < 1$, we can show that

$$\det(V_{k,T+1}) \leq (\lambda + (TL/d))^d.$$

Then from the above inequality and Eq.(29), we can conclude the proof. $\square$

Then from Eqs.(27), (28), and Lemma 4, we can conclude that

$$
\mathcal{R}^{\pi}(T) = \sum_{t\in[T]}\sum_{n\in[N]} \mathbb{E}[Q_n(t)(\mu(n|S_{k_{n,t}^*,t},\theta_{k_{n,t}^*}) - \mu(n|S_{k_{n,t},t},\theta_{k_{n,t}}))]
$$

$$
= \sum_{t\in[T]}\sum_{n\in[N]} \mathbb{E}[Q_n(t)(\mu(n|S_{k_{n,t}^*,t},\theta_{k_{n,t}^*}) - \mu(n|S_{k_{n,t},t},\theta_{k_{n,t}}))\mathbb{1}(E_t^1)]
$$

$$
+ \sum_{t\in[T]}\sum_{n\in[N]} \mathbb{E}[Q_n(t)(\mu(n|S_{k_{n,t}^*,t},\theta_{k_{n,t}^*}) - \mu(n|S_{k_{n,t},t},\theta_{k_{n,t}}))\mathbb{1}((E_t^1)^c)]
$$

$$
= \sum_{t\in[T]}\sum_{n\in[N]} \mathbb{E}[Q_n(t)(\mu(n|S_{k_{n,t}^*,t},\theta_{k_{n,t}^*}) - \mu(n|S_{k_{n,t},t},\theta_{k_{n,t}}))\mathbb{1}(E_t^1)] + \sum_{t\in[T]}\sum_{n\in[N]} t\mathbb{P}((E_t^1)^c)]
$$

$$
\leq \sum_{t\in[T]}\sum_{n\in[N]} \mathbb{E}[Q_n(t)(\widetilde{\mu}_t^{UCB}(n|S_{k_{n,t}^*,t},\hat{\theta}_{k_{n,t}^*,t}) - \mu(n|S_{k_{n,t},t},\theta_{k_{n,t}}))\mathbb{1}(E_t^1)] + \mathcal{O}(N/T)
$$

$$
\leq \sum_{t\in[T]}\sum_{n\in[N]} \mathbb{E}[Q_n(t)(\widetilde{\mu}_t^{UCB}(n|S_{k_{n,t},t},\hat{\theta}_{k_{n,t},t}) - \mu(n|S_{k_{n,t},t},\theta_{k_{n,t}}))\mathbb{1}(E_t^1)] + \mathcal{O}(N/T)
$$

$$
= \sum_{t\in[T]} \mathbb{E}[\sum_{k\in[K]}\sum_{n\in S_{k,t}} Q_n(t)(\widetilde{\mu}_t^{UCB}(n|S_{k,t},\hat{\theta}_{k,t}) - \mu(n|S_{k,t},\theta_k))\mathbb{1}(E_t^1)] + \mathcal{O}(N/T)
$$

$$
\leq 2\mathbb{E}\left[\beta_T \sum_{t\in[T]}\sum_{k\in[K]} \max_{n\in S_{k,t}} \|x_n\|_{V_{k,t}^{-1}} Q_n(t)\right] + \mathcal{O}(N/T)
$$

$$
\leq 2\mathbb{E}\left[\max_{t\in[T],n\in[N]} Q_n(t)\beta_T \sum_{t\in[T]}\sum_{k\in[K]} \max_{n\in S_{k,t}} \|x_n\|_{V_{k,t}^{-1}}\right] + \mathcal{O}(N/T)
$$

$$
\leq 2\mathbb{E}\left[\max_{t\in[T],n\in[N]} Q_n(t)\beta_T \sqrt{KT \sum_{t\in[T]}\sum_{k\in[K]} \max_{n\in S_{k,t}} \|x_n\|_{V_{k,t}^{-1}}^2}\right] + \mathcal{O}(N/T)
$$

$$
= \widetilde{\mathcal{O}}\left(\frac{d}{\kappa}\sqrt{KT}Q_{\max}\right),
$$

where the last equality comes from Lemma 4.

Now we provide the worst-case regret bound of $\mathcal{R}^{\pi}(T) = \widetilde{\mathcal{O}}\left(\left(\frac{dNK\min\{N,K\}^3}{\kappa^2\epsilon^3}\right)^{1/4} T^{3/4}\right)$ in the following.

We define event $E_t^1 = \{\|\hat{\theta}_{k,t} - \theta_k^*\|_{V_{k,t}} \leq \beta_t$ for all $k \in [K]\}$ where $\beta_t = C_1\sqrt{\lambda + \frac{d}{\kappa}\log(1 + tLK/d\lambda)}$, which holds with high probability as $\mathbb{P}(E_t^1) \geq 1 - 1/t^2$ from Lemma 1. We also define $E_{n,t}^2 = \{\max_{m\in S_{k,t}} \|x_m\|_{V_{k,t}^{-1}} \leq \zeta$ for $k = k_{n,t}\}$ for some constant $C_2 > 0$.

Then we have

$$
\mathcal{R}^{\pi}(T) = \sum_{t\in[T]}\sum_{n\in[N]} \mathbb{E}[(\mu(n|S_{k_{n,t}^*,t},\theta_{k_{n,t}^*}) - \mu(n|S_{k_{n,t},t},\theta_{k_{n,t}}))Q_n(t)]
$$

$$
\leq \sum_{t\in[T]}\sum_{n\in[N]} \mathbb{E}[(\mu(n|S_{k_{n,t}^*,t},\theta_{k_{n,t}^*}) - \mu(n|S_{k_{n,t},t},\theta_{k_{n,t}}))Q_n(t)\mathbb{1}(E_t^1\cap E_{n,t}^2)]
$$

$$
+ \sum_{t\in[T]}\sum_{n\in[N]} \mathbb{E}[(\mu(n|S_{k_{n,t}^*,t},\theta_{k_{n,t}^*}) - \mu(n|S_{k_{n,t},t},\theta_{k_{n,t}}))Q_n(t)(\mathbb{1}((E_t^1)^c) + \mathbb{1}((E_{n,t}^2)^c))].
$$

$$
\tag{30}
$$

Since $x/(1+x)$ is a non-decreasing function for $x > -1$ and $x_n^\top \theta_{k_{n,t}^*} \le x_n^\top \hat\theta_{k_{n,t}^*,t} + \beta_t \|x_n\|_{V_{k_{n,t}^*,t}^{-1}}$ under $E_t^1$, we have $\mu(n|S_{k_{n,t}^*,t}, \theta_{k_{n,t}^*}) \le \widetilde\mu_t^{UCB}(n|S_{k_{n,t}^*,t}, \hat\theta_{k_{n,t}^*,t})$. Then for the first term of Eq.(30), we have

$$
\sum_{t\in[T]} \mathbb{E}\left[\sum_{n\in[N]} (\mu(n|S_{k_{n,t}^*,t}, \theta_{k_{n,t}^*}) - \mu(n|S_{k_{n,t},t}, \theta_{k_{n,t}}))Q_n(t)\mathbb{1}(E_t^1 \cap E_{n,t}^2)\right]
$$

$$
\le \sum_{t\in[T]} \mathbb{E}\left[\sum_{n\in[N]} (\widetilde\mu_t^{UCB}(n|S_{k_{n,t}^*,t}, \hat\theta_{k_{n,t}^*,t}) - \mu(n|S_{k_{n,t},t}, \theta_{k_{n,t}}))Q_n(t)\mathbb{1}(E_t^1 \cap E_{n,t}^2)\right]
$$

$$
\le \sum_{t\in[T]} \mathbb{E}\left[\sum_{n\in[N]} (\widetilde\mu_t^{UCB}(n|S_{k_{n,t},t}, \hat\theta_{k_{n,t},t}) - \mu(n|S_{k_{n,t},t}, \theta_{k_{n,t}}))Q_n(t)\mathbb{1}(E_t^1 \cap E_{n,t}^2)\right]
$$

$$
\le \sum_{t\in[T]} \mathbb{E}\left[\sum_{n\in[N]} 2\beta_t \|x_n\|_{V_{k_{n,t},t}^{-1}} Q_n(t)\mathbb{1}(E_t^1 \cap E_{n,t}^2)\right]
$$

$$
\le \sum_{t\in[T]} \sum_{n\in[N]} 2\beta_t \zeta \mathbb{E}\left[Q_n(t)\right], \tag{31}
$$

where the second inequality comes from the UCB strategy of the algorithm, the last second inequality is obtained from Lemma 2, and the last inequality is obtained from $E_{n,t}^2$.

Now we provide a bound for the second term of Eq.(30). From Eq. (23), we have

$$
\sum_{t\in[T]} \sum_{n\in[N]} \mathbb{E}\left[(\mu(n|S_{k_{n,t}^*,t}, \theta_{k_{n,t}^*}) - \mu(n|S_{k_{n,t},t}, \theta_{k_{n,t}}))Q_n(t)\mathbb{1}((E_t^1)^c)\right]
$$

$$
= \mathcal{O}\left(\sum_{t\in[T]} \sum_{n\in[N]} t(1/t^2)\right) = \mathcal{O}(N\log(T)). \tag{32}
$$

Let $\mathcal{T}_n$ be the set of time steps $t \in [T]$ such that $Q_n(t) \ne 0$ and let $e_T = \sum_{n\in[N]} \sum_{t\in\mathcal{T}_n} \mathbb{1}((E_{n,t}^2)^c)$ and $h = \lceil 1/\zeta\beta_T \rceil$. Then if $t \le h$, we have $Q_n(t) \le t \le h$. Otherwise, we have

$$
Q_n(t) \le \sum_{s=t-h+1}^{t} (1/h)(Q_n(s) + (t-s)) \le (1/h)\sum_{s=1}^{t} Q_n(s) + (1/h)h^2 = (1/h)\sum_{s=1}^{t} Q_n(s) + h.
$$

Then, by following the steps in Eq.(24), we have

$$
\sum_{t\in[T]} \sum_{n\in[N]} \mathbb{E}[(\mu(n|S_{k_{n,t}^*,t}, \theta_{k_{n,t}^*}) - \mu(n|S_{k_{n,t},t}, \theta_{k_{n,t}}))Q_n(t)\mathbb{1}((E_{n,t}^2)^c)]
$$

$$
\le \sum_{t\in[T]} \sum_{n\in[N]} \zeta\beta_T \mathbb{E}\left[Q_n(t)\right] + 2\mathbb{E}[e_T]/\zeta\beta_T. \tag{33}
$$

Now we provide a bound for $\mathbb{E}[e_T]$. Define $N_{n,k}(t) = \sum_{s=1}^{t-1} \mathbb{1}(n \in S_{k,s})$ and $\widetilde{V}_{k,t} = \frac{\kappa}{2} \sum_{s=1}^{t-1} \sum_{n \in S_{k,s}} x_n x_n^\top$. Then, we have

$$
\begin{aligned}
e_T &= \sum_{n \in [N]} \sum_{t \in \mathcal{T}_n} \mathbb{1}((E_{n,t}^2)^c) \\
&\leq \sum_{n \in [N]} \sum_{t \in \mathcal{T}_n} \mathbb{1}(\|x_n\|_{V_{k_{n,t},t}^{-1}} \geq \zeta) \\
&\leq \sum_{n \in [N]} \sum_{t \in \mathcal{T}_n} \mathbb{1}(\|x_n\|_{\widetilde{V}_{k_{n,t},t}^{-1}} \geq \zeta) \\
&\leq \sum_{n \in [N]} \sum_{t \in \mathcal{T}_n} \mathbb{1}(1/N_{n,k_{n,t}}(t) \geq (\kappa/2)\zeta^2) \\
&\leq \sum_{n \in [N]} \sum_{t \in \mathcal{T}_n} \mathbb{1}(N_{n,k_{n,t}}(t) \leq 2/\kappa\zeta^2) \\
&\leq \sum_{n \in [N]} \sum_{k \in [K]} \sum_{t \in \mathcal{T}_n} \mathbb{1}(N_{n,k}(t) \leq 2/\kappa\zeta^2 \text{ and } k_{n,t} = k) \\
&\leq 2NK/\kappa\zeta^2.
\end{aligned}
\tag{34}
$$

Then from Eqs.(33), (34), we have

$$
\begin{aligned}
&\sum_{t \in [T]} \sum_{n \in [N]} \mathbb{E}[(\mu(n|S_{k_{n,t}^*,t}, \theta_{k_{n,t}^*}) - \mu(n|S_{k_{n,t},t}, \theta_{k_{n,t}}))Q_n(t)\mathbb{1}((E_{n,t}^2)^c)] \\
&\leq \sum_{t \in [T]} \sum_{n \in [N]} \zeta\beta_T \mathbb{E}[Q_n(t)] + \mathcal{O}(NK/\kappa\zeta^3\beta_T).
\end{aligned}
\tag{35}
$$

By putting the results of Eqs. (30), (31), (32), (35), and Theorem 1, by setting $\zeta = (\epsilon NK/\min\{N,K\}T\kappa\beta_T^2)^{1/4}$, for large enough $T$, we can obtain

$$
\begin{aligned}
\mathcal{R}^\pi(T) &= \mathcal{O}\left( \zeta\beta_T \sum_{t \in [T]} \sum_{n \in [N]} \mathbb{E}[Q_n(t)] + \frac{NK}{\kappa\zeta^3\beta_T} + N\log(T) \right) \\
&= \mathcal{O}\left( \frac{\zeta\beta_T \min\{N,K\}T}{\epsilon} + \frac{NK}{\kappa\zeta^3\beta_T} + N\log(T) \right) \\
&= \mathcal{O}\left( \frac{\beta_T^{1/2}T^{3/4}(NK\min\{N,K\}^3)^{1/4}}{\kappa^{1/4}\epsilon^{3/4}} + N\log(T) \right) \\
&= \widetilde{\mathcal{O}}\left( \left( \frac{dNK\min\{N,K\}^3}{\kappa^2\epsilon^3} \right)^{1/4} T^{3/4} \right)
\end{aligned}
\tag{36}
$$

### A.6  Proof of Theorm 3

We first define the set of queues $\mathbf{Q}(t) = [Q_n(t) : n \in [N]]$ and a Lyapunov function as $\mathcal{V}(\mathbf{Q}(t)) = \sum_{n \in [N]} Q_n(t)^2$. For simplicity, we use $D_n(t)$ for $D_n(t|S_{k_{n,t},t})$, $D_n^*(t)$ for $D_n(t|S_{k_{n,t}^*,t})$, and $\widetilde{\mu}_t^{TS}(n|S_{k,t})$ for $\widetilde{\mu}_t^{TS}(n|S_{k,t}, \{\widetilde{\theta}_{k,t}^{(i)}\}_{i \in [M]})$ when there is no confusion. Then we analyze the Lya-

punov drift as follows.

$$\sum_{t\in[T]} \mathbb{E}\left[\mathcal{V}(\mathbf{Q}(t+1)) - \mathcal{V}(\mathbf{Q}(t))\right]$$

$$= \sum_{t\in[T]}\sum_{n\in[N]} \mathbb{E}\left[(Q_n(t) + A_n(t) - D_n(t))^{+2} - Q_n(t)^2\right]$$

$$= \sum_{t\in[T]}\sum_{n\in[N]} \mathbb{E}\left[(Q_n(t) + A_n(t) - D_n^*(t))^2 - Q_n(t)^2\right]$$

$$+ \sum_{t\in[T]}\sum_{n\in[N]} \mathbb{E}\left[(Q_n(t) + A_n(t) - D_n(t))^{+2} - (Q_n(t) + A_n(t) - D_n^*(t))^2\right]$$

$$\leq 7\min\{N,K\}T - \sum_{t\in[T]}\sum_{n\in[N]} 2\epsilon\mathbb{E}[Q_n(t)] + 2\sum_{t\in[T]}\sum_{n\in[N]} \mathbb{E}\left[(D_n^*(t) - D_n(t))Q_n(t)\right], \quad (37)$$

where the last inequality can be obtained by following Eqs.(19) and (20).

Define event $E_t^1 = \{\|\hat{\theta}_{k,t} - \theta_k\|_{V_{k,t}} \leq \beta_t$ for all $k \in [K]\}$ where $\beta_t = C_1\sqrt{\lambda + \frac{d}{\kappa}\log(1 + tLK/d\lambda)}$, which holds with high probability as $\mathbb{P}(E_t^1) \geq 1 - 1/t^2$ from Lemma 1. We let $\gamma_t = \beta_t\sqrt{d\log(MKt)}$ and filtration $\mathcal{F}_{t-1}$ be the $\sigma$-algebra generated by random variables before time $t$.

**Lemma 5** (Lemma 10 in Oh and Iyengar [43]). *For any given $\mathcal{F}_{t-1}$, with probability at least $1 - \mathcal{O}(1/t^2)$, for all $n \in [N]$ and $k \in [K]$, we have*

$$|h_{n,k,t}^{TS} - x_n^\top\hat{\theta}_{k,t}| \leq \gamma_t\|x_n\|_{V_{k,t}^{-1}}.$$

**Lemma 6.** *With probability at least $1 - \mathcal{O}(1/t^2)$, for all $n \in [N]$ and $k \in [K]$, we have*

$$\widetilde{\mu}_t^{TS}(n|S_{k,t}) - \mu(n|S_{k,t}, \hat{\theta}_{k,t}) \leq \gamma_t\|x_n\|_{V_{k,t}^{-1}}.$$

*Proof.* From Lemma 5, with probability at least $1 - \mathcal{O}(1/t^2)$, we have $|h_{n,k,t}^{TS} - x_n^\top\hat{\theta}_{k,t}| \leq \gamma_t\|x_n\|_{V_{k,t}^{-1}}$. Let $u_{n,k,t} = x_n^\top\hat{\theta}_{k,t}$. Then by the mean value theorem, there exists $\bar{u}_{n,k,t} = (1-c)h_{n,k,t}^{TS} + cu_{n,k,t}$ for some $c \in (0,1)$ satisfying, for any $n \in S_{k,t}$ and $k \in [K]$,

$$\widetilde{\mu}_t^{TS}(n|S_{k,t}) - \mu(n|S_k, \hat{\theta}_{k,t}) = \frac{\exp(h_{n,k,t}^{TS})}{1 + \sum_{m\in S_{k,t}}\exp(h_{m,k,t}^{TS})} - \frac{\exp(u_{n,k,t})}{1 + \sum_{m\in S_{k,t}}\exp(u_{m,k,t})}$$

$$= \nabla_{v_n}\left(\frac{\sum_{m\in S_{k,t}}\exp(v_m)}{1 + \sum_{m\in S_{k,t}}\exp(v_m)}\right)\Bigg|_{v_n=\bar{u}_{n,k,t}} (h_{n,k,t}^{TS} - u_{n,k,t})$$

$$\leq \frac{\exp(\bar{u}_{n,k,t})|h_{n,k,t}^{TS} - u_{n,k,t}|}{1 + \sum_{n\in S_{k,t}}\exp(\bar{u}_{n,k,t})}$$

$$\leq |h_{n,k,t}^{TS} - u_{n,k,t}|$$

$$\leq \gamma_t\|x_n\|_{V_{k,t}^{-1}}.$$

$\square$

Then we define $E_t^2 = \{\widetilde{\mu}_t^{TS}(n|S_{k,t}) - \mu(n|S_{k,t}, \hat{\theta}_{k,t}) \leq \gamma_t\|x_n\|_{V_{k,t}^{-1}}; \forall n \in [N], \forall k \in [K]\}$, which holds with probability at least $1 - \mathcal{O}(1/t^2)$ from Lemma 6. We also define $E_{n,t}^3 = \{\|x_n\|_{V_{k,t}^{-1}} \leq$

$\epsilon/C_2(\gamma_t + \beta_t); \forall k \in [K]\}$ for some constant $C_2 \geq 17\sqrt{e\pi}$. Then, for bounding Eq.(37), we have

$$\sum_{t\in[T]}\sum_{n\in[N]}\mathbb{E}[(D_n^*(t) - D_n(t)Q_n(t)]$$

$$=\sum_{t\in[T]}\sum_{n\in[N]}\mathbb{E}[\mathbb{E}[(D_n^*(t) - D_n(t)Q_n(t)|Q_n(t)]]$$

$$=\mathbb{E}[\sum_{t\in[T]}\sum_{n\in[N]}\mu(n|S_{k_{n,t}^*,t},\theta_{k_{n,t}^*}) - \mu(n|S_{k_{n,t},t},\theta_{k_{n,t}}))Q_n(t)]$$

$$\leq\sum_{t\in[T]}\sum_{n\in[N]}\mathbb{E}[(\mu(n|S_{k_{n,t}^*,t},\theta_{k_{n,t}^*}) - \mu(n|S_{k_{n,t},t},\theta_{k_{n,t}}))Q_n(t)\mathbb{1}(E_t^1 \cap E_t^2 \cap E_{n,t}^3)]$$

$$+\sum_{t\in[T]}\sum_{n\in[N]}\mathbb{E}[(\mu(n|S_{k_{n,t}^*,t},\theta_{k_{n,t}^*}) - \mu(n|S_{k_{n,t},t},\theta_{k_{n,t}}))Q_n(t)(\mathbb{1}((E_t^1)^c) + \mathbb{1}((E_t^2)^c) + \mathbb{1}((E_{n,t}^3)^c))].$$

$$(38)$$

We provide a bound for the first term of Eq.(38). We first have

$$\sum_{t\in[T]}\sum_{n\in[N]}\mathbb{E}[(\mu(n|S_{k_{n,t}^*,t},\theta_{k_{n,t}^*}) - \mu(n|S_{k_{n,t},t},\theta_{k_{n,t}}))Q_n(t)\mathbb{1}(E_t^1 \cap E_t^2 \cap E_{n,t}^3)]$$

$$\leq\sum_{t\in[T]}\sum_{n\in[N]}\mathbb{E}[(\mu(n|S_{k_{n,t}^*,t},\theta_{k_{n,t}^*}) - \widetilde{\mu}_t^{TS}(n|S_{k_{n,t},t})$$

$$+ \widetilde{\mu}_t^{TS}(n|S_{k_{n,t},t}) - \mu(n|S_{k_{n,t},t},\theta_{k_{n,t}}))Q_n(t)\mathbb{1}(E_t^1 \cap E_t^2 \cap E_{n,t}^3)].$$

$$(39)$$

Recall that $\widetilde{\mu}_t^{UCB}(n|S_k,\hat{\theta}_{k,t}) = \frac{\exp(h_{n,k,t}^{UCB})}{1+\sum_{m\in S_k}\exp(h_{m,k,t}^{UCB})}$. Then we can show that since $x/(1+x)$ is a non-decreasing function for $x > -1$ and $x_n^\top\hat{\theta}_{k_{n,t},t} \leq x_n^\top\hat{\theta}_{k_{n,t},t} + \beta_t\|x_n\|_{V_{k_{n,t},t}^{-1}}$ under $E_t^1$, with Lemma 2, we have

$$\mu(n|S_{k_{n,t},t},\hat{\theta}_{k_{n,t},t}) - \mu(n|S_{k_{n,t},t},\theta_{k_{n,t}}) \leq \widetilde{\mu}_t^{UCB}(n|S_{k_{n,t},t},\hat{\theta}_{k_{n,t},t}) - \mu(n|S_{k_{n,t},t},\theta_{k_{n,t}}) \leq 2\beta_t\|x_n\|_{V_{k_{n,t},t}^{-1}}.$$

From the above inequality, the last two terms in Eq.(39) are bounded as

$$\sum_{t\in[T]}\sum_{n\in[N]}\mathbb{E}\left[\widetilde{\mu}_t^{TS}(n|S_{k_{n,t},t}) - \mu(n|S_{k_{n,t},t},\theta_{k_{n,t}}))Q_n(t)\mathbb{1}(E_t^1 \cap E_t^2 \cap E_{n,t}^3)\right]$$

$$\leq\sum_{t\in[T]}\sum_{n\in[N]}\mathbb{E}\left[(\widetilde{\mu}_t^{TS}(n|S_{k_{n,t},t}) - \mu(n|S_{k_{n,t},t},\hat{\theta}_{k_{n,t},t})\right.$$

$$\left.+ \mu(n|S_{k_{n,t},t},\hat{\theta}_{k_{n,t},t}) - \mu(n|S_{k_{n,t},t},\theta_{k_{n,t}}))Q_n(t)\mathbb{1}(E_t^1 \cap E_t^2 \cap E_{n,t}^3)\right]$$

$$\leq\sum_{t\in[T]}\sum_{n\in[N]}\mathbb{E}[(\widetilde{\mu}_t^{TS}(n|S_{k_{n,t},t}) - \mu(n|S_{k_{n,t},t},\hat{\theta}_{k_{n,t},t}) + 2\beta_t\|x_n\|_{V_{k_{n,t},t}^{-1}})Q_n(t)\mathbb{1}(E_t^1 \cap E_t^2 \cap E_{n,t}^3)]$$

$$=\sum_{t\in[T]}\sum_{k\in[K]}\mathbb{E}\left[\sum_{n\in S_{k,t}}(\widetilde{\mu}_t^{TS}(n|S_{k,t}) - \mu(n|S_{k,t},\hat{\theta}_{k,t}) + 2\beta_t\|x_n\|_{V_{k,t}^{-1}})Q_n(t)\mathbb{1}(E_t^1 \cap E_t^2 \cap E_{n,t}^3)\right]$$

$$\leq \sum_{t\in[T]}\sum_{k\in[K]}\mathbb{E}\left[\sum_{n\in S_{k,t}}(\gamma_t+2\beta_t)\|x_n\|_{V_{k,t}^{-1}}Q_n(t)\mathbb{1}(E_t^1\cap E_t^2\cap E_{n,t}^3)\right]$$

$$\leq \sum_{t\in[T]}\sum_{k\in[K]}\mathbb{E}\left[\sum_{n\in S_{k,t}}\frac{(\gamma_t+2\beta_t)\epsilon}{C_2(\gamma_t+\beta_t)}Q_n(t)\mathbb{1}(E_t^1\cap E_t^2\cap E_{n,t}^3)\right]$$

$$\leq \sum_{t\in[T]}\sum_{k\in[K]}\mathbb{E}\left[\sum_{n\in S_{k,t}}\frac{(\gamma_t+2\beta_t)\epsilon}{C_2(\gamma_t+\beta_t)}Q_n(t)\mathbb{1}(E_t^1)\right]$$

$$= \sum_{t\in[T]}\sum_{k\in[K]}\mathbb{E}\left[\mathbb{E}\left[\sum_{n\in S_{k,t}}\frac{(\gamma_t+2\beta_t)\epsilon}{C_2(\gamma_t+\beta_t)}Q_n(t)|E_t^1,\mathcal{F}_{t-1}\right]\mathbb{P}(E_t^1|\mathcal{F}_{t-1})\right]. \qquad (40)$$

Now we provide a bound for the first two terms in Eq.(39).

We define sets

$$\widetilde{\Theta}_t = \left\{\{\theta_k^{(i)}\}_{i\in[M],k\in[K]} : \left|\max_{i\in[M]}x_n^\top\theta_k^{(i)}-x_n^\top\hat{\theta}_{k,t}\right| \leq \gamma_t\|x_n\|_{V_{k,t}^{-1}}; \ \forall n\in[N],\forall k\in[K]\right\} \text{ and}$$

$$\widetilde{\Theta}_t^{opt} = \left\{\{\theta_k^{(i)}\}_{i\in[M],k\in[K]} : \sum_{k\in[K]}\sum_{n\in S_{k,t}}\widetilde{\mu}_t^{TS}(n|S_{k,t},\{\theta_k^{(i)}\}_{i\in[M]})Q_n(t)\right.$$

$$\left.> \sum_{k\in[K]}\sum_{n\in S_{k,t}^*}\mu(n|S_{k,t}^*,\theta_k)Q_n(t)\right\}\cap\widetilde{\Theta}_t.$$

Then we define event $E_t(\widetilde{\theta}) = \{\{\widetilde{\theta}_{k,t}^{(i)}\}_{i\in[M],k\in[K]} \in \widetilde{\Theta}_t^{opt}\}$. Recall $h_{n,k,t}^{TS} = \max_{i\in[M]}x_n^\top\widetilde{\theta}_{k,t}^{(i)}$. Then we have

$$\mathbb{E}\left[\mathbb{E}\left[\sum_{k\in[K]}\left(\sum_{n\in S_{k,t}^*}\mu(n|S_{k,t}^*,\theta_k)Q_n(t)-\sum_{n\in S_{k,t}}\widetilde{\mu}_t^{TS}(n|S_{k,t},\{\widetilde{\theta}_{k,t}^{(i)}\}_{i\in[M]})Q_n(t)\right)\mathbb{1}(E_t^1\cap E_t^2\cap E_{n,t}^3)|\mathcal{F}_{t-1}\right]\right]$$

$$\leq \mathbb{E}\left[\mathbb{E}\left[\left(\sum_{k\in[K]}\sum_{n\in S_{k,t}^*}\mu(n|S_{k,t}^*,\theta_k)Q_n(t)\right.\right.\right.$$

$$\left.\left.\left.- \inf_{\{\theta_l^{(i)}\}_{i\in[M],l\in[K]}\in\widetilde{\Theta}_t}\max_{\{S_k\}_{k\in[K]}\in\mathcal{M}(\mathcal{N}_t)}\sum_{k\in[K]}\sum_{n\in S_k}\widetilde{\mu}_t^{TS}(n|S_k,\{\theta_k^{(i)}\}_{i\in[M]})Q_n(t)\right)\mathbb{1}(E_t^1\cap E_t^2\cap E_{n,t}^3)|\mathcal{F}_{t-1}\right]\right]$$

$$= \mathbb{E}\left[\mathbb{E}\left[\left(\sum_{k\in[K]}\sum_{n\in S_{k,t}^*}\mu(n|S_{k,t}^*,\theta_k)Q_n(t)\right.\right.\right.$$

$$\left.\left.\left.- \inf_{\{\theta_l^{(i)}\}_{i\in[M],l\in[K]}\in\widetilde{\Theta}_t}\max_{\{S_k\}_{k\in[K]}\in\mathcal{M}(\mathcal{N}_t)}\sum_{k\in[K]}\sum_{n\in S_k}\widetilde{\mu}_t^{TS}(n|S_k,\{\theta_k^{(i)}\}_{i\in[M]})Q_n(t)\right)\mathbb{1}(E_t^1\cap E_t^2\cap E_{n,t}^3)|\mathcal{F}_{t-1},E_t(\widetilde{\theta})\right]\right]$$

$$\leq \mathbb{E}\left[\mathbb{E}\left[\left(\sum_{k\in[K]}\sum_{n\in S_{k,t}}\widetilde{\mu}_t^{TS}(n|S_{k,t},\{\widetilde{\theta}_{k,t}^{(i)}\}_{i\in[M]})\right.\right.\right.$$

$$\left.\left.\left.- \inf_{\{\theta_l^{(i)}\}_{i\in[M],l\in[K]}\in\widetilde{\Theta}_t}\sum_{k\in[K]}\sum_{n\in S_{k,t}}\widetilde{\mu}_t^{TS}(n|S_{k,t},\{\theta_k^{(i)}\}_{i\in[M]})\right)Q_n(t)\mathbb{1}(E_t^1\cap E_t^2\cap E_{n,t}^3)|\mathcal{F}_{t-1},E_t(\widetilde{\theta})\right]\right]$$

$$= \mathbb{E}\left[\mathbb{E}\left[\sup_{\{\theta_l^{(i)}\}_{i\in[M],l\in[K]}\in\widetilde{\Theta}_t}\sum_{k\in[K]}\sum_{n\in S_{k,t}}\left(\widetilde{\mu}_t^{TS}(n|S_{k,t},\{\widetilde{\theta}_{k,t}^{(i)}\}_{i\in[M]})\right.\right.\right.$$

$$\left.\left.\left.-\widetilde{\mu}_t^{TS}(n|S_{k,t},\{\theta_k^{(i)}\}_{i\in[M]})\right)Q_n(t)\mathbb{1}(E_t^1\cap E_t^2\cap E_{n,t}^3)|\mathcal{F}_{t-1},E_t(\widetilde{\theta})\right]\right]$$

$$\leq \mathbb{E}\left[\mathbb{E}\left[\sup_{\{\theta_l^{(i)}\}_{i\in[M],l\in[K]}\in\widetilde{\Theta}_t}\sum_{k\in[K]}\sum_{n\in S_{k,t}}\left|h_{n,k,t}^{TS}-\max_{j\in[M]}x_n^\top\theta_k^{(j)}\right|Q_n(t)\mathbb{1}(E_t^1\cap E_t^2\cap E_{n,t}^3)|\mathcal{F}_{t-1},E_t(\widetilde{\theta})\right]\right]$$

$$= \mathbb{E}\left[\mathbb{E}\left[\sup_{\{\theta_l^{(i)}\}_{i\in[M],l\in[K]}\in\widetilde{\Theta}_t}\sum_{k\in[K]}\sum_{n\in S_{k,t}}\left|h_{n,k,t}^{TS}-x_n^\top\hat{\theta}_{k,t}+x_n^\top\hat{\theta}_{k,t}-\max_{j\in[M]}x_n^\top\theta_k^{(j)}\right|\right.\right.$$

$$\left.\left.\times Q_n(t)\mathbb{1}(E_t^1\cap E_t^2\cap E_{n,t}^3)|\mathcal{F}_{t-1},E_t(\widetilde{\theta})\right]\right]$$

$$\leq 2\gamma_t\mathbb{E}\left[\sum_{k\in[K]}\sum_{n\in S_{k,t}}\mathbb{E}\left[\|x_n\|_{V_{k,t}^{-1}}Q_n(t)\mathbb{1}(E_t^1\cap E_t^2\cap E_{n,t}^3)|\mathcal{F}_{t-1},E_t(\widetilde{\theta})\right]\right]$$

$$\leq 2\gamma_t\mathbb{E}\left[\sum_{k\in[K]}\sum_{n\in S_{k,t}}\mathbb{E}\left[\frac{\epsilon}{C_2(\gamma_t+\beta_t)}Q_n(t)\mathbb{1}(E_t^1\cap E_t^2\cap E_{n,t}^3)|\mathcal{F}_{t-1},E_t(\widetilde{\theta})\right]\right]$$

$$\leq 2\gamma_t\mathbb{E}\left[\sum_{k\in[K]}\sum_{n\in S_{k,t}}\mathbb{E}\left[\frac{\epsilon}{C_2(\gamma_t+\beta_t)}Q_n(t)\mathbb{1}(E_t^1)|\mathcal{F}_{t-1},E_t(\widetilde{\theta})\right]\right]$$

$$= \frac{2\gamma_t\epsilon}{C_2(\gamma_t+\beta_t)}\mathbb{E}\left[\sum_{k\in[K]}\sum_{n\in S_{k,t}}\mathbb{E}\left[Q_n(t)|\mathcal{F}_{t-1},E_t(\widetilde{\theta}),E_t^1\right]\times\mathbb{P}(E_t^1|E_t(\widetilde{\theta}),\mathcal{F}_{t-1})\right]$$

$$= \frac{2\gamma_t\epsilon}{C_2(\gamma_t+\beta_t)}\mathbb{E}\left[\sum_{k\in[K]}\sum_{n\in S_{k,t}}\mathbb{E}\left[Q_n(t)|\mathcal{F}_{t-1},E_t(\widetilde{\theta}),E_t^1\right]\mathbb{P}(E_t^1|\mathcal{F}_{t-1})\right], \tag{41}$$

where the first inequality is obtained by the event $E_t^2$, the second inequality is obtained from $E_t(\widetilde{\theta})$, the third inequality can be easily obtained by following some of the proof steps in Lemma 2, the third last inequality is obtained from the definition of $\widetilde{\Theta}_t$ and event $E_t(\widetilde{\theta})$, and the last equality comes from independence between $E_t^1$ and $E_t(\widetilde{\theta})$ given $\mathcal{F}_{t-1}$.

We provide a lemma below for further analysis.

**Lemma 7.** *For all $t\in[T]$, we have*

$$\mathbb{P}\left(\sum_{k\in[K]}\sum_{n\in S_{k,t}}\widetilde{\mu}_t^{TS}(n|S_{k,t})Q_n(t)>\sum_{k\in[K]}\sum_{n\in S_{k,t}^*}\mu(n|S_{k,t}^*,\theta_k)Q_n(t)|\mathcal{F}_{t-1},E_t^1\right)\geq 1/4\sqrt{e\pi}.$$

*Proof.* Given $\mathcal{F}_{t-1}$, $x_n^\top\widetilde{\theta}_{k,t}^{(i)}$ follows Gaussian distribution with mean $x_n^\top\hat{\theta}_{k,t}$ and standard deviation $\beta_t\|x_n\|_{V_{k,t}^{-1}}$. Then we have

$$\mathbb{P}\left(\max_{i\in[M]}x_n^\top\widetilde{\theta}_{k,t}^{(i)}>x_n^\top\theta_k|\mathcal{F}_{t-1},E_t^1\right)=1-\mathbb{P}\left(x_n^\top\theta_{k,t}^{(i)}\leq x_n^\top\theta_k;\forall i\in[M]|\mathcal{F}_{t-1},E_t^1\right)$$

$$=1-\mathbb{P}\left(Z_i\leq\frac{x_n^\top\theta_k-x_n^\top\hat{\theta}_{k,t}}{\beta_t\|x_n\|_{V_{k,t}^{-1}}};\forall i\in[M]|\mathcal{F}_{t-1},E_t^1\right)$$

$$\geq 1-\mathbb{P}\left(Z\leq 1\right)^M,$$

where $Z_i$ and $Z$ are standard normal random variables. Then we can show that

$$\mathbb{P}\left(\sum_{k\in[K]}\sum_{n\in S_{k,t}}\widetilde{\mu}_t^{TS}(n|S_{k,t})Q_n(t) > \sum_{k\in[K]}\sum_{n\in S_{k,t}^*}\mu(n|S_{k,t}^*,\theta_k)Q_n(t)|\mathcal{F}_{t-1},E_t^1\right)$$

$$\geq \mathbb{P}\left(\sum_{k\in[K]}\sum_{n\in S_{k,t}^*}\widetilde{\mu}_t^{TS}(n|S_{k,t}^*)Q_n(t) > \sum_{k\in[K]}\sum_{n\in S_{k,t}^*}\mu(n|S_{k,t}^*,\theta_k)Q_n(t)|\mathcal{F}_{t-1},E_t^1\right)$$

$$\geq \mathbb{P}\left(\max_{i\in[M]} x_n^\top\widetilde{\theta}_{k,t}^{(i)} > x_n^\top\theta_k; \forall n\in S_{k,t}^*, \forall k\in[K]|\mathcal{F}_{t-1},E_t^1\right)$$

$$\geq 1 - LK\mathbb{P}(Z\leq 1)^M$$

$$\geq 1 - LK(1 - 1/4\sqrt{e\pi})^M$$

$$\geq \frac{1}{4\sqrt{e\pi}},$$

where the second last inequality is obtained from $\mathbb{P}(Z \leq 1) \leq 1 - 1/4\sqrt{e\pi}$ using the anti-concentration of standard normal distribution, and the last inequality comes from $M = \lceil 1 - \frac{\log KL}{\log(1-1/4\sqrt{e\pi})}\rceil$. $\qquad\square$

From Lemmas 5 and 7, for $t \geq t_0$ for some constant $t_0 > 0$, we have

$$\mathbb{P}(E_t(\widetilde{\theta})|\mathcal{F}_{t-1},E_t^1)$$

$$= \mathbb{P}\left(\sum_{k\in[K]}\sum_{n\in S_{k,t}}\widetilde{\mu}_t^{TS}(n|S_{k,t}) > \sum_{k\in[K]}\sum_{n\in S_{k,t}^*}\mu(n|S_{k,t}^*,\theta_k) \text{ and } \{\widetilde{\theta}_{k,t}^{(i)}\}_{i\in[M],k\in[K]}\in\widetilde{\Theta}_t|\mathcal{F}_{t-1},E_t^1\right)$$

$$= \mathbb{P}\left(\sum_{k\in[K]}\sum_{n\in S_{k,t}}\widetilde{\mu}_t^{TS}(n|S_{k,t})Q_n(t) > \sum_{k\in[K]}\sum_{n\in S_{k,t}^*}\mu(n|S_{k,t}^*,\theta_k)Q_n(t)|\mathcal{F}_{t-1},E_t^1\right)$$

$$\quad - \mathbb{P}(\{\widetilde{\theta}_{k,t}^{(i)}\}_{i\in[M],k\in[K]}\notin\widetilde{\Theta}_t|\mathcal{F}_{t-1},E_t^1)$$

$$\geq 1/4\sqrt{e\pi} - \mathcal{O}(1/t^2)$$

$$\geq 1/8\sqrt{e\pi}.$$

For simplicity of the proof, we ignore the time steps before (constant) $t_0$, which does not affect our final result. Hence, we have

$$\mathbb{E}[\sum_{k\in[K]}\sum_{n\in S_{k,t}} Q_n(t)|\mathcal{F}_{t-1},E_t^1]$$

$$\geq \mathbb{E}[\sum_{k\in[K]}\sum_{n\in S_{k,t}} Q_n(t)|\mathcal{F}_{t-1},E_t^1,E_t(\widetilde{\theta})]\mathbb{P}(E_t(\widetilde{\theta})|\mathcal{F}_{t-1},E_t^1)$$

$$\geq \mathbb{E}[\sum_{k\in[K]}\sum_{n\in S_{k,t}} Q_n(t)|\mathcal{F}_{t-1},E_t^1,E_t(\widetilde{\theta})]1/8\sqrt{e\pi}. \tag{42}$$

With (41) and (42), we have

$$\mathbb{E}[(\sum_{k\in[K]}\sum_{n\in S_{k,t}^*}\mu(n|S_{k,t}^*,\theta_k) - \sum_{k\in[K]}\sum_{n\in S_{k,t}}\widetilde{\mu}_t^{TS}(n|S_{k,t}))Q_n(t)\mathbb{1}(E_t^1\cap E_t^2\cap E_{n,t}^3)|\mathcal{F}_{t-1}]$$

$$\leq \frac{2\gamma_t\epsilon}{C_2(\gamma_t+\beta_t)}\mathbb{E}\left[\sum_{k\in[K]}\sum_{n\in S_{k,t}} Q_n(t)|\mathcal{F}_{t-1},E_t(\widetilde{\theta}),E_t^1\right]\mathbb{P}(E_t^1|\mathcal{F}_{t-1})$$

$$\leq \frac{16\sqrt{e\pi}\gamma_t\epsilon}{C_2(\gamma_t+\beta_t)}\mathbb{E}[\sum_{k\in[K]}\sum_{n\in S_{k,t}} Q_n(t)|\mathcal{F}_{t-1},E_t^1]\mathbb{P}(E_t^1|\mathcal{F}_{t-1}). \tag{43}$$

Then for the first term of Eq.(38), from Eqs.(39), (40), (43), for some $C_3 > 0$ we have

$$\sum_{t\in[T]}\sum_{n\in[N]}\mathbb{E}[(\mu(n|S_{k_{n,t}^*,t},\theta_{k_{n,t}^*}) - \mu(n|S_{k_{n,t},t},\theta_{k_{n,t}}))Q_n(t)\mathbb{1}(E_t^1\cap E_t^2\cap E_{n,t}^3)]$$

$$\leq \sum_{t\in[T]}\sum_{n\in[N]}\mathbb{E}[(\mu(n|S_{k_{n,t}^*,t},\theta_{k_{n,t}^*}) - \widetilde{\mu}_t^{TS}(n|S_{k_{n,t},t})$$

$$+ \widetilde{\mu}_t^{TS}(n|S_{k_{n,t},t}) - \mu(n|S_{k_{n,t},t},\theta_{k_{n,t}}))Q_n(t)\mathbb{1}(E_t^1\cap E_t^2\cap E_{n,t}^3)]$$

$$\leq \sum_{t\in[T]}\sum_{k\in[K]}\mathbb{E}[\mathbb{E}[\sum_{n\in S_{k,t}}\frac{(17\sqrt{e\pi}\gamma_t + 2\beta_t)\epsilon}{C_2(\gamma_t + \beta_t)}Q_n(t)|\mathcal{F}_{t-1}, E_t^1]\mathbb{P}(E_t^1|\mathcal{F}_{t-1})]$$

$$\leq \sum_{t\in[T]}\sum_{n\in[N]}C_3\epsilon\mathbb{E}[\mathbb{E}[Q_n(t)|E_t^1,\mathcal{F}_{t-1}]\mathbb{P}(E_t^1|\mathcal{F}_{t-1})]$$

$$= \sum_{t\in[T]}\sum_{n\in[N]}C_3\epsilon\mathbb{E}[Q_n(t)\mathbb{1}(E_t^1)]$$

$$\leq \sum_{t\in[T]}\sum_{n\in[N]}C_3\epsilon\mathbb{E}[Q_n(t)], \tag{44}$$

where the second last inequality comes from $C_2 \geq 17\sqrt{e\pi}$.

For the second term of Eq.(38), we first have

$$\sum_{t\in[T]}\sum_{n\in[N]}\mathbb{E}\left[(\mu(n|S_{k_{n,t}^*,t},\theta_{k_{n,t}^*}) - \mu(n|S_{k_{n,t},t},\theta_{k_{n,t}}))Q_n(t)\mathbb{1}((E_t^1)^c)\right]$$

$$\leq \sum_{t\in[T]}\sum_{n\in[N]}\mathbb{E}\left[Q_n(t)\mathbb{1}((E_t^1)^c)\right]$$

$$\leq \sum_{t\in[T]}\sum_{n\in[N]}t\mathbb{P}((E_t^1)^c)$$

$$= \mathcal{O}\left(\sum_{t\in[T]}\sum_{n\in[N]}t(1/t^2)\right) = \mathcal{O}(N\log(T)), \tag{45}$$

and

$$\sum_{t\in[T]}\sum_{n\in[N]}\mathbb{E}\left[(\mu(n|S_{k_{n,t}^*,t},\theta_{k_{n,t}^*}) - \mu(n|S_{k_{n,t},t},\theta_{k_{n,t}}))Q_n(t)\mathbb{1}((E_t^2)^c)\right]$$

$$\leq \sum_{t\in[T]}\sum_{n\in[N]}\mathbb{E}\left[Q_n(t)\mathbb{1}((E_t^2)^c)\right]$$

$$\leq \sum_{t\in[T]}\sum_{n\in[N]}t\mathbb{P}((E_t^2)^c)$$

$$= \mathcal{O}\left(\sum_{t\in[T]}\sum_{n\in[N]}t(1/t^2)\right) = \mathcal{O}(N\log(T)). \tag{46}$$

Let $\mathcal{T}_n$ be the set of time steps $t \in [T]$ such that $Q_n(t) \neq 0$ and let $e_T = \sum_{n\in[N]}\sum_{t\in\mathcal{T}_n}\mathbb{1}((E_{n,t}^3)^c)$ and $h = \lceil C_4 e_T/\epsilon \rceil$ for some constant $C_4 > 0$. Then if $t \leq h$, we have $Q_n(t) \leq t \leq h$. Otherwise, we have

$$Q_n(t) \leq \sum_{s=t-h+1}^{t}(1/h)(Q_n(s) + (t-s)) \leq (1/h)\sum_{s=1}^{t}Q_n(s) + (1/h)h^2 = (1/h)\sum_{s=1}^{t}Q_n(s) + h.$$

From the above, we have

$$\sum_{t\in[T]}\sum_{n\in[N]}\mathbb{E}[(\mu(n|S_{k_{n,t}^*,t},\theta_{k_{n,t}^*})-\mu(n|S_{k_{n,t},t},\theta_{k_{n,t}}))Q_n(t)\mathbb{1}((E_{n,t}^3)^c)]$$

$$=\sum_{n\in[N]}\sum_{t\in\mathcal{T}_n}\mathbb{E}[(\mu(n|S_{k_{n,t}^*,t},\theta_{k_{n,t}^*})-\mu(n|S_{k_{n,t},t},\theta_{k_{n,t}}))Q_n(t)\mathbb{1}((E_{n,t}^3)^c)]$$

$$\leq\sum_{n\in[N]}\sum_{t\in\mathcal{T}_n}\mathbb{E}[Q_n(t)\mathbb{1}((E_{n,t}^3)^c)]$$

$$\leq\sum_{n\in[N]}\sum_{t\in\mathcal{T}_n}\mathbb{E}[((\epsilon/C_4e_T)\sum_{s=1}^T Q_n(s)+2C_4e_T/\epsilon)\mathbb{1}((E_{n,t}^3)^c)]$$

$$\leq\mathbb{E}\left[(\epsilon/C_4)\sum_{n\in[N]}\sum_{s=1}^T Q_n(s)\right]+\mathbb{E}\left[2C_4e_T^2/\epsilon\right]$$

$$\leq\sum_{t\in[T]}\sum_{n\in[N]}(\epsilon/C_4)\mathbb{E}[Q_n(t)]+2C_4\mathbb{E}[e_T^2]/\epsilon. \tag{47}$$

Now we provide a bound for $\mathbb{E}[e_T^2]$. Define $N_{n,k}(t)=\sum_{s=1}^{t-1}\mathbb{1}(n\in S_{k,s})$ and $\widetilde{V}_{k,t}=(\kappa/2)\sum_{s=1}^{t-1}\sum_{n\in S_{k,s}}x_n x_n^\top$. Then, we have

$$e_T=\sum_{n\in[N]}\sum_{t\in\mathcal{T}_n}\mathbb{1}((E_{n,t}^2)^c)$$

$$\leq\sum_{n\in[N]}\sum_{t\in\mathcal{T}_n}\mathbb{1}(\|x_n\|_{V_{k_{n,t},t}^{-1}}\geq\epsilon/C_2(\gamma_t+\beta_t))$$

$$\leq\sum_{n\in[N]}\sum_{t\in\mathcal{T}_n}\mathbb{1}(\|x_n\|_{\widetilde{V}_{k_{n,t},t}^{-1}}\geq\epsilon/C_2(\gamma_t+\beta_t))$$

$$\leq\sum_{n\in[N]}\sum_{t\in\mathcal{T}_n}\mathbb{1}(1/N_{n,k_{n,t}}(t)\geq(\kappa/2)(\epsilon/C_2(\gamma_t+\beta_t))^2)$$

$$\leq\sum_{n\in[N]}\sum_{t\in\mathcal{T}_n}\mathbb{1}(N_{n,k_{n,t}}(t)\leq(2/\kappa)(C_2(\gamma_t+\beta_t)/\epsilon)^2)$$

$$\leq\sum_{n\in[N]}\sum_{k\in[K]}\sum_{t\in\mathcal{T}_n}\mathbb{1}(N_{n,k}(t)\leq(2/\kappa)(C_2(\gamma_t+\beta_t)/\epsilon)^2\text{ and }k_{n,t}=k)$$

$$\leq NK(2/\kappa)(C_2(\gamma_T+\beta_T)/\epsilon)^2.$$

From the above we have $\mathbb{E}[e_T^2]\leq N^2K^2(4/\kappa^2)(C_2(\gamma_T+\beta_T)/\epsilon)^4$. Then from Eq.(47) we have

$$\sum_{t\in[T]}\sum_{n\in[N]}\mathbb{E}[(\mu(n|S_{k_{n,t}^*,t},\theta_{k_{n,t}^*})-\mu(n|S_{k_{n,t},t},\theta_{k_{n,t}}))Q_n(t)\mathbb{1}((E_{n,t}^2)^c)]$$

$$\leq\sum_{t\in[T]}\sum_{n\in[N]}(\epsilon/C_4)\mathbb{E}[Q_n(t)]+\mathcal{O}(N^2K^2(\beta_T+\gamma_T)^4/\kappa^2\epsilon^5). \tag{48}$$

By putting the results of Eqs. (37), (38), (39), (40), (44), (45), (46), (48) altogether, we can obtain

$$
\mathbb{E}\left[\sum_{t\in[T]}\mathcal{V}(\mathbf{Q}(t+1))-\mathcal{V}(\mathbf{Q}(t))\right]
$$

$$
\leq 7\min\{N,K\}T-2\epsilon\sum_{t\in[T]}\sum_{n\in[N]}\mathbb{E}\left[Q_n(t)\right]+2\sum_{t\in[T]}\sum_{n\in[N]}\mathbb{E}\left[(D_n^*(t)-D_n(t))Q_n(t)\right]
$$

$$
\leq 7\min\{N,K\}T-2\epsilon\sum_{t\in[T]}\sum_{n\in[N]}\mathbb{E}\left[Q_n(t)\right]+2C_3\epsilon\sum_{t\in[T]}\sum_{n\in[N]}\mathbb{E}\left[Q_n(t)\right]
$$

$$
+\mathcal{O}(N\log(T))+(2\epsilon/C_4)\sum_{t\in[T]}\sum_{n\in[N]}\mathbb{E}\left[Q_n(t)\right]+\mathcal{O}(N^2K^2(\beta_T+\gamma_T)^4/\epsilon^5)
$$

$$
\leq 7\min\{N,K\}T+2(C_3+(1/C_4)-1)\epsilon\sum_{t\in[T]}\sum_{n\in[N]}\mathbb{E}\left[Q_n(t)\right]+\mathcal{O}(N^2K^2(\beta_T+\gamma_T)^4/\epsilon^5).
$$

Finally, with positive constants $C_3, C_4 > 0$ satisfying $C_3 + (1/C_4) < 1$, from $\mathcal{V}(\mathbf{Q}(1)) = 0$ and $\mathcal{V}(\mathbf{Q}(T+1)) \geq 0$, by using telescoping for the above inequality, we can conclude the proof by

$$
\frac{1}{T}\sum_{t\in[T]}\sum_{n\in[N]}\mathbb{E}[Q_n(t)]=\mathcal{O}\left(\frac{\min\{N,K\}}{\epsilon}+\frac{1}{T}\frac{d^4N^2K^2}{\kappa^4\epsilon^6}\text{polylog}(T)\right). \tag{49}
$$

### A.7  Proof of Theorem 4

We first provide the proof for the regret bound of $\widetilde{\mathcal{O}}\left(\frac{d^{3/2}}{\kappa}\sqrt{KT}Q_{\max}\right)$. We define event $E_t^1 = \{\|\hat{\theta}_{k,t}-\theta_k\|_{V_{k,t}}\leq\beta_t;\forall k\in[K]\}$ which holds at least probability of $1-1/t^2$ from Lemma 1. We let $\gamma_t=\beta_t\sqrt{d\log(Mt)}$. Then we also define $E_t^2=\{|h_{n,k,t}^{TS}-x_n^\top\hat{\theta}_{k,t}|\leq\gamma_t\|x_n\|_{V_{k,t}^{-1}};\forall n\in[N],\forall k\in[K]\}$, which holds at least probability of $1-1/t^2$ from Lemma 5.

Then we have

$$
\mathcal{R}^\pi(T)=\sum_{t\in[T]}\sum_{n\in[N]}\mathbb{E}[Q_n(t)(\mu(n|S_{k_{n,t}^*,t},\theta_{k_{n,t}^*})-\mu(n|S_{k_{n,t},t},\theta_{k_{n,t}}))]
$$

$$
=\sum_{t\in[T]}\sum_{n\in[N]}\mathbb{E}[Q_n(t)(\mu(n|S_{k_{n,t}^*,t},\theta_{k_{n,t}^*})-\widetilde{\mu}_t^{TS}(n|S_{k_{n,t},t})+\widetilde{\mu}_t^{TS}(n|S_{k_{n,t},t})-\mu(n|S_{k_{n,t},t},\theta_{k_{n,t}}))]
$$

$$
\leq\sum_{t\in[T]}\sum_{n\in[N]}\mathbb{E}[Q_n(t)(\mu(n|S_{k_{n,t}^*,t},\theta_{k_{n,t}^*})-\widetilde{\mu}_t^{TS}(n|S_{k_{n,t},t})+\widetilde{\mu}_t^{TS}(n|S_{k_{n,t},t})-\mu(n|S_{k_{n,t},t},\theta_{k_{n,t}}))\mathbb{1}(E_t^1\cap E_t^2)]
$$

$$
+\sum_{t\in[T]}\sum_{n\in[N]}\mathbb{E}[Q_n(t)(\mu(n|S_{k_{n,t}^*,t},\theta_{k_{n,t}^*})-\widetilde{\mu}_t^{TS}(n|S_{k_{n,t},t})+\widetilde{\mu}_t^{TS}(n|S_{k_{n,t},t})-\mu(n|S_{k_{n,t},t},\theta_{k_{n,t}}))\mathbb{1}((E_t^1)^c)]
$$

$$
+\sum_{t\in[T]}\sum_{n\in[N]}\mathbb{E}[Q_n(t)(\mu(n|S_{k_{n,t}^*,t},\theta_{k_{n,t}^*})-\widetilde{\mu}_t^{TS}(n|S_{k_{n,t},t})+\widetilde{\mu}_t^{TS}(n|S_{k_{n,t},t})-\mu(n|S_{k_{n,t},t},\theta_{k_{n,t}}))\mathbb{1}((E_t^2)^c)].
$$

$$\tag{50}$$

**Lemma 8.** *Under $E_t^2$, for all $k\in[K]$, we have*

$$
\left|\sum_{n\in S_{k,t}}(\widetilde{\mu}_t^{TS}(n|S_{k,t})-\mu(n|S_{k,t},\hat{\theta}_{k,t}))Q_n(t)\right|\leq\gamma_t\max_{n\in S_{k,t}}\|x_n\|_{V_{k,t}^{-1}}Q_n(t). \tag{51}
$$

*Proof.* Let $u_{n,k,t}=x_n^\top\hat{\theta}_{k,t}$. Under $E_t^2$, for any $k\in[K]$ we have $|h_{n,k,t}^{TS}-u_{n,k,t}|\leq\gamma_t\|x_n\|_{V_{k,t}^{-1}}$. Then by the mean value theorem, there exists $\bar{u}_{n,k,t}=(1-c)h_{n,k,t}^{TS}+cu_{n,k,t}$ for some $c\in(0,1)$

satisfying,

$$
\left| \sum_{n \in S_{k,t}} (\widetilde{\mu}_t^{TS}(n|S_k) - \mu(n|S_k, \theta_k))Q_n(t) \right|
$$

$$
= \left| \sum_{n \in S_{k,t}} \left( \frac{\exp(h_{n,k,t}^{TS})}{1 + \sum_{m \in S_k} \exp(h_{m,k,t}^{TS})} - \frac{\exp(u_{n,k,t})}{1 + \sum_{m \in S_k} \exp(u_{m,k,t})} \right) Q_n(t) \right|
$$

$$
= \left| \sum_{n \in S_{k,t}} \nabla_{v_n} \left( \frac{\sum_{m \in S_{k,t}} \exp(v_m)}{1 + \sum_{m \in S_k} \exp(v_m)} \right) \Big|_{v_n = \bar{u}_{n,k,t}} (h_{n,k,t}^{TS} - u_{n,k,t})Q_n(t) \right|
$$

$$
= \left| \frac{(1 + \sum_{n \in S_k} \exp(\bar{u}_{n,k,t}))(\sum_{n \in S_k} \exp(\bar{u}_{n,k,t})(h_{n,k,t}^{TS} - u_{n,k,t})Q_n(t))}{(1 + \sum_{n \in S_k} \exp(\bar{u}_{n,k,t}))^2} \right.
$$

$$
\left. - \frac{(\sum_{n \in S_{k,t}} \exp(\bar{u}_{n,k,t}))(\sum_{n \in S_{k,t}} \exp(\bar{u}_{n,k,t})(h_{n,k,t}^{TS} - u_{n,k,t})Q_n(t))}{(1 + \sum_{n \in S_k} \exp(\bar{u}_{n,k,t}))^2} \right|
$$

$$
\leq \sum_{n \in S_{k,t}} \frac{\exp(\bar{u}_{n,k,t})}{1 + \sum_{m \in S_{k,t}} \exp(\bar{u}_{m,k,t})} |h_{n,k,t}^{TS} - u_{n,k,t}|Q_n(t)
$$

$$
\leq \max_{n \in S_{k,t}} |h_{n,k,t}^{TS} - u_{n,k,t}|Q_n(t) \leq \gamma_t \max_{n \in S_{k,t}} \|x_n\|_{V_{k,t}^{-1}}Q_n(t).
$$

$\square$

**Lemma 9.** *Under $E_t^1$, for all $k \in [K]$ we have*

$$
\sum_{n \in S_{k,t}} (\mu(n|S_{k,t}, \hat{\theta}_{k,t}) - \mu(n|S_{k,t}, \theta_k))Q_n(t) \leq 2\beta_t \max_{n \in S_{k,t}} \|x_n\|_{V_{k,t}^{-1}}Q_n(t).
$$

*Proof.* Since $x/(1 + x)$ is a non-decreasing function for $x > -1$ and $x_n^\top \hat{\theta}_{k_{n,t},t} \leq x_n^\top \hat{\theta}_{k_{n,t},t} + \beta_t \|x_n\|_{V_{k_{n,t},t}^{-1}}$ under $E_t^1$, with the definition of $\widetilde{\mu}_t^{UCB}(n|S_{k,t})$, we have

$$
\sum_{n \in S_{k,t}} (\mu(n|S_{k,t}, \hat{\theta}_{k,t}) - \mu(n|S_{k,t}, \theta_k))Q_n(t) \leq \sum_{n \in S_{k,t}} (\widetilde{\mu}_t^{UCB}(n|S_{k,t}) - \mu(n|S_{k,t}, \theta_k))Q_n(t).
$$

Then let $u_{n,k,t} = x_n^\top \theta_k$. Under $E_t^1$, for any $n \in [N]$ and $k \in [K]$ we have $x_n^\top \hat{\theta}_{k,t} - \beta_t \|x_n\|_{V_{k,t}^{-1}} \leq x_n^\top \theta_k \leq x_n^\top \hat{\theta}_{k,t} + \beta_t \|x_n\|_{V_{k,t}^{-1}}$, which implies $0 \leq h_{n,k,t}^{UCB} - u_{n,k,t} \leq 2\beta_t \|x_n\|_{V_{k,t}^{-1}}$. Then by the

mean value theorem, there exists $\bar{u}_{n,k,t} = (1-c)h_{n,k,t}^{UCB} + cu_{n,k,t}$ for some $c \in (0,1)$ satisfying,

$$\sum_{n \in S_{k,t}} (\widetilde{\mu}_t^{UCB}(n|S_{k,t}, \hat{\theta}_{k,t}) - \mu(n|S_{k,t}, \theta_k))Q_n(t)$$

$$= \sum_{n \in S_k} \left( \frac{\exp(h_{n,k,t}^{UCB})}{1 + \sum_{m \in S_{k,t}} \exp(h_{m,k,t}^{UCB})} - \frac{\exp(u_{n,k,t})}{1 + \sum_{m \in S_{k,t}} \exp(u_{m,k,t})} \right) Q_n(t)$$

$$= \sum_{n \in S_{k,t}} \nabla_{v_n} \left( \frac{\exp(v_n)}{1 + \sum_{m \in S_{k,t}} \exp(v_m)} \right) \Big|_{v_n = \bar{u}_{n,k,t}} (h_{n,k,t}^{UCB} - u_{n,k,t})Q_n(t)$$

$$= \frac{(1 + \sum_{n \in S_k} \exp(\bar{u}_{n,k,t}))(\sum_{n \in S_k} \exp(\bar{u}_{n,k,t})(h_{n,k,t}^{UCB} - u_{n,k,t})Q_n(t))}{(1 + \sum_{n \in S_k} \exp(\bar{u}_{n,k,t}))^2}$$

$$- \frac{(\sum_{n \in S_{k,t}} \exp(\bar{u}_{n,k,t}))(\sum_{n \in S_{k,t}} \exp(\bar{u}_{n,k,t})(h_{n,k,t}^{UCB} - u_{n,k,t})Q_n(t))}{(1 + \sum_{n \in S_{k,t}} \exp(\bar{u}_{n,k,t}))^2}$$

$$\leq \sum_{n \in S_{k,t}} \frac{\exp(\bar{u}_{n,k,t})}{1 + \sum_{m \in S_{k,t}} \exp(\bar{u}_{m,k,t})} (h_{n,k,t}^{UCB} - u_{n,k,t})Q_n(t)$$

$$\leq \max_{n \in S_k} (h_{n,k,t}^{UCB} - u_{n,k,t})Q_n(t)$$

$$\leq 2\beta_t \max_{n \in S_{k,t}} \|x_n\|_{V_{k,t}^{-1}} Q_n(t).$$

$\square$

Then we first focus on the first term in the above. From Lemmas 8 and 9, we have

$$\sum_{t \in [T]} \sum_{n \in [N]} \mathbb{E}[\widetilde{\mu}_t^{TS}(n|S_{k_{n,t},t}) - \mu(n|S_{k_{n,t},t}, \theta_{k_{n,t}}))Q_n(t)\mathbb{1}(E_t^1 \cap E_t^2)]$$

$$\leq \sum_{t \in [T]} \sum_{n \in [N]} \mathbb{E}[(\widetilde{\mu}_t^{TS}(n|S_{k_{n,t},t}) - \mu(n|S_{k_{n,t},t}, \hat{\theta}_{k_{n,t},t})$$

$$+ \mu(n|S_{k_{n,t},t}, \hat{\theta}_{k_{n,t},t}) - \mu(n|S_{k_{n,t},t}, \theta_{k_{n,t}}))Q_n(t)\mathbb{1}(E_t^1 \cap E_t^2)]$$

$$\leq \sum_{t \in [T]} \mathbb{E}[\sum_{k \in [K]} \sum_{n \in S_{k,t}} (\widetilde{\mu}_t^{TS}(n|S_{k,t}) - \mu(n|S_{k,t}, \hat{\theta}_{k,t})$$

$$+ \mu(n|S_{k,t}, \hat{\theta}_{k,t}) - \mu(n|S_{k,t}, \theta_k))Q_n(t)\mathbb{1}(E_t^1 \cap E_t^2)]$$

$$\leq \sum_{t \in [T]} \sum_{k \in [K]} \mathbb{E}[(\gamma_t + 2\beta_t) \max_{n \in S_{k,t}} \|x_n\|_{V_{k,t}^{-1}} Q_n(t)\mathbb{1}(E_t^1)]$$

$$= \sum_{t \in [T]} \sum_{k \in [K]} \mathbb{E}[\mathbb{E}[(\gamma_t + 2\beta_t) \max_{n \in S_{k,t}} \|x_n\|_{V_{k,t}^{-1}} Q_n(t)|E_t^1, \mathcal{F}_{t-1}]\mathbb{P}(E_t^1|\mathcal{F}_{t-1})]. \qquad (52)$$

We define sets

$$\widetilde{\Theta}_t = \left\{ \{\theta_k^{(i)}\}_{i \in [M], k \in [K]} : \left| \sum_{n \in S_{k,t}} (\widetilde{\mu}_t^{TS}(n|S_{k,t}, \{\theta_k^{(i)}\}_{i \in [M]}) - \mu(n|S_{k,t}, \hat{\theta}_{k,t}))Q_n(t) \right| \right.$$

$$\left. \leq \gamma_t \max_{n \in S_{k,t}} \|x_n\|_{V_{k,t}^{-1}} Q_n(t); \ \forall k \in [K] \right\}$$

and

$$\widetilde{\Theta}_t^{opt} = \left\{ \{\theta_k^{(i)}\}_{i \in [M], k \in [K]} : \sum_{k \in [K]} \sum_{n \in S_{k,t}} \widetilde{\mu}_t^{TS}(n|S_{k,t}, \{\theta_k^{(i)}\}_{i \in [M]})Q_n(t) \right.$$

$$\left. > \sum_{k \in [K]} \sum_{n \in S_{k,t}^*} \mu(n|S_{k,t}^*, \theta_k)Q_n(t) \right\} \cap \widetilde{\Theta}_t.$$

We define event $E_t(\widetilde{\theta}) = \{\{\widetilde{\theta}_{k,t}^{(i)}\}_{i\in[M],k\in[K]} \in \widetilde{\Theta}_t^{opt}\}$. Then by following the proof steps in Eq.(41), we have

$$\mathbb{E}\left[\mathbb{E}\left[\left(\sum_{k\in[K]}\sum_{n\in S_{k,t}^*}\mu(n|S_{k,t}^*,\theta_k)Q_n(t) - \sum_{k\in[K]}\sum_{n\in S_{k,t}}\widetilde{\mu}_t^{TS}(n|S_{k,t},\{\widetilde{\theta}_{k,t}^{(i)}\}_{i\in[M]})Q_n(t)\right)\mathbb{1}(E_t^1\cap E_t^2)|\mathcal{F}_{t-1}\right]\right]$$

$$\leq \mathbb{E}\left[\mathbb{E}\left[\left(\sum_{k\in[K]}\sum_{n\in S_{k,t}^*}\mu(n|S_{k,t}^*,\theta_k)Q_n(t)\right.\right.\right.$$

$$\left.\left.\left. - \inf_{\{\theta_l^{(i)}\}_{i\in[M],l\in[K]}\in\widetilde{\Theta}_t}\sum_{k\in[K]}\sum_{n\in S_{k,t}}\widetilde{\mu}_t^{TS}(n|S_{k,t},\{\theta_k^{(i)}\}_{i\in[M]})Q_n(t)\right)\mathbb{1}(E_t^1\cap E_t^2)|\mathcal{F}_{t-1},E_t(\widetilde{\theta})\right]\right]$$

$$\leq \mathbb{E}\left[\mathbb{E}\left[\sup_{\{\theta_l^{(i)}\}_{i\in[M],l\in[K]}\in\widetilde{\Theta}_t}\sum_{k\in[K]}\sum_{n\in S_{k,t}}\left(\widetilde{\mu}_t^{TS}(n|S_{k,t},\{\widetilde{\theta}_{k,t}^{(i)}\}_{i\in[M]})\right.\right.\right.$$

$$\left.\left.\left. - \widetilde{\mu}_t^{TS}(n|S_{k,t},\{\theta_k^{(i)}\}_{i\in[M]})\right)Q_n(t)\mathbb{1}(E_t^1\cap E_t^2)|\mathcal{F}_{t-1},E_t(\widetilde{\theta})\right]\right]$$

$$\leq \mathbb{E}\left[\mathbb{E}\left[\sup_{\{\theta_l^{(i)}\}_{i\in[M],l\in[K]}\in\widetilde{\Theta}_t}\sum_{k\in[K]}\sum_{n\in S_{k,t}}\left(\widetilde{\mu}_t^{TS}(n|S_{k,t},\{\widetilde{\theta}_{k,t}^{(i)}\}_{i\in[M]}) - \mu(n|S_{k,t},\hat{\theta}_{k,t})\right.\right.\right.$$

$$\left.\left.\left. + \mu(n|S_{k,t},\hat{\theta}_{k,t}) - \widetilde{\mu}_t^{TS}(n|S_{k,t},\{\theta_k^{(i)}\}_{i\in[M]})\right)Q_n(t)\mathbb{1}(E_t^1\cap E_t^2)|\mathcal{F}_{t-1},E_t(\widetilde{\theta})\right]\right]$$

$$\leq 2\gamma_t\mathbb{E}\left[\sum_{k\in[K]}\mathbb{E}\left[\max_{n\in S_{k,t}}\|x_n\|_{V_{k,t}^{-1}}Q_n(t)\mathbb{1}(E_t^1)\big|\mathcal{F}_{t-1},E_t(\widetilde{\theta})\right]\right]$$

$$= 2\gamma_t\mathbb{E}\left[\sum_{k\in[K]}\mathbb{E}\left[\max_{n\in S_{k,t}}\|x_n\|_{V_{k,t}^{-1}}Q_n(t)\big|\mathcal{F}_{t-1},E_t(\widetilde{\theta}),E_t^1\right]\times\mathbb{P}(E_t^1|E_t(\widetilde{\theta}),\mathcal{F}_{t-1})\right]$$

$$= 2\gamma_t\mathbb{E}\left[\sum_{k\in[K]}\mathbb{E}\left[\max_{n\in S_{k,t}}\|x_n\|_{V_{k,t}^{-1}}Q_n(t)\big|\mathcal{F}_{t-1},E_t(\widetilde{\theta}),E_t^1\right]\mathbb{P}(E_t^1|\mathcal{F}_{t-1})\right]$$

$$\leq 32\sqrt{e\pi}\gamma_t\mathbb{E}\left[\mathbb{E}\left[\sum_{k\in[K]}\max_{n\in S_{k,t}}\|x_n\|_{V_{k,t}^{-1}}Q_n(t)|\mathcal{F}_{t-1},E_t^1\right]\mathbb{P}(E_t^1|\mathcal{F}_{t-1})\right],$$

$$\tag{53}$$

where the last inequality is obtained from (42). Then from Eqs.(52) and (53), for some constant $C_2 > 0$, we have

$$\sum_{t\in[T]}\sum_{n\in[N]}\mathbb{E}[Q_n(t)(\mu(n|S_{k_{n,t}^*,t},\theta_{k_{n,t}^*}) - \widetilde{\mu}_t^{TS}(n|S_{k_{n,t},t}) + \widetilde{\mu}_t^{TS}(n|S_{k_{n,t},t}) - \mu(n|S_{k_{n,t},t},\theta_{k_{n,t}}))\mathbb{1}(E_t^1\cap E_t^2)]$$

$$\leq \sum_{t\in[T]}\sum_{k\in[K]}\mathbb{E}[\mathbb{E}[\max_{n\in S_{k,t}}(33\sqrt{e\pi}\gamma_t + 2\beta_t)\|x_n\|_{V_{k,t}^{-1}}Q_n(t)|E_t^1,\mathcal{F}_{t-1}]\mathbb{P}(E_t^1|\mathcal{F}_{t-1})]$$

$$\leq C_2\sum_{t\in[T]}\sum_{k\in[K]}\mathbb{E}\left[\max_{n\in S_{k,t}}(\gamma_t + \beta_t)\|x_n\|_{V_{k,t}^{-1}}Q_n(t)\mathbb{1}(E_t^1)\right]$$

$$\leq C_2\sum_{t\in[T]}\sum_{k\in[K]}\mathbb{E}\left[\max_{n\in S_{k,t}}(\gamma_t + \beta_t)\|x_n\|_{V_{k,t}^{-1}}Q_n(t)\right].$$

$$\tag{54}$$

For the second and third terms of Eq.(50), we have

$$\sum_{t\in[T]}\sum_{n\in[N]}\mathbb{E}\left[(\mu(n|S_{k_{n,t}^*,t},\theta_{k_{n,t}^*})-\mu(n|S_{k_{n,t},t},\theta_{k_{n,t}}))Q_n(t)\mathbb{1}((E_t^1)^c)\right]$$

$$\leq \sum_{t\in[T]}\sum_{n\in[N]}\mathbb{E}\left[Q_n(t)\mathbb{1}((E_t^1)^c)\right]$$

$$\leq \sum_{t\in[T]}\sum_{n\in[N]}t\mathbb{P}((E_t^1)^c)$$

$$=\mathcal{O}\left(\sum_{t\in[T]}\sum_{n\in[N]}t(1/t^2)\right)=\mathcal{O}(N\log(T)), \tag{55}$$

and

$$\sum_{t\in[T]}\sum_{n\in[N]}\mathbb{E}\left[(\mu(n|S_{k_{n,t}^*,t},\theta_{k_{n,t}^*})-\mu(n|S_{k_{n,t},t},\theta_{k_{n,t}}))Q_n(t)\mathbb{1}((E_t^2)^c)\right]$$

$$\leq \sum_{t\in[T]}\sum_{n\in[N]}\mathbb{E}\left[Q_n(t)\mathbb{1}((E_t^2)^c)\right]$$

$$\leq \sum_{t\in[T]}\sum_{n\in[N]}t\mathbb{P}((E_t^2)^c)$$

$$=\mathcal{O}\left(\sum_{t\in[T]}\sum_{n\in[N]}t(1/t^2)\right)=\mathcal{O}(N\log(T)). \tag{56}$$

Finally from Eqs.(50), (54), (55), and (56), we can conclude that

$$\mathcal{R}^\pi(T)=\mathcal{O}\left(\sum_{t\in[T]}\sum_{k\in[K]}\mathbb{E}[\max_{n\in S_{k,t}}(\gamma_t+\beta_t)\|x_n\|_{V_{k,t}^{-1}}Q_n(t)]+N\log(T)\right)$$

$$=\mathcal{O}\left(\gamma_T\mathbb{E}[\max_{t\in[T],n\in[N]}Q_n(t)\sum_{t\in[T]}\sum_{k\in[K]}\max_{n\in S_{k,t}}\|x_n\|_{V_{k,t}^{-1}}]+N\log(T)\right)$$

$$=\mathcal{O}\left(\gamma_T\mathbb{E}[\max_{t\in[T],n\in[N]}Q_n(t)\sqrt{KT\sum_{t\in[T]}\sum_{k\in[K]}\max_{n\in S_{k,t}}\|x_n\|_{V_{k,t}^{-1}}^2}]+N\log(T)\right)$$

$$=\widetilde{\mathcal{O}}\left(\frac{d^{3/2}}{\kappa}\sqrt{KT}Q_{\max}\right),$$

where the last equality is obtained from Lemma 4.

Here, we provide the proof for the worst-case regret bound of $\widetilde{\mathcal{O}}\left(\left(\frac{d^2NK\min\{N,K\}^3}{\kappa^2\epsilon^3}\right)^{1/4}T^{3/4}\right)$.

We define event $E_t^1 = \{\|\hat{\theta}_{k,t}-\theta_k\|_{V_{k,t}}\leq\beta_t$ for all $k\in[K]\}$ where $\beta_t = C_1\sqrt{\lambda+\frac{d}{\kappa}\log(1+tLK/d\lambda)}$, which holds with high probability as $\mathbb{P}(E_t^1)\geq 1-1/t^2$ from Lemma 1. We also define $E_t^2 = \{\widetilde{\mu}_t^{TS}(n|S_{k,t})-\mu(n|S_{k,t},\hat{\theta}_{k,t})\leq\gamma_t\|x_n\|_{V_{k,t}^{-1}};\forall n\in[N],\forall k\in[K]\}$, which holds with high probability as $\mathbb{P}(E_t^2)\geq 1-\mathcal{O}(1/t^2)$, and define $E_{n,t}^3 = \{\|x_n\|_{V_{k,t}^{-1}}\leq\zeta;\forall k\in[K]\}$.

Then we have

$$
\mathcal{R}^\pi(T)
$$

$$
= \sum_{t\in[T]}\sum_{n\in[N]} \mathbb{E}[Q_n(t)(\mu(n|S_{k^*_{n,t},t},\theta_{k^*_{n,t}}) - \mu(n|S_{k_{n,t},t},\theta_{k_{n,t}}))]
$$

$$
\leq \sum_{t\in[T]}\sum_{n\in[N]} \mathbb{E}[(\mu(n|S_{k^*_{n,t},t},\theta_{k^*_{n,t}}) - \mu(n|S_{k_{n,t},t},\theta_{k_{n,t}}))Q_n(t)\mathbb{1}(E^1_t \cap E^2_t \cap E^3_{n,t})]
$$

$$
+ \sum_{t\in[T]}\sum_{n\in[N]} \mathbb{E}[(\mu(n|S_{k^*_{n,t},t},\theta_{k^*_{n,t}}) - \mu(n|S_{k_{n,t},t},\theta_{k_{n,t}}))Q_n(t)(\mathbb{1}((E^1_t)^c) + \mathbb{1}((E^2_t)^c) + \mathbb{1}((E^3_{n,t})^c))].
$$

$$
(57)
$$

We provide a bound for the first term of Eq.(57). We first have

$$
\sum_{t\in[T]}\sum_{n\in[N]} \mathbb{E}[(\mu(n|S_{k^*_{n,t},t},\theta_{k^*_{n,t}}) - \mu(n|S_{k_{n,t},t},\theta_{k_{n,t}}))Q_n(t)\mathbb{1}(E^1_t \cap E^2_t \cap E^3_{n,t})]
$$

$$
\leq \sum_{t\in[T]}\sum_{n\in[N]} \mathbb{E}[(\mu(n|S_{k^*_{n,t},t},\theta_{k^*_{n,t}}) - \widetilde{\mu}^{TS}_t(n|S_{k_{n,t},t})
$$

$$
+ \widetilde{\mu}^{TS}_t(n|S_{k_{n,t},t}) - \mu(n|S_{k_{n,t},t},\theta_{k_{n,t}}))Q_n(t)\mathbb{1}(E^1_t \cap E^2_t \cap E^3_{n,t})].
$$

$$
(58)
$$

By following the steps in Eq.(40), the last two terms in Eq.(58) are bounded as

$$
\sum_{t\in[T]}\sum_{n\in[N]} \mathbb{E}[\widetilde{\mu}^{TS}_t(n|S_{k_{n,t},t}) - \mu(n|S_{k_{n,t},t},\theta_{k_{n,t}}))Q_n(t)\mathbb{1}(E^1_t \cap E^2_t \cap E^3_{n,t})]
$$

$$
\leq \sum_{t\in[T]}\sum_{k\in[K]} \mathbb{E}\left[ \mathbb{E}\left[ \sum_{n\in S_{k,t}} (\gamma_t + 2\beta_t)\zeta Q_n(t)|E^1_t, \mathcal{F}_{t-1} \right] \mathbb{P}(E^1_t|\mathcal{F}_{t-1}) \right].
\qquad (59)
$$

Now we provide a bound for the first two terms in Eq.(58).

We define sets

$$
\widetilde{\Theta}_t = \left\{ \{\theta^{(i)}_k\}_{i\in[M],k\in[K]} : \left|\max_{i\in[M]} x_n^\top\theta^{(i)}_k - x_n^\top\hat{\theta}_{k,t}\right| \leq \gamma_t\|x_n\|_{V^{-1}_{k,t}}; \ \forall n\in[N], \forall k\in[K] \right\} \text{ and}
$$

$$
\widetilde{\Theta}^{opt}_t = \left\{ \{\theta^{(i)}_k\}_{i\in[M],k\in[K]} : \sum_{k\in[K]}\sum_{n\in S_{k,t}} \widetilde{\mu}^{TS}_t(n|S_{k,t},\{\theta^{(i)}_k\}_{i\in[M]})Q_n(t) > \sum_{k\in[K]}\sum_{n\in S^*_{k,t}} \mu(n|S^*_{k,t},\theta_k)Q_n(t) \right\}\cap\widetilde{\Theta}_t.
$$

By following the steps in Eq.(41) with Eq. (42), we have

$$
\mathbb{E}[\mathbb{E}[(\sum_{k\in[K]}\sum_{n\in S^*_{k,t}} \mu(n|S^*_{k,t},\theta_k) - \sum_{k\in[K]}\sum_{n\in S_{k,t}} \widetilde{\mu}^{TS}_t(n|S_{k,t},\{\widetilde{\theta}^{(i)}_{k,t}\}_{i\in[M]}))Q_n(t)\mathbb{1}(E^1_t \cap E^2_t \cap E^3_{n,t})|\mathcal{F}_{t-1}]]
$$

$$
= 2\gamma_t\zeta\mathbb{E}\left[ \sum_{k\in[K]}\sum_{n\in S_{k,t}} \mathbb{E}\left[ Q_n(t)|\mathcal{F}_{t-1},\{\widetilde{\theta}^{(i)}_{k,t}\}_{i\in[M],k\in[K]} \in \widetilde{\Theta}^{opt}_t, E^1_t \right] \mathbb{P}(E^1_t|\mathcal{F}_{t-1}) \right]
$$

$$
\leq 16\sqrt{e\pi}\gamma_t\zeta\mathbb{E}\left[ \sum_{k\in[K]}\sum_{n\in S_{k,t}} Q_n(t)|\mathcal{F}_{t-1}, E^1_t \right] \mathbb{P}(E^1_t|\mathcal{F}_{t-1}).
$$

$$
(60)
$$

Then for the first term of Eq.(57), from Eqs.(58), (59), for some $C_3 > 0$ we have

$$\sum_{t \in [T]} \sum_{n \in [N]} \mathbb{E}[(\mu(n|S_{k^*_{n,t},t}, \theta_{k^*_{n,t}}) - \mu(n|S_{k_{n,t},t}, \theta_{k_{n,t}}))Q_n(t)\mathbb{1}(E^1_t \cap E^2_t \cap E^3_{n,t})]$$

$$\leq \sum_{t \in [T]} \sum_{n \in [N]} \mathbb{E}[(\mu(n|S_{k^*_{n,t},t}, \theta_{k^*_{n,t}}) - \widetilde{\mu}^{TS}_t(n|S_{k_{n,t},t})$$

$$+ \widetilde{\mu}^{TS}_t(n|S_{k_{n,t},t}) - \mu(n|S_{k_{n,t},t}, \theta_{k_{n,t}}))Q_n(t)\mathbb{1}(E^1_t \cap E^2_t \cap E^3_{n,t})]$$

$$\leq \sum_{t \in [T]} \sum_{k \in [K]} \mathbb{E}\left[\mathbb{E}\left[\sum_{n \in S_{k,t}} (17\sqrt{e\pi}\gamma_t + 2\beta_t)\zeta Q_n(t)|\mathcal{F}_{t-1}, E^1_t\right]\mathbb{P}(E^1_t|\mathcal{F}_{t-1})\right]$$

$$\leq \sum_{t \in [T]} \sum_{n \in [N]} (17\sqrt{e\pi}\gamma_t + 2\beta_t)\zeta\mathbb{E}[\mathbb{E}[Q_n(t)|E^1_t, \mathcal{F}_{t-1}]\mathbb{P}(E^1_t|\mathcal{F}_{t-1})]$$

$$= \sum_{t \in [T]} \sum_{n \in [N]} (17\sqrt{e\pi}\gamma_t + 2\beta_t)\zeta\mathbb{E}[Q_n(t)\mathbb{1}(E^1_t)]$$

$$\leq (17\sqrt{e\pi}\gamma_T + 2\beta_T)\zeta \sum_{t \in [T]} \sum_{n \in [N]} \mathbb{E}[Q_n(t)]. \tag{61}$$

For the second term of Eq.(57), by following the steps in Eqs.(45), (46) we have

$$\sum_{t \in [T]} \sum_{n \in [N]} \mathbb{E}\left[(\mu(n|S_{k^*_{n,t},t}, \theta_{k^*_{n,t}}) - \mu(n|S_{k_{n,t},t}, \theta_{k_{n,t}}))Q_n(t)\mathbb{1}((E^1_t)^c)\right] = \mathcal{O}(N\log(T)), \tag{62}$$

and

$$\sum_{t \in [T]} \sum_{n \in [N]} \mathbb{E}\left[(\mu(n|S_{k^*_{n,t},t}, \theta_{k^*_{n,t}}) - \mu(n|S_{k_{n,t},t}, \theta_{k_{n,t}}))Q_n(t)\mathbb{1}((E^2_t)^c)\right] = \mathcal{O}(N\log(T)). \tag{63}$$

Let $\mathcal{T}_n$ be the set of time steps $t \in [T]$ such that $Q_n(t) \neq 0$ and let $e_T = \sum_{n \in [N]} \sum_{t \in \mathcal{T}_n} \mathbb{1}((E^3_{n,t})^c)$ and $h = \lceil 1/(17\sqrt{e\pi}\gamma_T + 2\beta_T)\zeta \rceil$. Then if $t \leq h$, we have $Q_n(t) \leq t \leq h$. Otherwise, we have

$$Q_n(t) \leq \sum_{s=t-h+1}^{t} (1/h)(Q_n(s) + (t-s)) \leq (1/h)\sum_{s=1}^{t} Q_n(s) + (1/h)h^2 = (1/h)\sum_{s=1}^{t} Q_n(s) + h.$$

Then, by following the steps in Eq.(47), we have

$$\sum_{t \in [T]} \sum_{n \in [N]} \mathbb{E}[(\mu(n|S_{k^*_{n,t},t}, \theta_{k^*_{n,t}}) - \mu(n|S_{k_{n,t},t}, \theta_{k_{n,t}}))Q_n(t)\mathbb{1}((E^2_{n,t})^c)]$$

$$\leq \sum_{t \in [T]} \sum_{n \in [N]} (17\sqrt{e\pi}\gamma_T + 2\beta_T)\zeta\mathbb{E}[Q_n(t)] + 2\mathbb{E}[e_T]/(17\sqrt{e\pi}\gamma_T + 2\beta_T)\zeta. \tag{64}$$

Now we provide a bound for $\mathbb{E}[e_T]$. Define $N_{n,k}(t) = \sum_{s=1}^{t-1} \mathbb{1}(n \in S_{k,s})$ and $\widetilde{V}_{k,t} = (\kappa/2) \sum_{s=1}^{t-1} \sum_{n \in S_{k,s}} x_n x_n^\top$. Then, we have

$$
\begin{aligned}
e_T &= \sum_{n \in [N]} \sum_{t \in \mathcal{T}_n} \mathbb{1}((E_{n,t}^2)^c) \\
&\leq \sum_{n \in [N]} \sum_{t \in \mathcal{T}_n} \mathbb{1}(\|x_n\|_{V_{k_{n,t},t}^{-1}} \geq \zeta) \\
&\leq \sum_{n \in [N]} \sum_{t \in \mathcal{T}_n} \mathbb{1}(\|x_n\|_{\widetilde{V}_{k_{n,t},t}^{-1}} \geq \zeta) \\
&\leq \sum_{n \in [N]} \sum_{t \in \mathcal{T}_n} \mathbb{1}(1/N_{n,k_{n,t}}(t) \geq (\kappa/2)\zeta^2) \\
&\leq \sum_{n \in [N]} \sum_{t \in \mathcal{T}_n} \mathbb{1}(N_{n,k_{n,t}}(t) \leq 2/\kappa\zeta^2) \\
&\leq \sum_{n \in [N]} \sum_{k \in [K]} \sum_{t \in \mathcal{T}_n} \mathbb{1}(N_{n,k}(t) \leq 2/\kappa\zeta^2 \text{ and } k_{n,t} = k) \\
&\leq 2NK/\kappa\zeta^2.
\end{aligned}
\tag{65}
$$

Then from Eqs.(64), (65), we have

$$
\sum_{t \in [T]} \sum_{n \in [N]} \mathbb{E}[(\mu(n|S_{k_{n,t}^*,t}, \theta_{k_{n,t}^*}) - \mu(n|S_{k_{n,t},t}, \theta_{k_{n,t}}))Q_n(t)\mathbb{1}((E_{n,t}^2)^c)]
$$

$$
\leq \mathcal{O}\left( \sum_{t \in [T]} \sum_{n \in [N]} \zeta(\beta_T + \gamma_T)\mathbb{E}\left[Q_n(t)\right] + \frac{NK}{\kappa\zeta^3(\beta_T + \gamma_T)} \right).
\tag{66}
$$

By putting the results of Eqs. (57), (58), (59), (61), (62), (63), (66), and Theorem 3, by setting $\zeta = (\epsilon NK/\min\{N,K\}\kappa T(\beta_T + \gamma_T)^2)^{1/4}$, for large enough $T$, we can obtain

$$
\begin{aligned}
\mathcal{R}^\pi(T) &= \mathcal{O}\left( \zeta(\beta_T + \gamma_T) \sum_{t \in [T]} \sum_{n \in [N]} \mathbb{E}[Q_n(t)] + \frac{NK}{\kappa\zeta^3(\beta_T + \gamma_T)} + N\log(T) \right) \\
&= \mathcal{O}\left( \frac{\zeta(\beta_T + \gamma_T)\min\{N,K\}T}{\epsilon} + \frac{NK}{\kappa\zeta^3(\beta_T + \gamma_T)} + N\log(T) \right) \\
&= \mathcal{O}\left( \frac{(\beta_T + \gamma_T)^{1/2}T^{3/4}(NK\min\{N,K\}^3)^{1/4}}{\kappa^{1/4}\epsilon^{3/4}} + N\log(T) \right) \\
&= \widetilde{\mathcal{O}}\left( \left(\frac{d^2 NK\min\{N,K\}^3}{\kappa^2\epsilon^3}\right)^{1/4} T^{3/4} \right).
\end{aligned}
$$

## A.8 Proof of Lemma 1

We first define

$$
\bar{f}_{k,t}(\theta) = \mathbb{E}_y[f_{k,t}(\theta)|\mathcal{F}_{t-1}] \text{ and } \bar{g}_{k,t}(\theta) = \mathbb{E}_y[g_{k,t}(\theta)|\mathcal{F}_{t-1}],
$$

where $\mathcal{F}_{t-1}$ is the filtration contains outcomes for time $s$ such that $s \leq t-1$ and $y = \{y_{n,t} : n \in S_{k,t}\}$. From Lemma 10 in Oh and Iyengar [44], by taking expectation over $y$ gives

$$
\bar{f}_{k,t}(\hat{\theta}_{k,t}) \leq \bar{f}_{k,t}(\theta_k) + \bar{g}_{k,t}(\hat{\theta}_{k,t})^\top (\hat{\theta}_{k,t} - \theta_k) - \frac{\kappa}{2}(\theta_k - \hat{\theta}_{k,t})^\top W_{k,t}(\theta_k - \hat{\theta}_{k,t}),
$$

where $W_{k,t} = \sum_{n \in S_{k,t}} x_n x_n^\top$. Then with $\bar{f}_{k,t}(\theta_k) \leq \bar{f}_{k,t}(\hat{\theta}_{k,t})$ from Lemma 12 in Oh and Iyengar [44], we have

$$
\begin{aligned}
0 \leq{} & \bar{f}_{k,t}(\hat{\theta}_{k,t}) - \bar{f}_{k,t}(\theta_k) \\
\leq{} & \bar{g}_{k,t}(\hat{\theta}_{k,t})^\top (\hat{\theta}_{k,t} - \theta_k) - \frac{\kappa}{2} \|\theta_k - \hat{\theta}_{k,t}\|^2_{W_{k,t}} \\
={} & g_{k,t}(\hat{\theta}_{k,t})^\top (\hat{\theta}_{k,t} - \theta_k) - \frac{\kappa}{2} \|\theta_k - \hat{\theta}_{k,t}\|^2_{W_{k,t}} + (\bar{g}_{k,t}(\hat{\theta}_{k,t}) - g_{k,t}(\hat{\theta}_{k,t}))^\top (\hat{\theta}_{k,t} - \theta_k) \\
\leq{} & \frac{1}{2} \|g_{k,t}(\hat{\theta}_{k,t})\|^2_{V_{k,t+1}^{-1}} + \frac{1}{2} \|\hat{\theta}_{k,t} - \theta_k\|^2_{V_{k,t+1}} - \frac{1}{2} \|\hat{\theta}_{k,t+1} - \theta_k\|^2_{V_{k,t+1}} \\
& - \frac{\kappa}{2} \|\theta_k - \hat{\theta}_{k,t}\|^2_{W_{k,t}} + (\bar{g}_{k,t}(\hat{\theta}_{k,t}) - g_{k,t}(\hat{\theta}_{k,t}))^\top (\hat{\theta}_{k,t} - \theta_k) \\
\leq{} & 2 \max_{n \in S_{k,t}} \|x_n\|^2_{V_{k,t+1}^{-1}} + \frac{1}{2} \|\hat{\theta}_{k,t} - \theta_k\|^2_{V_{k,t+1}} - \frac{1}{2} \|\hat{\theta}_{k,t+1} - \theta_k\|^2_{V_{k,t+1}} \\
& - \frac{\kappa}{2} \|\theta_k - \hat{\theta}_{k,t}\|^2_{W_{k,t}} + (\bar{g}_{k,t}(\hat{\theta}_{k,t}) - g_{k,t}(\hat{\theta}_{k,t}))^\top (\hat{\theta}_{k,t} - \theta_k) \\
\leq{} & 2 \max_{n \in S_{k,t}} \|x_n\|^2_{V_{k,t+1}^{-1}} + \frac{1}{2} \|\hat{\theta}_{k,t} - \theta_k\|^2_{V_{k,t}} + \frac{\kappa}{4} \|\hat{\theta}_{k,t} - \theta_k\|^2_{W_{k,t}} - \frac{1}{2} \|\hat{\theta}_{k,t+1} - \theta_k\|^2_{V_{k,t+1}} \\
& - \frac{\kappa}{2} \|\theta_k - \hat{\theta}_{k,t}\|^2_{W_{k,t}} + (\bar{g}_{k,t}(\hat{\theta}_{k,t}) - g_{k,t}(\hat{\theta}_{k,t}))^\top (\hat{\theta}_{k,t} - \theta_k) \\
\leq{} & 2 \max_{n \in S_{k,t}} \|x_n\|^2_{V_{k,t+1}^{-1}} + \frac{1}{2} \|\hat{\theta}_{k,t} - \theta_k\|^2_{V_{k,t}} - \frac{\kappa}{4} \|\hat{\theta}_{k,t} - \theta_k\|^2_{W_{k,t}} - \frac{1}{2} \|\hat{\theta}_{k,t+1} - \theta_k\|^2_{V_{k,t+1}} \\
& + (\bar{g}_{k,t}(\hat{\theta}_{k,t}) - g_{k,t}(\hat{\theta}_{k,t}))^\top (\hat{\theta}_{k,t} - \theta_k), \tag{67}
\end{aligned}
$$

where the third inequality comes from Lemma 11 in Oh and Iyengar [44] and the fourth inequality comes from Lemma 13 in Oh and Iyengar [44]. We note that, although our estimator lies in $\Theta$, we can still utilize Lemma 11 from Oh and Iyengar [44] by following the same proof steps.

Hence, from the above, we have

$$
\|\hat{\theta}_{k,t+1} - \theta_k\|^2_{V_{k,t+1}} \leq 4 \max_{n \in S_{k,t}} \|x_n\|^2_{V_{k,t+1}^{-1}} + \|\hat{\theta}_{k,t} - \theta_k\|^2_{V_{k,t}} - \frac{\kappa}{2} \|\hat{\theta}_{k,t} - \theta_k\|^2_{W_{k,t}} + 2(\bar{g}_{k,t}(\hat{\theta}_{k,t}) - g_{k,t}(\hat{\theta}_{k,t}))^\top (\hat{\theta}_{k,t} - \theta_k).
$$

Then using telescoping by summing the above over $t$, with at least probability $1 - \delta$, we have

$$
\begin{aligned}
\|\hat{\theta}_{k,t+1} - \theta_k\|^2_{V_{k,t+1}} \leq{} & \lambda + 4 \sum_{s=1}^{t} \max_{n \in S_{k,s}} \|x_n\|^2_{V_{k,s+1}^{-1}} - \frac{\kappa}{2} \sum_{s=1}^{t} \|\hat{\theta}_{k,s} - \theta_k\|^2_{W_{k,s}} \\
& + 2 \sum_{s=1}^{t} (\bar{g}_{k,s}(\hat{\theta}_{k,s}) - g_{k,s}(\hat{\theta}_{k,s}))^\top (\hat{\theta}_{k,s} - \theta_k) \\
\leq{} & \lambda + 4 \sum_{s=1}^{t} \max_{n \in S_{k,s}} \|x_n\|^2_{V_{k,s+1}^{-1}} - \frac{\kappa}{2} \sum_{s=1}^{t} \|\hat{\theta}_{k,s} - \theta_k\|^2_{W_{k,s}} \\
& + \frac{\kappa}{2} \sum_{s=1}^{t} \|\theta_k - \hat{\theta}_{k,t}\|^2_{W_{k,s}} + \frac{2C_1}{\kappa} \log((\log tL) t^2 K / \delta) \\
\leq{} & \lambda + 4 \sum_{s=1}^{t} \max_{n \in S_{k,s}} \|x_n\|^2_{V_{k,s+1}^{-1}} + \frac{2C_1}{\kappa} \log((\log tL) t^2 K / \delta) \\
\leq{} & \lambda + 16(d/\kappa) \log(1 + (tL/d\lambda)) + \frac{2C_1}{\kappa} \log((\log tL) t^2 K / \delta), \tag{68}
\end{aligned}
$$

where the second inequality is obtained from Lemma 14 in Oh and Iyengar [44] and the last one is obtained from Lemma 4

Then with $\delta = 1/t^2$, we can conclude that

$$
\|\hat{\theta}_{k,t} - \theta_k\|_{V_{k,t}} \leq C_1 \sqrt{\lambda + \frac{d}{\kappa} \log(1 + tLK/d\lambda)}.
$$

## A.9 $\alpha$-approximation Oracle

In this section, we provide a detailed explanation for $\alpha$-approxiamtion oracle to reduce the computation. Instead of obtaining the exact solution, the $\alpha$-approximation oracle, denoted by $\mathbb{O}^\alpha$, outputs $\{S_k^\alpha\}_{k\in[K]}$ satisfying $\sum_{k\in[K]} f_k(S_k^\alpha) \geq \max_{\{S_k\}_{k\in[K]}\in\mathcal{M}(\mathcal{N}_t)} \sum_{k\in[K]} \alpha f_k(S_k)$. Such an oracle can be constructed using a straightforward greedy policy as outlined in prior work [26, 10]. Then for assortments $\{S_{k,t}^\alpha\}_{k\in[K]}$ for $t\in[T]$ from Algorithms 1 and 2 using $\mathbb{O}^\alpha$, we can obtain the same queue length bounds and regret bounds for $\alpha$-regret in Theorems 1, 2, 3, and 4 under an $\alpha$-slackness assumption, respectively.

We first consider the following traffic slackness assumption with $0 < \alpha < 1$ instead of Assumption 3.

**Assumption 4.** *For some traffic slackness $0 < \epsilon < 1$, for each $t\in[T]$, there exists $\{S_{k,t}\}_{k\in[K]}\in \mathcal{M}(\mathcal{N})$ which satisfies $\lambda_n + \epsilon \leq \alpha\mu(n|S_{k,t},\theta_k)$ for all $n\in S_{k,t}$ and $k\in[K]$.*

### A.9.1 $\alpha$-approximation Oracle for Algorithm 1

Here we introduce an algorithm (Algorithm 3) by modifying Algorithm 1 using an $\alpha$-approximation oracle. We explain the distinct parts of the algorithm as follows. We define an oracle $\mathbb{O}^\alpha$, which outputs $\{S_{k,t}^\alpha\}_{k\in[K]}$ satisfying

$$\max_{\{S_k\}_{k\in[K]}\in\mathcal{M}(\mathcal{N}_t)} \sum_{k\in[K]}\sum_{n\in S_k} \alpha Q_n(t)\widetilde{\mu}_t^{UCB}(n|S_k,\hat{\theta}_{k,t}) \leq \sum_{k\in[K]}\sum_{n\in S_{k,t}^\alpha} Q_n(t)\widetilde{\mu}_t^{UCB}(n|S_{k,t}^\alpha,\hat{\theta}_{k,t}). \tag{69}$$

---

**Algorithm 3** $\alpha$-approximated UCB-Queueing Matching Bandit

**Input:** $\lambda, \kappa, C_1 > 0$
**for** $t = 1,\ldots,T$ **do**
    **for** $k\in[K]$ **do**
        $\hat{\theta}_{k,t} \leftarrow \mathrm{argmin}_{\theta\in\Theta}\, g_{k,t}(\hat{\theta}_{k,t-1})^\top(\theta - \hat{\theta}_{k,t-1}) + \frac{1}{2}\|\theta - \hat{\theta}_{k,t-1}\|_{V_{k,t}}^2$
    $\{S_{k,t}^\alpha\}_{k\in[K]} \leftarrow \mathbb{O}^\alpha$ from (69)
    Offer $\{S_{k,t}^\alpha\}_{k\in[K]}$ and observe preference feedback $y_{n,t}\in\{0,1\}$ for all $n\in S_{k,t}$, $k\in[K]$

---

For the stability analysis, we provide the following theorem.

**Theorem 5.** *The time average expected queue length of Algorithm 3 is bounded as*

$$\mathcal{Q}(T) = \mathcal{O}\left(\frac{\min\{N,K\}}{\epsilon} + \frac{1}{T}\frac{d^2 N^2 K^2}{\kappa^4\epsilon^6}\mathrm{polylog}(T)\right),$$

*which implies that the algorithm achieves stability as*

$$\lim_{T\to\infty}\mathcal{Q}(T) = \mathcal{O}\left(\frac{\min\{N,K\}}{\epsilon}\right).$$

*Proof.* Here we provide only the proof parts which are different from Theorem 1. We analyze the Lyapunov drift as follows.

$$\sum_{t\in[T]}\mathcal{V}(\mathbf{Q}(t+1)) - \mathcal{V}(\mathbf{Q}(t))$$

$$= \sum_{t\in[T]}\sum_{n\in[N]}(Q_n(t) + A_n(t) - D_n(t))^{+2} - Q_n(t)^2$$

$$= \sum_{t\in[T]}\sum_{n\in[N]}(Q_n(t) + A_n(t) - \alpha D_n^*(t))^2 - Q_n(t)^2$$

$$+ \sum_{t\in[T]}\sum_{n\in[N]}(Q_n(t) + A_n(t) - D_n(t))^{+2} - \sum_{t\in[T]}\sum_{n\in[N]}(Q_n(t) + A_n(t) - \alpha D_n^*(t))^2. \tag{70}$$

For the first two terms in Eq.(70), by following the same procedure of Eqs.(16) and (17), under Assumption 4, we can obtain

$$\sum_{t\in[T]}\sum_{n\in[N]}\mathbb{E}[(Q_n(t)+A_n(t)-\alpha D_n^*(t))^2-Q_n(t)^2]$$

$$\leq \sum_{t\in[T]}\sum_{n\in[N]}2\mathbb{E}[(\lambda_n-\alpha\mu(n|S_{k,t}^*,\theta_k))Q_n(t)]+2NT$$

$$\leq -\sum_{t\in[T]}\sum_{n\in[N]}2\epsilon\mathbb{E}[Q_n(t)]+2\min\{N,K\}T, \tag{71}$$

For the last two terms in Eq.(18), by following the steps for Eq.(20), we have

$$\sum_{t\in[T]}\sum_{n\in[N]}(Q_n(t)+A_n(t)-D_n(t))^{+2}-\sum_{t\in[T]}\sum_{n\in[N]}(Q_n(t)+A_n(t)-\alpha D_n^*(t))^2$$

$$\leq 4\sum_{t\in[T]}\sum_{n\in[N]}A_n(t)+2\sum_{t\in[T]}\sum_{n\in[N]}(\alpha D_n^*(t)-D_n(t))Q_n(t)+\min\{N,K\}T$$

$$\leq 2\sum_{t\in[T]}\sum_{n\in[N]}(\alpha D_n^*(t)-D_n(t))Q_n(t)+5\min\{N,K\}T. \tag{72}$$

We also have

$$\sum_{t\in[T]}\sum_{n\in[N]}\mathbb{E}[(\alpha D_n^*(t)-D_n(t)Q_n(t)]$$

$$\leq \sum_{t\in[T]}\sum_{n\in[N]}\mathbb{E}[(\alpha\mu(n|S_{k_{n,t}^*,t},\theta_{k_{n,t}^*})-\mu(n|S_{k_{n,t},t}^\alpha,\theta_{k_{n,t}}))Q_n(t)\mathbb{1}(E_t^1\cap E_{n,t}^2)]$$

$$+\sum_{t\in[T]}\sum_{n\in[N]}\mathbb{E}[(\alpha\mu(n|S_{k_{n,t}^*,t},\theta_{k_{n,t}^*})-\mu(n|S_{k_{n,t},t}^\alpha,\theta_{k_{n,t}}))Q_n(t)(\mathbb{1}((E_t^1)^c)+\mathbb{1}((E_{n,t}^2)^c))].$$

$$\tag{73}$$

Then for the first term of Eq.(73), we have

$$\sum_{t\in[T]}\mathbb{E}\left[\sum_{n\in[N]}(\alpha\mu(n|S_{k_{n,t}^*,t},\theta_{k_{n,t}^*})-\mu(n|S_{k_{n,t},t}^\alpha,\theta_{k_{n,t}}))Q_n(t)\mathbb{1}(E_t^1\cap E_{n,t}^2)\right]$$

$$\leq \sum_{t\in[T]}\mathbb{E}\left[\sum_{n\in[N]}(\alpha\widetilde{\mu}_t^{UCB}(n|S_{k_{n,t}^*,t},\hat{\theta}_{k_{n,t}^*,t})-\mu(n|S_{k_{n,t},t}^\alpha,\theta_{k_{n,t}}))Q_n(t)\mathbb{1}(E_t^1\cap E_{n,t}^2)\right]$$

$$\leq \sum_{t\in[T]}\mathbb{E}\left[\sum_{n\in[N]}(\widetilde{\mu}_t^{UCB}(n|S_{k_{n,t},t}^\alpha,\hat{\theta}_{k_{n,t},t})-\mu(n|S_{k_{n,t},t}^\alpha,\theta_{k_{n,t}}))Q_n(t)\mathbb{1}(E_t^1\cap E_{n,t}^2)\right]$$

$$\leq \sum_{t\in[T]}\mathbb{E}\left[\sum_{n\in[N]}2\beta_t\|x_n\|_{V_{k_{n,t},t}^{-1}}Q_n(t)\mathbb{1}(E_t^1\cap E_{n,t}^2)\right]$$

$$\leq C_2\epsilon\sum_{t\in[T]}\sum_{n\in[N]}\mathbb{E}[Q_n(t)]. \tag{74}$$

The rest of the proofs can be easily obtained from the proof steps in Theorem 1. $\qquad\square$

Now, we investigate the regret of Algorithm 3. The $\alpha$-regret regarding policy $\pi$ is defined as

$$\mathcal{R}^{\alpha,\pi}(T)=\sum_{t\in[T]}\sum_{n\in[N]}\mathbb{E}\left[(\alpha\mu(n|S_{k_{n,t}^*,t},\theta_{k_{n,t}^*})-\mu(n|S_{k_{n,t},t},\theta_{k_{n,t}}))Q_n(t)\right].$$

The algorithm achieves the following regret bound.

**Theorem 6.** *The policy $\pi$ of Algorithm 3 achieves a regret bound of*

$$\mathcal{R}^{\alpha,\pi}(T) = \widetilde{\mathcal{O}}\left(\min\left\{\frac{d}{\kappa}\sqrt{KT}Q_{\max}, \left(\frac{dNK\min\{N,K\}^3}{\kappa^2\epsilon^3}\right)^{1/4}T^{3/4}\right\}\right).$$

*Proof.* In this proof, we provide only the parts that are different from the proof of Theorem 2.

We first provide the proof for regret bound of $\mathcal{R}^{\alpha,\pi}(T) = \widetilde{\mathcal{O}}\left(\frac{d}{\kappa}\sqrt{KT}Q_{\max}\right)$.

We can show that

$$\mathcal{R}^{\alpha,\pi}(T) = \sum_{t\in[T]}\sum_{n\in[N]}\mathbb{E}[Q_n(t)(\alpha\mu(n|S_{k^*_{n,t},t},\theta_{k^*_{n,t}}) - \mu(n|S^\alpha_{k_{n,t},t},\theta_{k_{n,t}}))]$$

$$= \sum_{t\in[T]}\sum_{n\in[N]}\mathbb{E}[Q_n(t)(\alpha\mu(n|S_{k^*_{n,t},t},\theta_{k^*_{n,t}}) - \mu(n|S^\alpha_{k_{n,t},t},\theta_{k_{n,t}}))\mathbb{1}(E^1_t)]$$

$$+ \sum_{t\in[T]}\sum_{n\in[N]}\mathbb{E}[Q_n(t)(\alpha\mu(n|S_{k^*_{n,t},t},\theta_{k^*_{n,t}}) - \mu(n|S^\alpha_{k_{n,t},t},\theta_{k_{n,t}}))\mathbb{1}((E^1_t)^c)]$$

$$= \sum_{t\in[T]}\sum_{n\in[N]}\mathbb{E}[Q_n(t)(\alpha\mu(n|S_{k^*_{n,t},t},\theta_{k^*_{n,t}}) - \mu(n|S^\alpha_{k_{n,t},t},\theta_{k_{n,t}}))\mathbb{1}(E^1_t)] + \sum_{t\in[T]}\sum_{n\in[N]}t\mathbb{P}((E^1_t)^c)]$$

$$\leq \sum_{t\in[T]}\sum_{n\in[N]}\mathbb{E}[Q_n(t)(\alpha\widetilde{\mu}^{UCB}_t(n|S_{k^*_{n,t},t},\hat{\theta}_{k^*_{n,t},t}) - \mu(n|S^\alpha_{k_{n,t},t},\theta_{k_{n,t}}))\mathbb{1}(E^1_t)] + \mathcal{O}(N/T)$$

$$\leq \sum_{t\in[T]}\sum_{n\in[N]}\mathbb{E}[Q_n(t)((\widetilde{\mu}^{UCB}_t(n|S^\alpha_{k_{n,t},t},\hat{\theta}_{k_{n,t},t}) - \mu(n|S^\alpha_{k_{n,t},t},\theta_{k_{n,t}}))\mathbb{1}(E^1_t)] + \mathcal{O}(N/T)$$

$$\leq \sum_{t\in[T]}\mathbb{E}[\sum_{k\in[K]}\sum_{n\in S_{k,t}}Q_n(t)(\widetilde{\mu}^{UCB}_t(n|S^\alpha_{k,t}) - \mu_t(n|S^\alpha_{k,t}))\mathbb{1}(E^1_t)] + \mathcal{O}(N/T)$$

$$\leq 2\mathbb{E}\left[\beta_T\sum_{t\in[T]}\sum_{k\in[K]}\max_{n\in S^\alpha_{k,t}}\|x_n\|_{V^{-1}_{k,t}}Q_n(t)\right] + \mathcal{O}(N/T)$$

$$\leq 2\mathbb{E}\left[\max_{t\in[T],n\in[N]}Q_n(t)\beta_T\sum_{t\in[T]}\sum_{k\in[K]}\max_{n\in S^\alpha_{k,t}}\|x_n\|_{V^{-1}_{k,t}}\right] + \mathcal{O}(N/T)$$

$$\leq 2\mathbb{E}\left[\max_{t\in[T],n\in[N]}Q_n(t)\beta_T\sqrt{KT\sum_{t\in[T]}\sum_{k\in[K]}\max_{n\in S^\alpha_{k,t}}\|x_n\|^2_{V^{-1}_{k,t}}}\right] + \mathcal{O}(N/T)$$

$$= \widetilde{\mathcal{O}}\left(\frac{d}{\kappa}\sqrt{KT}Q_{\max}\right).$$

By following the similar steps above and the proofs for Theorem 2, we can easily obtain the worst-case regret bound of $\mathcal{R}^{\alpha,\pi}(T) = \widetilde{\mathcal{O}}\left(\left(\frac{dNK\min\{N,K\}^3}{\kappa^2\epsilon^3}\right)^{1/4}T^{3/4}\right)$, which conclude the proof. $\square$

### A.9.2 $\alpha$-approximation Oracle for Algorithm 2

We can obtain similar results for Algorithm 2 as in the case of Algorithm 3. Here we introduce an algorithm (Algorithm 4) by modifying Algorithm 2 using an $\alpha$-approximation oracle. We explain the distinct parts of the algorithm as follows. We define an oracle $\mathbb{O}^\alpha$, which outputs $\{S^\alpha_{k,t}\}_{k\in[K]}$ satisfying

$$\max_{\{S_k\}_{k\in[K]}\in\mathcal{M}(\mathcal{N}_t)}\sum_{k\in[K]}\sum_{n\in S_k}\alpha Q_n(t)\widetilde{\mu}^{TS}_t(n|S_k,\{\widetilde{\theta}^{(i)}_{k,t}\}_{i\in[M]}) \leq \sum_{k\in[K]}\sum_{n\in S^\alpha_{k,t}}Q_n(t)\widetilde{\mu}^{TS}_t(n|S^\alpha_{k,t},\{\widetilde{\theta}^{(i)}_{k,t}\}_{i\in[M]}).$$

(75)

**Algorithm 4** $\alpha$-approximated Thompson Sampling-Queueing Matching Bandit

---

**Input:** $\lambda, M, \kappa, C_1 > 0$

**for** $t = 1, \ldots, T$ **do**

    **for** $k \in [K]$ **do**

        $\hat{\theta}_{k,t} \leftarrow \text{argmin}_{\theta \in \Theta} \, g_{k,t}(\hat{\theta}_{k,t-1})^{\top}(\theta - \hat{\theta}_{k,t-1}) + \frac{1}{2}\|\theta - \hat{\theta}_{k,t-1}\|^2_{V_{k,t}}$

        Sample $\{\widetilde{\theta}^{(i)}_{k,t}\}_{i \in [M]}$ independently from $\mathcal{N}(\hat{\theta}_{k,t}, \beta_t^2 V_{k,t}^{-1})$

    $\{S^\alpha_{k,t}\}_{k \in [K]} \leftarrow \mathbb{O}^\alpha$ from (75)

    Offer $\{S^\alpha_{k,t}\}_{k \in [K]}$ and observe preference feedback $y_{n,t} \in \{0, 1\}$ for all $n \in S_{k,t}$, $k \in [K]$

---

For the stability analysis, we provide the following theorem.

**Theorem 7.** *The time average expected queue length of Algorithm 4 is bounded as*

$$\mathcal{Q}(T) = \mathcal{O}\left(\frac{\min\{N, K\}}{\epsilon} + \frac{1}{T}\frac{d^4 N^2 K^2}{\kappa^4 \epsilon^6}\text{polylog}(T)\right),$$

*which implies that the algorithm achieves stability as*

$$\lim_{T \to \infty} \mathcal{Q}(T) = \mathcal{O}\left(\frac{\min\{N, K\}}{\epsilon}\right).$$

*Proof.* Here we provide only the proof parts which are different from Theorem 3. We analyze the Lyapunov drift as follows with Assumption 4.

$$\sum_{t \in [T]} \mathbb{E}\left[\mathcal{V}(\mathbf{Q}(t+1)) - \mathcal{V}(\mathbf{Q}(t))\right]$$

$$= \sum_{t \in [T]} \sum_{n \in [N]} \mathbb{E}\left[(Q_n(t) + A_n(t) - D_n(t))^{+2} - Q_n(t)^2\right]$$

$$= \sum_{t \in [T]} \sum_{n \in [N]} \mathbb{E}\left[(Q_n(t) + A_n(t) - \alpha D_n^*(t))^2 - Q_n(t)^2\right]$$

$$+ \sum_{t \in [T]} \sum_{n \in [N]} \mathbb{E}\left[(Q_n(t) + A_n(t) - D_n(t))^{+2} - (Q_n(t) + A_n(t) - \alpha D_n^*(t))^2\right]$$

$$\leq 7 \min\{N, K\}T - \sum_{t \in [T]} \sum_{n \in [N]} 2\epsilon\mathbb{E}[Q_n(t)] + 2 \sum_{t \in [T]} \sum_{n \in [N]} \mathbb{E}\left[(\alpha D_n^*(t) - D_n(t))Q_n(t)\right], \quad (76)$$

Then, for bounding Eq.(76), we have

$$\sum_{t \in [T]} \sum_{n \in [N]} \mathbb{E}[(\alpha D_n^*(t) - D_n(t)Q_n(t)]$$

$$\leq \sum_{t \in [T]} \sum_{n \in [N]} \mathbb{E}[(\alpha\mu(n|S_{k^*_{n,t},t}, \theta_{k^*_{n,t}}) - \mu(n|S^\alpha_{k_{n,t},t}, \theta_{k_{n,t}}))Q_n(t)\mathbb{1}(E_t^1 \cap E_t^2 \cap E_{n,t}^3)]$$

$$+ \sum_{t \in [T]} \sum_{n \in [N]} \mathbb{E}[(\alpha\mu(n|S_{k^*_{n,t},t}, \theta_{k^*_{n,t}}) - \mu(n|S^\alpha_{k_{n,t},t}, \theta_{k_{n,t}}))Q_n(t)(\mathbb{1}((E_t^1)^c) + \mathbb{1}((E_t^2)^c) + \mathbb{1}((E_{n,t}^3)^c))].$$

$$(77)$$

We provide a bound for the first term of Eq.(77). We first have

$$\sum_{t \in [T]} \sum_{n \in [N]} \mathbb{E}[(\alpha\mu(n|S_{k^*_{n,t},t}, \theta_{k^*_{n,t}}) - \mu(n|S^\alpha_{k_{n,t},t}, \theta_{k_{n,t}}))Q_n(t)\mathbb{1}(E_t^1 \cap E_t^2 \cap E_{n,t}^3)]$$

$$\leq \sum_{t \in [T]} \sum_{n \in [N]} \mathbb{E}[(\alpha\mu(n|S_{k^*_{n,t},t}, \theta_{k^*_{n,t}}) - \widetilde{\mu}_t^{TS}(n|S^\alpha_{k_{n,t},t})$$

$$+ \widetilde{\mu}_t^{TS}(n|S^\alpha_{k_{n,t},t}) - \mu(n|S^\alpha_{k_{n,t},t}, \theta_{k_{n,t}}))Q_n(t)\mathbb{1}(E_t^1 \cap E_t^2 \cap E_{n,t}^3)].$$

$$(78)$$

Now we provide a bound for the first two terms in Eq.(78).

We define sets

$$\widetilde{\Theta}_t = \left\{ \{\theta_k^{(i)}\}_{i\in[M],k\in[K]} : \left|\max_{i\in[M]} x_n^\top \theta_k^{(i)} - x_n^\top \hat{\theta}_{k,t}\right| \le \gamma_t \|x_n\|_{V_{k,t}^{-1}}; \ \forall n \in [N], \forall k \in [K] \right\} \text{ and}$$

$$\widetilde{\Theta}_t^{opt} = \left\{ \{\theta_k^{(i)}\}_{i\in[M],k\in[K]} : \sum_{k\in[K]} \sum_{n\in S_{k,t}^\alpha} \widetilde{\mu}_t^{TS}(n|S_{k,t}^\alpha, \{\theta_k^{(i)}\}_{i\in[M]}) Q_n(t) \right.$$
$$\left. > \sum_{k\in[K]} \sum_{n\in S_{k,t}^*} \alpha\mu(n|S_{k,t}^*, \theta_k) Q_n(t) \right\} \cap \widetilde{\Theta}_t.$$

Then we define event $E_t(\widetilde{\theta}) = \{\{\widetilde{\theta}_{k,t}^{(i)}\}_{i\in[M],k\in[K]} \in \widetilde{\Theta}_t^{opt}\}$. Recall $h_{n,k,t}^{TS} = \max_{i\in[M]} x_n^\top \widetilde{\theta}_{k,t}^{(i)}$. Then we have

$$\mathbb{E}\left[\mathbb{E}\left[\sum_{k\in[K]} \left(\sum_{n\in S_{k,t}^*} \alpha\mu(n|S_{k,t}^*, \theta_k) Q_n(t) - \sum_{n\in S_{k,t}^\alpha} \widetilde{\mu}_t^{TS}(n|S_{k,t}^\alpha, \{\widetilde{\theta}_{k,t}^{(i)}\}_{i\in[M]}) Q_n(t)\right) \mathbb{1}(E_t^1 \cap E_t^2 \cap E_{n,t}^3)|\mathcal{F}_{t-1}, E_t(\widetilde{\theta})\right]\right]$$

$$\le \mathbb{E}\left[\mathbb{E}\left[\left(\sum_{k\in[K]} \sum_{n\in S_{k,t}^*} \alpha\mu(n|S_{k,t}^*, \theta_k) Q_n(t)\right.\right.\right.$$

$$\left.\left.\left. - \inf_{\{\theta_l^{(i)}\}_{i\in[M],l\in[K]}\in\widetilde{\Theta}_t} \sum_{k\in[K]} \sum_{n\in S_{k,t}^\alpha} \widetilde{\mu}_t^{TS}(n|S_{k,t}^\alpha, \{\theta_k^{(i)}\}_{i\in[M]}) Q_n(t)\right) \mathbb{1}(E_t^1 \cap E_t^2 \cap E_{n,t}^3)|\mathcal{F}_{t-1}, E_t(\widetilde{\theta})\right]\right]$$

$$\le \mathbb{E}\left[\mathbb{E}\left[\left(\sum_{k\in[K]} \sum_{n\in S_{k,t}^\alpha} \widetilde{\mu}_t^{TS}(n|S_{k,t}^\alpha, \{\widetilde{\theta}_{k,t}^{(i)}\}_{i\in[M]})\right.\right.\right.$$

$$\left.\left.\left. - \inf_{\{\theta_l^{(i)}\}_{i\in[M],l\in[K]}\in\widetilde{\Theta}_t} \sum_{k\in[K]} \sum_{n\in S_{k,t}^\alpha} \widetilde{\mu}_t^{TS}(n|S_{k,t}^\alpha, \{\theta_k^{(i)}\}_{i\in[M]})\right) Q_n(t)\mathbb{1}(E_t^1 \cap E_t^2 \cap E_{n,t}^3)|\mathcal{F}_{t-1}, E_t(\widetilde{\theta})\right]\right]$$

$$= \frac{2\gamma_t \epsilon}{C_2(\gamma_t + \beta_t)} \mathbb{E}\left[\sum_{k\in[K]} \sum_{n\in S_{k,t}^\alpha} \mathbb{E}\left[Q_n(t)|\mathcal{F}_{t-1}, E_t(\widetilde{\theta}), E_t^1\right] \mathbb{P}(E_t^1|\mathcal{F}_{t-1})\right],$$

$$(79)$$

We provide a lemma below for further analysis.

**Lemma 10.** *For all $t \in [T]$, we have*

$$\mathbb{P}\left(\sum_{k\in[K]} \sum_{n\in S_{k,t}^\alpha} \widetilde{\mu}_t^{TS}(n|S_{k,t}) Q_n(t) > \sum_{k\in[K]} \sum_{n\in S_{k,t}^*} \alpha\mu(n|S_{k,t}^*, \theta_k) Q_n(t)|\mathcal{F}_{t-1}, E_t^1\right) \ge 1/4\sqrt{e\pi}$$

*Proof.* Given $\mathcal{F}_{t-1}$, $x_n^\top \widetilde{\theta}_{k,t}^{(i)}$ follows Gaussian distribution with mean $x_n^\top \hat{\theta}_{k,t}$ and standard deviation $\beta_t \|x_n\|_{V_{k,t}^{-1}}$. Then we have

$$\mathbb{P}\left(\max_{i\in[M]} x_n^\top \widetilde{\theta}_{k,t}^{(i)} > x_n^\top \theta_k|\mathcal{F}_{t-1}, E_t^1\right) = 1 - \mathbb{P}\left(x_n^\top \theta_{k,t}^{(i)} \le x_n^\top \theta_k; \forall i \in [M]|\mathcal{F}_{t-1}, E_t^1\right)$$

$$= 1 - \mathbb{P}\left(Z_i \le \frac{x_n^\top \theta_k - x_n^\top \hat{\theta}_{k,t}}{\beta_t \|x_n\|_{V_{k,t}^{-1}}}; \forall i \in [M]|\mathcal{F}_{t-1}, E_t^1\right)$$

$$\ge 1 - \mathbb{P}\left(Z \le 1\right)^M,$$

where $Z_i$ and $Z$ are standard normal random variables. Then we can show that

$$\mathbb{P}\left(\sum_{k\in[K]}\sum_{n\in S_{k,t}^\alpha}\widetilde{\mu}_t^{TS}(n|S_{k,t}^\alpha)Q_n(t) > \sum_{k\in[K]}\sum_{n\in S_{k,t}^*}\alpha\mu(n|S_{k,t}^*,\theta_k)Q_n(t)|\mathcal{F}_{t-1},E_t^1\right)$$

$$\geq \mathbb{P}\left(\sum_{k\in[K]}\sum_{n\in S_{k,t}^\alpha}\alpha\widetilde{\mu}_t^{TS}(n|S_{k,t}^\alpha)Q_n(t) > \sum_{k\in[K]}\sum_{n\in S_{k,t}^*}\alpha\mu(n|S_{k,t}^*,\theta_k)Q_n(t)|\mathcal{F}_{t-1},E_t^1\right)$$

$$\geq \mathbb{P}\left(\sum_{k\in[K]}\sum_{n\in S_{k,t}^*}\alpha\widetilde{\mu}_t^{TS}(n|S_{k,t}^*)Q_n(t) > \sum_{k\in[K]}\sum_{n\in S_{k,t}^*}\alpha\mu(n|S_{k,t}^*,\theta_k)Q_n(t)|\mathcal{F}_{t-1},E_t^1\right)$$

$$= \mathbb{P}\left(\sum_{k\in[K]}\sum_{n\in S_{k,t}^*}\widetilde{\mu}_t^{TS}(n|S_{k,t}^*)Q_n(t) > \sum_{k\in[K]}\sum_{n\in S_{k,t}^*}\mu(n|S_{k,t}^*,\theta_k)Q_n(t)|\mathcal{F}_{t-1},E_t^1\right)$$

$$\geq \mathbb{P}\left(\max_{i\in[M]} x_n^\top\widetilde{\theta}_{k,t}^{(i)} > x_n^\top\theta_k; \forall n \in S_{k,t}^*, \forall k \in [K]|\mathcal{F}_{t-1},E_t^1\right)$$

$$\geq 1 - LK\mathbb{P}(Z \leq 1)^M$$

$$\geq 1 - LK(1 - 1/4\sqrt{e\pi})^M$$

$$\geq \frac{1}{4\sqrt{e\pi}},$$

where the second last inequality is obtained from $\mathbb{P}(Z \leq 1) \leq 1 - 1/4\sqrt{e\pi}$ using the anti-concentration of standard normal distribution, and the last inequality comes from $M = \lceil 1 - \frac{\log KL}{\log(1-1/4\sqrt{e\pi})} \rceil$. □

The rest of the proof can be easily obtained by following the proof steps in Theorem 3. □

Now, we investigate the regret of Algorithm 4. The $\alpha$-regret regarding policy $\pi$ is defined as

$$\mathcal{R}^{\alpha,\pi}(T) = \sum_{t\in[T]}\sum_{n\in[N]}\mathbb{E}\left[(\alpha\mu(n|S_{k_{n,t}^*,t}^*,\theta_{k_{n,t}^*}) - \mu(n|S_{k_{n,t},t},\theta_{k_{n,t}}))Q_n(t)\right].$$

The algorithm achieves the following regret bound.

**Theorem 8.** *The policy $\pi$ of Algorithm 4 achieves a regret bound of*

$$\mathcal{R}^{\alpha,\pi}(T) = \widetilde{\mathcal{O}}\left(\min\left\{\frac{d^{3/2}}{\kappa}\sqrt{KT}Q_{\max}, \left(\frac{d^2NK\min\{N,K\}^3}{\kappa^2\epsilon^3}\right)^{1/4}T^{3/4}\right\}\right).$$

*Proof.* Here we provide only the proof parts which are different from Theorem 4.

We first provide the proof for the regret bound of $\widetilde{\mathcal{O}}\left(\frac{d^{3/2}}{\kappa}\sqrt{KT}Q_{\max}\right)$. We have

$$\mathcal{R}^\pi(T) = \sum_{t\in[T]}\sum_{n\in[N]}\mathbb{E}[Q_n(t)(\alpha\mu(n|S_{k_{n,t}^*,t}^*,\theta_{k_{n,t}^*}) - \mu(n|S_{k_{n,t},t}^\alpha,\theta_{k_{n,t}}))]$$

$$\leq \sum_{t\in[T]}\sum_{n\in[N]}\mathbb{E}[Q_n(t)(\alpha\mu(n|S_{k_{n,t}^*,t}^*,\theta_{k_{n,t}^*}) - \widetilde{\mu}_t^{TS}(n|S_{k_{n,t},t}^\alpha) + \widetilde{\mu}_t^{TS}(n|S_{k_{n,t},t}^\alpha) - \mu(n|S_{k_{n,t},t}^\alpha,\theta_{k_{n,t}}))\mathbb{1}(E_t^1 \cap E_t^2)]$$

$$+ \sum_{t\in[T]}\sum_{n\in[N]}\mathbb{E}[Q_n(t)(\alpha\mu(n|S_{k_{n,t}^*,t}^*,\theta_{k_{n,t}^*}) - \widetilde{\mu}_t^{TS}(n|S_{k_{n,t},t}^\alpha) + \widetilde{\mu}_t^{TS}(n|S_{k_{n,t},t}^\alpha) - \mu(n|S_{k_{n,t},t}^\alpha,\theta_{k_{n,t}}))\mathbb{1}((E_t^1)^c)]$$

$$+ \sum_{t\in[T]}\sum_{n\in[N]}\mathbb{E}[Q_n(t)(\alpha\mu(n|S_{k_{n,t}^*,t}^*,\theta_{k_{n,t}^*}) - \widetilde{\mu}_t^{TS}(n|S_{k_{n,t},t}^\alpha) + \widetilde{\mu}_t^{TS}(n|S_{k_{n,t},t}^\alpha) - \mu(n|S_{k_{n,t},t}^\alpha,\theta_{k_{n,t}}))\mathbb{1}((E_t^2)^c)].$$

By following the similar proof steps in Theorem 7, we can obtain

$$\sum_{t\in[T]}\sum_{n\in[N]}\mathbb{E}[Q_n(t)(\alpha\mu(n|S_{k^*_{n,t},t},\theta_{k^*_{n,t}}) - \widetilde{\mu}^{TS}_t(n|S^\alpha_{k_{n,t},t})$$

$$+ \widetilde{\mu}^{TS}_t(n|S^\alpha_{k_{n,t},t}) - \mu(n|S^\alpha_{k_{n,t},t},\theta_{k_{n,t}}))\mathbb{1}(E^1_t\cap E^2_t)]$$

$$\leq C_2 \sum_{t\in[T]}\sum_{k\in[K]}\mathbb{E}\left[\max_{n\in S^\alpha_{k,t}}(\gamma_t+\beta_t)\|x_n\|_{V^{-1}_{k,t}}Q_n(t)\right].$$

The rest of the proof can be easily obtained from the proof steps in Theorem 4 by incorporating techniques in the proof of Theorem 7. □

## A.10 Additional Experiments

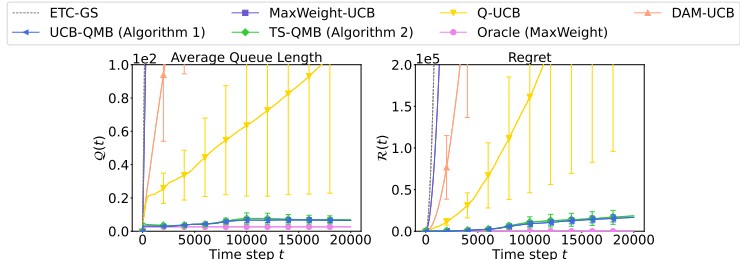

Figure 3: Experimental results with $N = 4, K = 3, L = 2, d = 2$ for (left) average queue length and (right) regret

## A.11 Notation Table

Table 1: Notation Table.

| | |
|---|---|
| $T$ | Time horizon |
| $N$ | Number of agents (queues) |
| $K$ | Number of arms (servers) |
| $d$ | Dimension of model parameters and features |
| $x_n$ | Feature vector of agent $n$ |
| $\theta_k$ | Model parameter vector of arm $k$ |
| $Q_n(t)$ | Length of agent $n$ at time $t$ |
| $\epsilon$ | Traffic slackness parameter |
| $\mu(n|S_k,\theta_k)$ | Service rate of arm $k$ with $\theta_k$ for a job in agent $n$ given assortment $S_k$ |
| $n_0$ | Null agent |
| $\lambda_n$ | Job arrival rate for agent $n$ |
| $k^\pi_{n,t}(= k_{n,t})$ | A server that agent $n$ is assigned at time $t$ according to $\pi$ (simply $k_{n,t}$) |
| $A_n(t)$ | The number of arrival jobs in agent $n$ at time $t$; Random variable with mean $\lambda_n$ |
| $D_n(t|S_k)$ | The number of departure job in agent $n$ by arm $k$ given $S_k$; Random variable with mean $\mu(n|S_k,\theta_k)$ |
| $\mathcal{Q}(T)$ | Average queue lengths over horizon time $T$ |
| $\mathcal{R}^\pi(T)$ | Cumulative regret under $\pi$ over $T$ |
| $\kappa$ | Regularity parameter for MNL |
| $\mathcal{N}_t$ | Set of non-empty agents at time $t$ |
| $\mathcal{M}()$ | Set of feasible disjoint assortments given a set of agents |
| $S_{k,t}$ | Set of agents assigned to arm $k$ by policy $\pi$ |
| $S^*_{k,t}$ | Set of agents assigned to arm $k$ by the oracle $\pi^*$ |
| $k^{\pi^*}_{n,t}(= k^*_{n,t})$ | A server that agent $n$ is assigned by $\pi^*$ at time $t$ |
| $y_{n,t}$ | Preference feedback, $\{0,1\}$, for assigned agent $n$ at time $t$ |

