# OpenReview forum: "Queueing Matching Bandits with Preference Feedback"
_NeurIPS.cc/2024/Conference — NeurIPS 2024 poster_

### Official Review · Reviewer_PAuc · 2024-07-09

**Soundness:** 3
**Presentation:** 2
**Contribution:** 3
**Rating:** 4
**Confidence:** 3

**Summary:**

This paper proposes an algorithm that learn optimal allocation in a multi-arm bandit problem involving queues.
The algorithm  ensures the system stability while having a sub-linear regret.

**Strengths:**

The algorithm leverages on both max-weight and UCB (or TS) to provide stability and no-regret in a multi-arm bandit problem involving queueing.

**Weaknesses:**

- There are notation confusions.

 - The use of $\mu_{n,k} (S| \theta)$, $\mu_t(n | S, \theta)$, $\mu (n | S, \theta)$, $\mu (n|S)$  make reading the technical parts very hard to understand and confusing

-V is the classical Lyapounov function and also used for the norm of $x$.
The norm $|| . ||_{V_{k,y}^{-1}} $ is used for both $\theta$ ( a matrix) and $x_n$ (a vector) while only defined using  $x$. This is not clear to me.


- Some constants are not well explained, for example, $\lambda$ is used in line 225 and 227 without any explanation, then disappears in all further results.

**Questions:**

1. The authors explain that the notion of stability that they use is the existence of Cezaro limit of the queue sizes. This is weaker than what they call uniform stability (expected queue lengths are bounded).
Then they introduce $ Q_{max}$ the expectation of the maximum queue length over $T$ steps. $Q_{max}$  appears in the regret bound as a constant independent of $T$. This seems to imply that the stability is uniform. But this is not the case here. Can you explain where is the catch?

**Limitations:**

The authors claim that their model fill up a gap between previous models and real-applications. This claimed is not backed up by any concrete example. Furthermore, the assumptions that they add about Bernoulli arrivals, unit size jobs, information patterns known/unknown  by the learner, the technical assumptions  (esp. assumption 3 that is strictly stronger than a stability condition because it involves the fixed allocation $\mu$), and the need of well-defined posterior distributions for their algorithm does not go in the direction of practical applications.



The stability assumption (assumption 3) uses the predefined allocation $\mu$ while in general it should not depend on the decisions of the scheduler.

---

> ### Author Rebuttal · Authors · 2024-08-07
>
> We appreciate your time to review our paper and comments. Below, we address each comment.
>
> **There are notation confusions:**
>
> We appreciate your helpful comments in clarifying our paper. We will include a notation table in our final (Please refer to Table 1 in the attached PDF). As mentioned in line 238, we use $\mu(n|S_k)$ instead of $\mu(n|S_k,\theta_k)$ for simplicity when there is no confusion. Additionally, we have identified some notation errors: $\mu_t(n|S_k,\theta_k)$, $\mu_{n,k}(S_k|\theta_k)$ should each be replaced by $\mu(n|S_k,\theta_k)$. We will correct these in the final version. To reduce notation confusion, we will also use $\mathcal{V}$ for the Lyapunov function in the final.
>
> **$||\cdot||\_{V\_{k,t}^{-1}}$ is used for $\theta$ without definition for a matrix:**
>
> Parameter $\theta$ for the model of service rate is a d-dimensional vector rather than a matrix. Therefore, $||\cdot||\_{V\_{k,t}^{-1}}$ can be defined for $\theta$.
>
> **$\lambda$ disappears in results:**
>
> As mentioned in lines 228 and 282, we set the regularizer factor $\lambda=1$, which does not affect the order of bounds for stability or regret. For instance, $\beta_t$ includes $\lambda$ as $\sqrt{\lambda+\frac{d}{\kappa}\log(1+tLK/(d\lambda))}$. The stability bound contains $O(\beta\_T^4)$ as in Eq.(6), which can be bounded by $O((d^2/\kappa^2)\text{polylog}(T))$ ignoring constant $\lambda$ for large enough $T$.
>
> **$Q_{max}$ appears in regret bound as a constant independent of $T$:**
>
> Our regret bounds of $\tilde{O}(\min\\{\sqrt{T}Q_{\max},T^{3/4}\\})$ consist of the two terms. We argue that the bound of $T^{3/4}$ must hold for the worst case and this does not have to imply that $Q_{\max}$ is a constant. The two terms in the regret bound are obtained from separate regret analyses as shown in proof sketch of Theorem 2. Indeed, to obtain the worst-case bound of $T^{3/4}$, we utilize our stability results for average queue lengths in Theorems 1,3, as briefly shown in Eq.(10) and following lines 271-273.
>
> **Evidence that the model fills up a gap between previous models and real-applications:**
>
> As mentioned in lines 40–49, in real-world, the service rate may be influenced by the relative preferences among multiple assigned jobs, as in ride-hailing platforms and online job markets. For example, in ride-hailing platforms, a driver may receive multiple rider requests and select one based on their preferences. Additionally, a job in an empty queue cannot be assigned to a server for feedback, reflecting real-world where a non-existent rider cannot be recommended to a driver. Lastly, real-world scenarios often involve observable feature information, such as details about locations of riders. All of these practical considerations have not been addressed in the previous literature while addressed in our model.
>
> **Concerns about assumptions:**
>
> **Bernoulli arrivals with unit size jobs:** The Bernoulli arrival with unit job model has been considered in queueing bandit [15, 48, 32], and our setting follows this. However, we can easily extend binary random arrival $A_n(t)\in\\{0,1\\}$ to the case where random variable $A_n(t)$ lies in $[0,M]$ for some $M>0$ with mean $\lambda_n\in[0,M]$. This extension allows for multi-unit jobs rather than just single-unit jobs.  For instance, under the extended setting, Eq. (2) becomes $\sum_t\sum_n \mathbb{E}[(Q_n(t)+A_n(t)-D_n^*(t))^2-Q_n(t)^2] \le -\sum_t\sum_n 2\epsilon \mathbb{E}[Q_n(t)]+(M^2+1)NT.$ This leads to achieve stability with $\mathcal{Q}(T)=O((M^2+1)N/\epsilon)$.
>
> **Information patterns:**  We assume that features for jobs are known, but the service rate for each server is unknown. The unknown service rates have been considered in almost all queueing matching bandits [32,52,46,15,23,48]. Importantly, our setting is more general than the existing works because we use a feature-based model for the service rate while the previous work typically considers a non-contextual MAB setting. Additionally, the known feature information is the most standard setting in many contextual bandits [1,2,3,5,33,43,44].
>
> **Assumption 3 is stronger than a stability condition and depends on the scheduler decision:** We emphasize that Assumption 3 is *not strictly stronger* than a stability condition (Definition 1). This is because, through Assumption 3, we can only guarantee the existence of assortments $S\_k$'s such that, for some $0<\epsilon<1$, $\lambda_n+\epsilon \le \mu(n|S_k,\theta_k)$ for $n\in S_k$ and $k\in[K]$.  For instance, in the worst case, there is a unique allocation assortment $S_k$ satisfying $\lambda_n+\epsilon \le \mu(n|S\_k,\theta\_k)$ for all $n\in S_k$ and $k\in[K]$. In our setting of unknown $\theta_k$, our algorithm does not know such $S\_k$'s which satisfies the Assumption 3. Therefore, our algorithms require finding such allocation assortment $S\_k$ for stabilizing queues by learning $\theta_k$ over horizon time $T$ from the feedback of the algorithm's decision $S\_{k,t}$ at each time $t$. Also, note that the existence of assortments $S_k$ in Assumption 3 is *regardless of* our scheduler $S\_{k,t}$.  We highlight that the nature of it is not different from the stability assumptions in [15,46,32,52], in which they assume the existence of allocations such that arrival rates are less than service rates with $\epsilon$.
>
> **Need of well-defined posterior distributions for their algorithm:** We respectfully disagree with your comment. In this comment, you may be referring to our Thompson sampling algorithm (Algorithm 2). We also clarify that we do not impose any specific distribution on the true parameter $\theta_k$. Our analysis in this work is in the frequentist regret where $\theta_k$'s are fixed, which holds for the worst-case scenario under Assumptions 1–3. *Hence, there are no prior or posterior assumptions.* Our TS-based algorithm uses a Gaussian distribution sampling, but this does not imply that the true parameter must follow a particular distribution.

---

> > ### Author Response · Authors · 2024-08-13
> >
> > Dear Reviewer PAuc,
> >
> > Thank you again for taking the time to review our paper. We sincerely hope our responses have adequately addressed your questions and comments. If they have, we kindly ask you to reconsider the rating. If you have any last-minute questions, we would be more than happy to answer them.
> >
> >
> > Sincerely,
> > Authors

---

### Official Review · Reviewer_9oLk · 2024-07-12

**Soundness:** 3
**Presentation:** 3
**Contribution:** 3
**Rating:** 7
**Confidence:** 1

**Summary:**

This study examines multi-class multi-server asymmetric queueing systems, where jobs arrive randomly, and unknown job-server service rates are modeled by a feature-based Multinomial Logit (MNL) function. The proposed UCB and Thompson Sampling algorithms aim to stabilize the queues while learning service rates, achieving system stability with an average queue length bound and sublinear regret, and demonstrating their performance through experimental results.

**Strengths:**

The authors propose a novel and practical framework for queueing matching bandits, introducing feature-based multinomial logit functions for service rates and preference feedback, which are investigated for the first time. Their UCB and Thompson Sampling algorithms achieve stability with average queue length bounds and sublinear regret, outperforming previously suggested methods, as demonstrated by their experimental results.

**Weaknesses:**

The quality of the figures can be improved.

**Questions:**

How about the performance and calculation efficiency when N, K, L, and d become larger?

---

> ### Author Rebuttal · Authors · 2024-08-07
>
> We appreciate your feedback and positive evaluation of our work. Below, we address each of your comments.
>
>
> **The quality of the figures can be improved:**
>
> We will make sure to improve the quality of the figures in our final version.
>
> **How about the performance and calculation efficiency when N, K, L, and d become larger?**
>
> We provided stability and regret analysis in Theorems 1,2,3, and 4. Specifically, our algorithms (Algorithms 1 and 2) achieve stability with $\mathcal{Q}(T)=\mathcal{O}(N/\epsilon)$ for large $T$. They also achieves regret bound of $\mathcal{R}(T)=\mathcal{O}(\min\\{(d/\kappa)\sqrt{KT}Q_{\max},N(dK/\epsilon^3\kappa)^{1/4}T^{3/4}\\})$ and $\mathcal{R}(T)=\mathcal{O}(\min\\{(d^{3/2}/\kappa)\sqrt{KT}Q_{\max},N(d^2K/\epsilon^3\kappa)^{1/4}T^{3/4}\\})$, respectively.
>
> Regarding computational cost, the exact combinatorial optimization is NP-hard (so is true for almost all other combinatorial optimization problems). However, as mentioned in lines 295-298, we can overcome these computational issues using an $\alpha$-approximation oracle, as described in Appendix A.8. In summary, by employing a straightforward greedy policy [26, 10] with the $\alpha$-approximation oracle to solve the combinatorial problem, under an approximated stability assumption (Assumption 4), our algorithms (Algorithms 3 and 4) achieve both stability and a sublinear $\alpha$-regret bound. Please refer to Appendix A.8 for more details.

---

### Official Review · Reviewer_117Z · 2024-07-13

**Soundness:** 3
**Presentation:** 3
**Contribution:** 3
**Rating:** 7
**Confidence:** 4

**Summary:**

The work introduces a new bandits framework that handles the problem of matching queuing jobs (agents) with preferential servers (arms). The work extends beyond the match making problem with the objective of stabilizing the queue by learning the preferential nature of agents to arms. The authors propose two new algorithms based on UCB and Thompson sampling algorithms and analyze theoretical guarantees with respect to stabilizing the length of the queues. They also extend their study by analyzing the regret bound for the proposed algorithms.

**Strengths:**

* The authors develops a new bandits formulation for addressing queueing matching fixing some of the nuances of the earlier developed framework for the same.

* The work captures the important aspect of this problem with a structured model using features for estimating service rates in comparison with bandits for queues or earlier works. Also, the consideration of availability of assignment of an agent available only in an non-empty queue is an welcome addition.

* The formulation is reasonable and interesting mathematically and the work studies the stability of the queueing matching problem and analyses the theoretical guarantees with regards to regret as well.

* The authors developed two new algorithms for the framework and showed theoretical bounds for the same highlighting the stability comparing it with the existing work.

**Weaknesses:**

* The developed bandits framework is an incremental extension of bandits for queues & match making problem discussed in the work. Though the work highlights some important consideration, the previous algorithm seem to work for the proposed algorithm.

* In the experiments, the authors have only studied the setting when N=KL and One does wonder why the other regime N<KL is not included.

**Questions:**

* How does this work in terms of theory compares with some of the existing algorithm in this problem space. A discussion on regret comparison can help draw conclusion on the performance of the algorithms in this work.

* The experimental section compares the authors’ proposed algorithm QCB-QMB and TS-QMB with others algorithms that are applicable to this problem. The experiments compare them in one settings when N=KL. Also, experimental setting when N<KL can help the behavior of the proposed algorithms in different operating regimes.

* The regret analysis in Theorems 2 and 4 provides bounds in terms of the maximum queue length $Q_{max}$. But, the relationship between $Q_{max}$ and the system parameters N, K, \& \epsilon is not clearly established. Providing insights on Q_{max} in terms of these parameters would give more insight into the regret performance.

* The experimental results in Section 6 compare the proposed algorithms with previously suggested methods for queueing bandits or matching bandits. However, the choice of baselines seems limited. Providing comparisons with a wider range of relevant algorithms, such as those handling contextual bandits with queueing constraints, would strengthen the empirical evaluation.

* The paper establishes sublinear regret bounds, but it does not provide any lower bounds on the regret. Deriving problem-specific lower bounds would give a better understanding of the optimality of the proposed algorithms and highlight any potential gaps between the upper and lower bounds.

**Limitations:**

* The proposed framework is evolutionary extension to the existing framework of bandits for queues & match making problem.

* The experimental section only studies setting in certain regimes when N=KL and lacks detailed empirical study and a proper baseline algorithms to be compared.

---

> ### Author Rebuttal · Authors · 2024-08-07
>
> We appreciate your feedback and positive evaluation of our work. Below, we address each of your comments.
>
> **The previous algorithm seems to work for the proposed algorithm:**
>
> Our problem includes novel factors, which are inspired by the real-world such as online job markets or ride-hailing platforms. *Due to the unique aspects of our problem, the existing algorithms (either in queuing bandits or in matching bandits) do not provide a tight regret guarantee in our problem*. Below, we outline the novelty of our problem compared to previous works.
>
> In queueing bandits, to our best knowledge, none of the previous works has focused on a structured model with features for service rates. Furthermore, earlier studies either did not allow the assignment of different types of jobs to a server simultaneously [32, 52], or, if they did [46,15], the service rates for each job-server assignment remained constant regardless of the entire assignment. In contrast, we adopt a structured model incorporating features for unknown asymmetric service rates. We also enable the assignment of multiple agents to the same server, where the server stochastically selects at most one of the assigned agents based on its undisclosed (relative) preference. More details can be found in lines 98–114.
>
> Regarding matching bandits, the primary goal of the previous work [34,45,35,9,54,29] has been to minimize regret by achieving an optimal stable matching through learning agents' side preferences via stochastic reward feedback under static settings and deterministic arm behavior with known preferences. In contrast, we consider dynamic environments with job arrivals for agents. Additionally, we propose that arm behavior is stochastic with unknown preferences, requiring the learning of arms' preferences. Our main objective is to stabilize queue lengths rather than finding a stable matching. Hence, the existing matching bandit methods cannot solve our problem. More details are provided in lines 120–129.
>
> As mentioned in the introduction, it is also worth mentioning online matching problems [27,39,40,18,16,28], where the sole focus is on optimizing matching rather than learning underlying models from bandit feedback. In contrast, our study concentrates on bandits related to learning latent utilities by managing the exploration&exploitation tradeoff while establishing the stability of the queueing systems. Importantly online matching problems do not address the stability of systems.
>
> **In the experiments, $N<KL$ is not included:**
>
> Our model is a preference-based structured model, allowing that multi-agent can be assigned to an arm, which aligns with real-world applications such as the online job market, where multiple applicants (agents) are applying to employers (arms). Therefore, with $N$ agents and $K$ arms, we used an instance of $N>K$ with $N=4$, $K=2$, and $L=2$, in our experiments.
> Incorporating your feedback, we have included additional experiments for $N<KL$ in the attached pdf, where our algorithm still performs superior to the other algorithms (except the oracle that knows the true preference). We will include these additional experimental results in the final.
>
> **A discussion on regret comparison with other work:**
>
> First, we emphasize that previous studies on multi-class multi-server systems closely related to our work, [46,15,52], have not considered regret analysis, whereas our work addresses both stability and regret analysis. Additionally, the algorithms proposed in prior studies do not apply to our setting, as we use a structured model with features and allow multiple queues to be assigned to a server with preference feedback. Therefore, a direct theoretical comparison is not available. However, we provide an existing result for regret to clarify our contribution.
>
> There exists a prior work [23] that studied regret analysis regarding $Q_n(t)\mu(n|S_k,\theta_k)$, as in our regret definition, under single-server.  In [23], a regret bound is provided as $\tilde{O}(\max\\{\sqrt{T}Q_{\max},T^{3/4}\\})$ (with $\delta=1/2$ for a stationary setting in Lemma 5.7). In the previous work, the worst-case bound is not guaranteed from the maximum in regret. We emphasize that our algorithms achieve a sublinear regret bound of $\tilde{O}(\min\\{\sqrt{T}Q_{\max},T^{3/4}\\})$ (Theorems 2,4), even in the worst-case from the minimum in regret. We also note that our algorithms achieve stability results (Theorems 1,3) (specifically, $\mathcal{Q}(T)=O(N/\epsilon)$ when $T$ is sufficiently large), which match that for the Oracle (Proposition 1).
>
>  **Providing insights on $Q_{max}$ in regret in terms of model parameters:**
>
>  In our analysis of the regret bound of $\min\\{\sqrt{T}Q_{\max},T^{3/4}\\}$ (Theorems 2,4), we allow for $Q_{\max}$ to be instance-dependent. However, regardless of how $Q_{\max}$ is bounded, we can still obtain the worst-case sublinear regret bound of $T^{3/4}$ by using our stability results in Theorem 1 or 3 (More details are shown in the proof sketch in Theorem 2). While we can obtain the worst-case bound without a bound for $Q_{\max}$, it would be an interesting future work to analyze $Q_{\max}$ directly.
>
>  **The choice of baselines seems limited:**
>
> To our knowledge, we have considered appropriate baseline algorithms, such as [32, 15, 52], for multi-queue multi-server bandits with asymmetric service rates, which are the most closely related work. We also include the algorithm from [34] as a baseline, which is proposed for matching bandits. We would be more than happy to include additional baselines in our final if there are any others we need to consider.
>
> **It does not provide lower bounds on the regret:**
>
>  It remains an open problem to establish regret lower bounds, as mentioned in Appendix A.1. However, we highlight that the regret analysis has not been studied in closely related studies [46,15,52], and our sublinear regret could be the first step towards no regret algorithm for queuing matching bandits.

---

> > ### Comment · Reviewer_117Z · 2024-08-14
> >
> > Thank you for responding to my comments. The authors response to Q2 (understanding the performance in the experimental setting: N<KL) and Q3 (regret comparison with applicable algorithms) have clarified my initial concerns. I find the authors rebuttal useful and suggest including these rebuttal comments in the next version of this work.

---

> > > ### Author Response · Authors · 2024-08-14
> > >
> > > Thank you for maintaining a positive rating of our paper. We will incorporate the discussion, including the additional experiments and regret comparison, into our final version.

---

### Official Review · Reviewer_c5Ej · 2024-07-20

**Soundness:** 3
**Presentation:** 2
**Contribution:** 2
**Rating:** 6
**Confidence:** 2

**Summary:**

This paper studies queueing matching bandits. It proposes a framework that involves multiple queues and multiple servers: in each round, jobs arrive randomly at each queue; the learner assigns jobs to servers; and each server picks and serves its preferred job according to a feature-based linear model. The goal is to stabilize the system, so that the average queue length is finite as the horizon grows to infinity. The paper proposes two algorithms, one based on Upper Confidence Bound and the other based on Thompson sampling. They achieve system stability as well as sublinear regret.

**Strengths:**

- The proposed framework seems interesting and introduces a few novel elements, including a feature-based service function and a preference-based job assignment model. The setting is motivated by important real-world applications, such as ride-hailing platforms.

- The paper proposes two algorithms, and they both achieve the same asymptotic bound on the average queue length as an oracle baseline.

- The paper also provides regret analyses for the algorithms, which are not considered in the closely related works.

**Weaknesses:**

- Both algorithms require solving an NP-hard combinatorial optimization problem, which may be impractical in large-scale systems.

- The empirical evaluation is limited to synthetic data.

- The presentation can be improved. Since the setting is rather specialized, the first two sections can be challenging for readers who do not know much about queueing systems. For example, in the introduction section, it would be nice if the authors could provide explanations for key terms such as service rates, stability of systems, multi-class queueing, etc. In addition, in Section 5, there are detailed, step-by-step descriptions of the algorithms, but not a discussion on the intuition behind (which probably would not take too long).

- Different jobs from each queue have a common, known representation $x_n$, which may not reflect real-world scenarios.

**Questions:**

- While the paper presents regret bounds, it is unclear to me what they tell us beyond the stability result. Could you provide some intuition behind the definition of cumulative regret, specifically how $Q_n(t)$ is incorporated and what it signifies?

- Could you briefly discuss the technical challenges in the analysis due to having a feature-based service function and only preference feedback, given existing works on bandits for queues and MLN bandits?

**Limitations:**

Limitations have been discussed.

---

> ### Author Rebuttal · Authors · 2024-08-07
>
> We appreciate your feedback and positive evaluation of our work. Below, we address each comment.
>
> **Require solving an NP-hard optimization:**
>
> NP-hardness is common in almost all assortment (and many other combinatorial) optimization problems. Hence, we do not think that this is particularly a weakness of our method. Rather, our work includes the results under approximation optimization addressing this challenge! As mentioned in lines 295-298, we can overcome the computational challenges using an $\alpha$-approximation oracle. In short, by employing a greedy policy [26, 10] with the $\alpha$-approximation oracle to solve the combinatorial problem, and under an approximated stability assumption (Assumption 4), our algorithms (Algorithms 3 and 4) achieve stability and a sublinear $\alpha$-regret bound. Please refer to Appendix A.8 for more details.
>
> **The empirical evaluation is limited to synthetic data:**
>
> Almost all bandit literature uses synthetic data because real-time online feedback is required to compute regret in each round. Specifically, true reward values are needed for this evaluation. While offline datasets from the real world can be used, they eventually need to be adapted for the online setting, making them semi-synthetic. This adaptation assumes some true reward function, thus still classifying the experiment as "synthetic."
>
> We emphasize that the main contribution of this work is to propose a novel problem and propose new algorithms with theoretical analysis. We also point out that many works on bandits for queues [15,46,52,30,32] and on matching bandits [34,45,35,9,52,29] have conducted experiments solely on synthetic datasets, similar to our paper.
>
> **The presentation can be improved. For example..:**
>
> We sincerely appreciate your suggestions. We will incorporate your comments into our final version.
>
> **Different jobs from each queue have a common representation $x_n$:**
>
> While our work focuses on $x_n$ for $n \in [N]$, we emphasize that *previous queueing bandit literature [15, 52, 23, 46, 32] typically considers unstructured MAB models for service rates even without feature information at all.* To the best of our knowledge, our work is the first to consider a structured model with features for service rates.  An extension to varying features with respect to time would be an interesting direction for future work. A more detailed explanation of the novelty of our setting is provided in lines 98–114 of our paper.
>
> **Intuition behind the definition of cumulative regret, specifically how $Q_n(t)$ is incorporated:**
>
> The goal of this problem is to stabilize the queues in the system. To achieve this, we consider maximizing the weighted service rates $Q_n(t)\mu(n|S,\theta)$, where priority in the assortment scheduling is given to queues with either large queue lengths or high service rates. For instance, if the service rates are similar across queues, the optimization prioritizes scheduling queues with large queue lengths to stabilize the overall system.
>
> Then the oracle strategy of MaxWeight, denoted as $\\{S_{k,t}^*\\}\_{k\in[K]}$, is based on maximizing weighted service rate $\sum\_{k} \sum_{n\in S_{k,t}^*} Q_n(t) \mu(n|S_{k,t}^*, \theta_k)$ under known $\theta\_k$, which are known to have optimal throughput keeping the queues in networks stable [51, 38, 37, 49]. By analyzing regret, we can estimate the performance of our algorithms in comparison to the Oracle (MaxWeight) directly. We note that our algorithms (Algorithms 1,2), operated under unknown $\theta_k$, are based on this MaxWeight strategy by maximizing $\sum\_k \sum\_n Q\_n(t) \tilde{\mu\_t}^{UCB, TS}(n|S\_k, \hat{\theta}\_{k,t}) $ where $\tilde{\mu\_t}^{UCB, TS}$ is an estimator for the service rate. One natural way to compare our algorithm against the oracle is to measure the difference between $\sum_{k\in[K]}\sum_{n\in S_{k,t}^*}Q_n(t)\mu(n|S_{k,t}^*,\theta_k)$ and $\sum_{k\in[K]}\sum_{n\in S_k}Q_n(t)\mu(n|S_{k,t},\theta_k)$, as in our regret definition, where $S_{k,t}$ is the output of our algorithms. We point out that the closely related work on queueing bandits [46, 15, 52] only conducted stability analysis, without considering regret analysis.
>
> **Technical challenges in the analysis:**
>
> We emphasize that this is the first work to connect the two distinct problems of queues and MNL bandits. We note that previous queueing bandits did not deal with preference feedback, and previous MNL bandits did not deal with stability analysis under a dynamic environment.  Therefore, we needed to design a framework that integrates the two different problems.
>
> In the stability analysis, it is crucial how to relate a term regarding queue lengths and a term regarding learning MNL model. For this purpose, we examine the Lyapunov function drift, and derive a bound regarding the summation of queue lengths and learning part in the MNL bandit strategy, as shown in Eqs.(2), (3). By further analyzing the learning parts as in Eqs. (4),(5), and (6), and by rearranging the results, we can finally obtain a bound for average queue lengths.
>
> For regret analysis, due to our regret definition, we need to consider not only the difference between the service rate of our algorithm and oracle, but also the queue length of the system at each time, which is the main difference from previous MNL bandits. We first derive a regret bound of $\widetilde{\mathcal{O}}(\sqrt{T}Q_{\max})$ using $Q_{\max}$ with the techniques from bandits. Next, to establish the worst-case bound of $\tilde{O}(T^{3/4})$, we integrate the regret analysis with the queue length bound results of our stability analysis, by carefully analyzing the event of bounded elliptical potentials with the optimally tuned parameter $\zeta$ as in Eqs. (9), (10). We believe our framework for regret analysis could be useful for this line of future research.

---

> > ### Comment · Reviewer_c5Ej · 2024-08-12
> >
> > Thank you. I have read your responses and I intend to maintain my score.

---

> > > ### Author Response · Authors · 2024-08-12
> > > **Thank you for your comment**
> > >
> > > Thank you for maintaining a positive rating on our paper. We will incorporate the discussion, including the presentation suggestions, into our final version.

---

### Author Rebuttal · Authors · 2024-08-07

Thank you for taking the time to review our paper. We are encouraged by your feedback, noting that our problem is both novel and interesting, inspired by real-world challenges. We have attached a PDF that includes additional experimental results and a notation table, addressing the reviewers' comments. Below, we provide detailed responses to each of the reviewers' comments.

---

### Decision · Program_Chairs · 2024-09-25

**Decision:**

Accept (poster)

**Comment:**

The paper proposes a queueing matching bandits framework using UCB and Thompson sampling algorithms to stabilize multi-class, multi-server systems and achieve sublinear regret. Reviewers acknowledged the framework's novelty, emphasizing its structured feature-based model and practical applications. However, concerns included computational complexity, presentation clarity, limited empirical data, and reliance on synthetic experiments. Despite these issues, most reviewers supported acceptance, highlighting the theoretical contribution and potential impact, while one reviewer recommended rejection due to confusion over notation and assumptions. The authors addressed these concerns, suggesting improvements and clarifying their model’s practical relevance.  Overall, the paper is sufficiently novel, makes interesting contributions and is acceptable to NeurIPS.